# Isotopic evidence for alteration of nitrous oxide emissions and producing pathways contribution under nitrifying conditions

Guillaume Humbert[1, 2, *], Mathieu Sébilo[1, 3], Justine Fiat[4], Longqi Lang[5], Ahlem Filali[4], Véronique Vaury[1], Mathieu Spérandio[5], Anniet M. Laverman[2]

[1]Sorbonne Université, CNRS, INRA, IRD, UPD, UPEC, Institute of Ecology and Environmental Sciences – Paris, iEES, F-75005 Paris, France
[2]Centre National de la Recherche Scientifique (CNRS), ECOBIO – UMR 6553, Université de Rennes, 35042 Rennes, France
[3]CNRS/Univ. Pau & Pays Adour/E2S UPPA, Institut des Sciences Analytiques et de Physico-Chimie pour l'Environnement et les Matériaux, UMR 5254, 64000, Pau, France
[4]Irstea, UR PROSE, CS 10030, F-92761, Antony Cedex, France
[5]LISBP, Université de Toulouse, CNRS, INRA, INSA, Toulouse, France

*Correspondence to*: Guillaume Humbert (g.humbert86@gmail.com)

**Abstract.** Nitrous oxide ($N_2O$) emissions by a nitrifying biofilm reactor were investigated with $N_2O$ isotopocules. The nitrogen isotopomer site preference of $N_2O$ ($^{15}N$-SP) indicated the contribution of producing and consuming pathways in response to changes in oxygenation level (from 0 to 21 % $O_2$ in the gas mix), temperature (from 13.5 to 22.3 °C), and ammonium concentrations (from 6.2 to 62.1 mg N $L^{-1}$). Nitrite reduction, either nitrifier-denitrification or heterotrophic denitrification, was the main $N_2O$ producing pathway under the tested conditions. Difference between oxidative and reductive rates of nitrite consumption was discussed in relation to $NO_2^-$ concentrations and $N_2O$ emissions. Hence, nitrite oxidation rates seem to decrease as compared to ammonium oxidation rates at temperatures above 20 °C and under oxygen-depleted atmosphere, increasing $N_2O$ production by the nitrite reduction pathway. Below 20 °C, a difference in temperature sensitivity between hydroxylamine and ammonium oxidation rates is most likely responsible for an increase in the $N_2O$ production via the hydroxylamine oxidation pathway (nitrification). A negative correlation between the reaction kinetics and the apparent isotope fractionation was additionally shown from the variations of $\delta^{15}N$ and $\delta^{18}O$ values of $N_2O$ produced from ammonium. The approach and results obtained here, for a nitrifying biofilm reactor under variable environmental conditions, should allow application and extrapolation on $N_2O$ emissions from other systems such as lakes, soils and sediments.

## 1 Introduction

Nitrogen (N) cycling lies on numerous biological processes exploited and altered by anthropic activities (Bothe et al., 2007). One of the major issues related to N cycle alteration is the production of nitrous oxide ($N_2O$) a potent ozone-depleting and greenhouse gas whose emissions exponentially increased during the industrial era (Crutzen et al., 1979; IPCC, 2014; Ravishankara et al., 2009). Wastewater resource recovery facilities (WRRFs) contribute to about 3 % of annual global anthropogenic $N_2O$ sources (ca. 6.7 ± 1.3 Tg N-N2O in 2011; IPCC, 2014); with 0 to 25 % of the influent nitrogen loads

emitted as N$_2$O (Law et al., 2012b). The challenges in mitigation of these emissions rely on the understanding of the N$_2$O producing processes and their controls.

Two microbial processes are responsible for the production of N$_2$O (nitrification and heterotrophic denitrification), with only one of these capable of consuming it (denitrification; Fig. 1a; Kampschreur et al., 2009). Nitrification is the oxidation of ammonium to nitrite (NO$_2^-$) via the intermediate hydroxylamine (NH$_2$OH) conducted by ammonia oxidizers and the subsequent oxidation of NO$_2^-$ to nitrate (NO$_3^-$) by nitrite oxidizers. During nitrification, N$_2$O can be produced as reaction side-product from hydroxylamine oxidation by biotic, abiotic or hybrid processes (Caranto et al., 2016; Heil et al., 2015; Terada et al., 2017). Heterotrophic denitrification and nitrifier-denitrification produce N$_2$O from nitrite reduction conducted by denitrifiers and ammonium oxidizers, respectively.

Temperature, electron donor and acceptor concentrations have been identified to control the N$_2$O emission from WRRFs (Bollon et al., 2016; Kampschreur et al., 2009; Tumendelger et al., 2014, 2016; Wunderlin et al., 2012). These variables may induce N$_2$O accumulation due to inhibition or disturbance of enzyme activity (Betlach and Tiedje, 1981; Kim et al., 2008; Otte et al., 1996). In addition to this, the different N$_2$O producing processes, nitrification, nitrifier-denitrification or heterotrophic denitrification, are rarely observed independently from each other in heterogeneous environments like wastewater, natural waters, soils or sediments. However, the understanding of the influence that environmental conditions have on the balance between these processes and the N$_2$O producing pathways remain to a large extent unexplored.

In order to decipher N$_2$O producing/consuming pathways, the analysis of N$_2$O isotopocules, molecules that only differ in either the number or position of isotopic substitutions, has been applied (Koba et al., 2009; Sutka et al., 2006) (Figs. 1b-d). The isotope composition of substrates and fractionation mechanisms influence both nitrogen and oxygen isotope ratios of N$_2$O (reported as $\delta^{15}$N and $\delta^{18}$O, respectively, Fig. 1b). Basically, the oxygen atom in the N$_2$O molecule produced by hydroxylamine oxidation originates from atmospheric dissolved oxygen with $\delta^{18}$O of 23.5 ‰ (Andersson and Hooper, 1983; Hollocher et al., 1981; Kroopnick and Craig, 1972), while the oxygen atom in N$_2$O produced by nitrite reduction originates from nitrite that has undergone oxygen-exchange with water (Kool et al., 2007; Snider et al., 2012). Nonetheless, the $\delta^{18}$O-N$_2$O resulting from the nitrite reduction conducted by the nitrifiers ranges from 13 to 35 ‰ (Snider et al., 2012). In contrast, the N$_2$O produced by the heterotrophic denitrifiers through the nitrite reduction pathway has $\delta^{18}$O over 35 ‰ (Snider et al., 2013). However, the O-exchange between the N$_2$O precursors and water can decrease it to values below 35 ‰ (Snider et al., 2015). Therefore, the $\delta^{18}$O alone does not enable differentiation between the N$_2$O producing pathways.

In combination with $\delta^{18}$O, the $\delta^{15}$N-N$_2$O allows to identify the N$_2$O producing pathways (Fig. 1b). However, the isotope fractionations (or isotope effects) largely influence the $\delta^{15}$N-N$_2$O due to wide variations between and within the reactions involved in the nitrogen cycle (Denk et al., 2017). The isotopic fractionation results from the difference in equilibrium constant or reaction rate observed between the heavier and lighter isotopes in both abiotic and biotic processes. The net isotope effects ($\Delta$) approximated from the difference between $\delta^{15}$N of product and substrate characterize the production of compounds resulting from sequential or branched reactions and have been recently reviewed (Denk et al., 2017; Toyoda et al., 2017). So

far, only two estimates of the net isotope effect of N$_2$O production by ammonium oxidation via hydroxylamine of -46.5 and -32.9 ‰ have been proposed (Sutka et al., 2006; Yamazaki et al., 2014). These values are imbricated between -52.8 and -6 ‰, the range of net isotope effects related to the N$_2$O production through nitrite reduction performed by nitrifiers or heterotrophic denitrifiers (Lewicka-Szczebak et al., 2014; Sutka et al., 2008).

Similarly to isotope ratios, the nitrogen isotopomer site preference ([15]N-SP), the difference between the relative abundances of N$_2$O molecules enriched in [15]N at central (N$^\alpha$) position and terminal (N$^\beta$) position, differ according to N$_2$O producing pathway (Figs. 1c and d). During heterotrophic or nitrifier-denitrification the [15]N-SP of N$_2$O produced from nitrate or nitrite ranges from -10.7 to 0.1 ‰, while ranging from 13.1 to 36.6 ‰ when N$_2$O results from hydroxylamine oxidation (Frame and Casciotti, 2010; Jung et al., 2014; Sutka et al., 2006; Yamazaki et al., 2014). Finally, N$_2$O reduction to N$_2$ by heterotrophic denitrifiers increases the values of $\delta^{15}$N, $\delta^{18}$O and [15]N-SP of residual N$_2$O with specific pairwise ratios (Jinuntuya-Nortman et al., 2008; Webster and Hopkins, 1996; Yamagishi et al., 2007).

Nitrogen and oxygen isotope ratios of N$_2$O have lower potential for N$_2$O source identification as compared to [15]N-SP. However, we believe that the use of both isotope approaches should strengthen the conclusions from [15]N-SP and reveal additional isotope effects (Fig. 1).

The aim of the current study is to improve our understanding regarding the effects of key environmental variables (oxygenation, temperature, NH$_4^+$ concentrations) on N$_2$O production and emission rates. More specifically using nitrogen and oxygen isotope ratios as well as [15]N-SP of N$_2$O should allow deciphering the different producing and consuming pathways under these different conditions. In order to achieve this, the nitrifying biomass of a submerged fixed-bed biofilm reactor was investigated. Among the wastewater treatment systems, the biofilm systems are adapted to large urban areas owing to their compactness, flexibility and reliability. An increase of their development is expected in response to the additional 2.5 billion humans predicted in urban areas by 2050 (United Nations Population Division, 2018). However, the biofilm systems have received much less attention than the suspended biomass systems and the relations between the N$_2$O producing/consuming pathways and controls remain largely unknown (Sabba et al., 2018; Todt and Dörsch, 2016). Although applied here to the nitrifying biomass of a WRRF, the research questions addressed regard a diversity of environments including natural waters, soils and sediments: i) Does the nitrifying biomass emit N$_2$O and what are the producing pathways at play?; ii) Do oxygenation, temperature, and NH$_4^+$ concentration alter the N$_2$O emissions, and what are the involved processes? We hypothesize that the isotope signature of N$_2$O allows identifying the N$_2$O origins and the assessment of pathway contribution to N$_2$O emissions. The results of this study should improve the mechanistic understanding as well as improved prediction of N$_2$O emissions from WRRFs, currently suffering from high uncertainty.

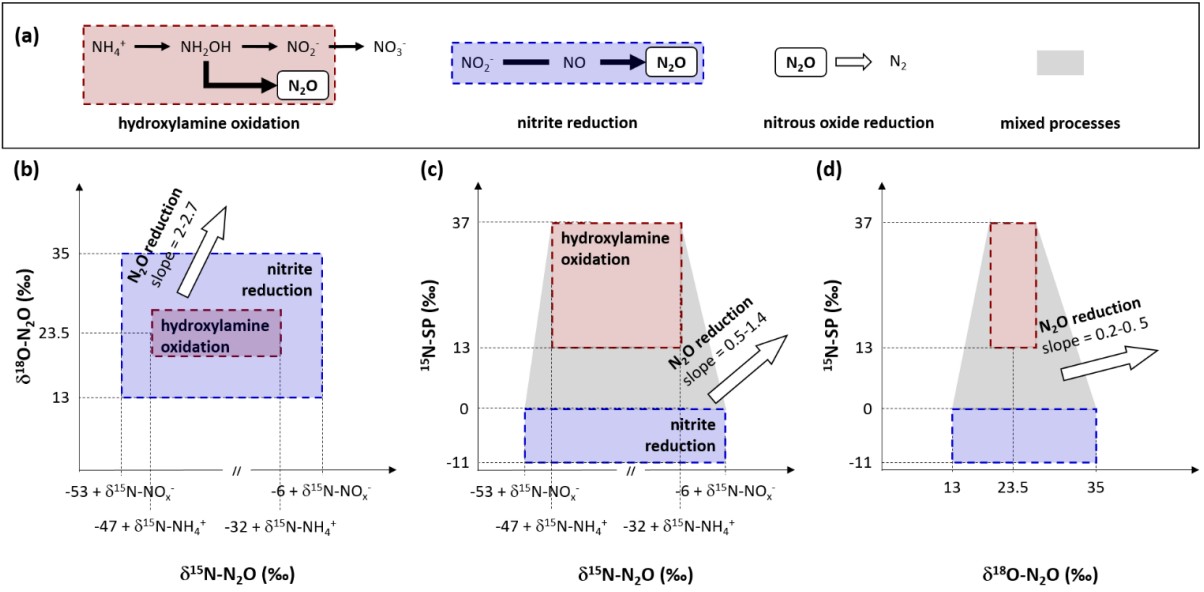

**Figure 1: N₂O producing and consuming pathways at play during nitrification and heterotrophic denitrification. Substrate isotope composition, isotope effects and $^{15}$N-SP values from the literature were used to propose the ranges of $^{15}$N (Lewicka-Szczebak et al., 2014; Sutka et al., 2006, 2008; Yamazaki et al., 2014), $^{18}$O (Andersson and Hooper, 1983; Hollocher et al., 1981; Kool et al., 2007; Kroopnick and Craig, 1972; Snider et al., 2012), and $^{15}$N-SP (Frame and Casciotti, 2010; Jung et al., 2014; Sutka et al., 2006; Yamazaki et al., 2014), as well the slopes relating them with each other during N₂O reduction to N₂ (Jinuntuya-Nortman et al., 2008; Webster and Hopkins, 1996; Yamagishi et al., 2007). The assumptions made and the calculations performed are detailed in the text.**

## 2 Material and methods

### 2.1 Experimental setup for nitrifying experiments

Experiments were carried out with colonized polystyrene beads (diameter 4 mm) sampled from the nitrification biologically active filters (BAF) of a domestic WRRF (Seine Centre, France). In this WRRF, wastewater (240,000 m³ d⁻¹) passes through a pre-treatment stage, followed by a physicochemical decantation, and tertiary biological treatment. The latter is composed of three biofiltration processes; (i) carbon elimination (24 Biofor®), (ii) nitrification (29 Biostyr®), and (iii) denitrification (12 Biofor®). Nitrifying Biostyrs® are submerged fixed-bed biofilm reactors with a unitary section of 111 m² and a filter bed of 3 m high. This unit is operated to receive a nominal load of 0.7 kg NH₄⁺-N m⁻³ d⁻¹.

A lab-scale reactor with a working volume of 9.9 L (colonized Biostyrene® beads and interstitial volume) and a headspace of 1.4 L was operated in continuous down-flow counter-current mode for seven weeks (i.e. solution was down-flowing, while air was up-flowing; Fig. S1). Mass flow meters (F-201CV, Bronkhorst, France) sustained the inflow gas rate at 0.5 L min⁻¹. A peristaltic pump (R3425H12B, Sirem, France) pumped feeding solution from a feeding tank into the reactor at 0.2 L min⁻¹, in order to maintain a hydraulic retention time (HRT) of 27.8 ± 0.6 min. A water jacket monitored by a cryogenic regulator (WK

500, Lauda, Germany) controlled the reactor temperature. The feeding solution consisted of ammonium chloride ($NH_4Cl$) as substrate, monobasic potassium phosphate ($KH_2PO_4$) as phosphorus source for bacterial growth, and sodium hydrogen carbonate ($NaHCO_3$) as pH buffer and inorganic carbon source in 100 or 150 L of tap water (average $0.2 \pm 0.4$, $2.4 \pm 1.1$, and

$2.5 \pm 1.3$ mg N $L^{-1}$ of $NO_2^-$, $NO_3^-$ and sum of both $NO_x^-$ molecules, respectively).

The influence of environmental conditions on the ammonium oxidation rates and the $N_2O$ emissions from various combinations of oxygenation levels, temperatures and ammonium concentrations were tested in twenty-four experiments (Table 1). Note that two of them were used twice; as oxygenation and concentration tests. The oxygenation tests were carried out by mixing compressed air and pure nitrogen gas to reach 0 to 21 % $O_2$ in the gas mixture (Fig. S2a). The tests were performed at five

substrate concentrations and at a temperature between 19.2 and 20.6 °C. The temperature tests were carried out by cooling the feeding solution directly in the feeding tank (22.3 to 13.5 °C), with an inflow ammonium concentration close to the nominal load that received the nitrifying biomass; i.e. 20.3-21.1 mg $NH_4^+$-N $L^{-1}$. The ammonium concentration tests were run at an increase (6.2, 28.6 and 62.1 mg $NH_4^+$-N $L^1$) and a decrease (56.1, 42.9, 42.7 and 20.2 mg $NH_4^+$-N $L^{-1}$) in the $NH_4^+$ concentrations in the feeding solution, at temperatures ranging from 19.0 to 19.8 °C. Atmospheric oxygenation level (i.e. 21

% $O_2$ in the gas mixture) was imposed for both tests (Figs. S2b and c). This gas mixture using compressed air with 21 % $O_2$ was considered hereafter as optimal as compared to the oxygen-depleted atmosphere used during the oxygenation tests. Noticeably, the atmospheric oxygenation level is the condition that represents the most optimal conditions of oxygenation applied in nitrification BAF of domestic WRRF.

**Table 1. Detailed average conditions (± standard deviation) of oxygenation, temperature and concentration tests.**

| inflow [$NH_4^+$] | inflow gas rate | $O_2$ in gas mix | temperature |
|:---:|:---:|:---:|:---:|
| *mg N $L^{-1}$* | *L $min^{-1}$* | *%* | *°C* |
| *oxygenation tests* | | | |
| 25.1 ±0.5 | 0.4 | 0 | 19.2 ±0.1 |
| 23.8 ±0.6 | 0.53 | 4.2 | 19.9 ±0.1 |
| 25.1 ±0.5 | 0.53 | 4.2 | 19.2 ±0.1 |
| 37.3 ±0.6 | 0.5 | 4.2 | 20.5 ±0.1 |
| 23.8 ±0.6 | 0.51 | 10.5 | 20.2 ±0.1 |
| 25.1 ±0.5 | 0.51 | 10.5 | 19.2 ±0.1 |
| 37.3 ±0.6 | 0.5 | 10.5 | 20.6 ±0.1 |
| 23.8 ±0.6 | 0.5 | 16.8 | 20.1 ±0.1 |
| 25.1 ±0.5 | 0.5 | 16.8 | 19.3 ±0.1 |
| 37.3 ±0.6 | 0.5 | 16.8 | 20.6 ±0.1 |
| 20.2 ±0.5 | 0.5 | 21 | 19.5 ±0.1 |
| 25.1 ±0.5 | 0.57 | 21 | 19.6 ±0.5 |

| | | | |
|---|---|---|---|
| 28.6 ±0.5 | 0.5 | 21 | 19.6 ±0.1 |
| *temperature tests* | | | |
| 20.3 ±0.3 | 0.5 | 21 | 13.5 ±0.2 |
| 21.1 ±n.a. | 0.5 | 21 | 15.5 ±0.1 |
| 21.1 ±n.a. | 0.5 | 21 | 16.2 ±0.1 |
| 20.3 ±0.3 | 0.5 | 21 | 18.2 ±0.1 |
| 21.1 ±n.a. | 0.5 | 21 | 20.3 ±0.1 |
| 20.3 ±0.3 | 0.5 | 21 | 22.3 ±0.1 |
| *$NH_4^+$ concentration tests* | | | |
| 6.2 ±0.1 | 0.5 | 21 | 19.6 ±0.0 |
| 20.2 ±0.5 | 0.5 | 21 | 19.5 ±0.1 |
| 28.6 ±0.5 | 0.5 | 21 | 19.6 ±0.1 |
| 42.7 ±1.0 | 0.5 | 21 | 19.3 ±0.0 |
| 42.9 | 0.5 | 21 | 19.0 ±0.0 |
| 56.1 ±0.3 | 0.5 | 21 | 19.0 ±0.1 |
| 62.1 ±0.4 | 0.5 | 21 | 19.8 ±0.0 |

*Note that two experiments tested both oxygenation and ammonium concentration.*

## 2.2 Reactor monitoring, sampling and concentrations analysis

Dissolved oxygen, temperature (Visiferm DO Arc 120, Hamilton, Switzerland) and pH (H8481 HD, SI Analytics, France) were continuously measured at the top of the reactor and data were recorded at 10 second intervals. The $N_2O$ concentration was continuously analyzed by an infrared photometer (Rosemount™ X-STREAM X2GP, Emerson, Germany) in outflow reactor gas after drying through a condenser and a hydrophobic gas filter (0.2 µm). Minute averages are used for monitored data hereafter. Gas samples were taken for $N_2O$ isotopic signature determination by outlet gas pipe derivation into a sealed glass vial of 20 ml. The vial was first flushed with the sampling gas for > 45 sec prior to 1-5 min sampling. Gas samples were then stored in the dark at room temperature until analysis. Note that gas sampling was lacking for 5 of the 13 oxygenation tests.

The feeding solutions were characterized from 1 to 5 replicate samples collected in the feeding tank. For each tested condition, the outflow was characterized within 5 days from 1 to 14 replicate samples immediately filtered through a 0.2 µm syringe filter and stored at 4 °C. Outflow sampling started after at least one hydraulic retention time (28 ±1 min). Ammonium was analyzed using the Nessler colorimetric method, according to AFNOR NF T90-015 (DR 2800, Hach, Germany). Nitrite and nitrate were measured by ionic chromatography (IC25, Dionex, USA).

## 2.3 Stable isotope measurements

Atmospheric $N_2$ and Vienna Standard Mean Ocean Water (VSMOW) are the references used for the nitrogen and oxygen isotopes ratios, respectively, expressed in the conventional δ-notation, in per-mil (‰). Nitrogen and oxygen isotope ratios of nitrate and nitrite were determined separately following a modified protocol of McIlvin and Altabet (McIlvin and Altabet, 2005; Semaoune et al., 2012). Nitrogen isotope ratios of ammonium were determined following the protocol of Zhang et al. (2007). These methods consist in the conversion of the substrate (ammonium or nitrite or nitrate) into dissolved $N_2O$. The $\delta^{15}N$

and $\delta^{18}O$ for ammonium, nitrite, and nitrate were hence determined from a calibration curve created with a combination of nitrate or ammonium standards that have undergone the same chemical conversion as the samples (USGS-32, $\delta^{15}N\text{-}NO_3^- = 180$ ‰, $\delta^{18}O\text{-}NO_3^- = 25.7$ ‰; USGS-34, $\delta^{15}N\text{-}NO_3^- = -1.8$ ‰, $\delta^{18}O\text{-}NO_3^- = -27.9$ ‰ and USGS-35 $\delta^{15}N\text{-}NO_3^- = 2.7$ ‰, $\delta^{18}O\text{-}NO_3^- = 57.5$ ‰; or IAEA-N1, $\delta^{15}N\text{-}NH_4^+ = 0.4$ ‰, IAEA-305A, $\delta^{15}N\text{-}NH_4^+ = 39.8$ ‰, USGS-25, $\delta^{15}N\text{-}NH_4^+ = -30.4$ ‰). The quality of calibration was controlled with additional international standards (IAEA-NO-3, $\delta^{15}N\text{-}NO_3^- = 4.7$ ‰, $\delta^{18}O\text{-}NO_3^- =$

25.6 ‰; or IAEA-N2, $\delta^{15}N\text{-}NH_4^+ = 20.3$ ‰). Basically, an analytical sequence was comprised of triplicate standards for calibration and quality controls and duplicate samples. The average of the analytical replicates was then used for calibration, quality control and as result.

Since no international standards were available for $N_2O$ isotopes, these were determined the same day as nitrate or ammonium standard analysis insuring correct functioning of the method and analysis. In addition to this, the internal $N_2O$ standards were

previously calibrated by exchange with the laboratory of Naohiro Yoshida and Sakae Toyoda at the Tokyo Institute of Technology. All isotope measurements were determined using an isotope ratio mass spectrometer (IRMS, DeltaVplus; Thermo Scientific) in continuous-flow with a purge and trap system coupled with a Finnigan GasBench II system (Thermo Scientific). The precision was 0.8 ‰, 1.5 ‰ and 2.5 ‰ for $\delta^{15}N$, $\delta^{18}O$, and $^{15}N\text{-}SP$, respectively.

## 2.4 Data processing and statistics

The effects of environmental conditions on nitrification were assessed from 4 indices. The ammonium oxidation rate (AOR) was estimated in each experiment for time $\geq 1$ HRT from the difference between influent and effluent $NH_4^+$ concentrations multiplied by the liquid flow rate (kg $NH_4^+\text{-}N$ d$^{-1}$). The nitrification efficiency was defined as the ratio between AOR and influent ammonium load. The $N_2O$ emission rate ($N_2O\text{-}ER$) was calculated by multiplying the measured $N_2O$ concentration by the gas flow rate (mg $N_2O\text{-}N$ min$^{-1}$). The $N_2O$ emission factor ($N_2O\text{-}EF$) was defined as the ratio between $N_2O\text{-}ER$ and AOR

(% of oxidized $NH_4^+\text{-}N$). The measurements related to liquid or gas samples were averaged by experiment; i.e. average of data obtained from the samples collected after one hydraulic retention time.

Statistical analysis were performed using the R software (R Core Team, 2014). The value of 0.05 was used as significance level for spearman correlations (*cor.test* function) and linear regressions (*lm* function). *Adjusted r²* was provided as *r²* for the latter.

## 2.5 Estimation of ranges of nitrogen isotope ratio in biologically produced $N_2O$

As shown in Fig. 1, the pairwise relationships between $\delta^{15}N$, $\delta^{18}O$ and $^{15}N$-SP assist the determination of the producing and consuming pathways at play. The N atoms that compose the $N_2O$ molecule originate from $NH_4^+$ molecules when produced by hydroxylamine oxidation, while originating from the N atoms of $NO_3^-$ or $NO_2^-$ molecules when produced by nitrite reduction ($NO_x^-$ molecules). However, the nitrogen isotope ratio of $N_2O$ does not equal those of its substrates as it depends on isotope effects associated to each reaction step of $N_2O$ producing process. The isotope effect of reaction step can be determined from the isotope composition of substrates or products. Although being performed on a few tests here, the obtained value can only be applied to a limited number of environmental conditions. The use of estimates from the literature seems therefore suitable. Several equations enable to approximate the isotope effect and its effect on the isotope ratios of substrate and product pools involved in a reaction. These equations vary according to the assumptions made on the system boundaries (Denk et al., 2017). The nitrifying reactor used in this study can be described as an open-system continuously supplied by an infinite substrate pool with constant isotopic composition ($NH_4^+{}_{,in}$). A small amount of the infinite substrate pool is transformed into a product pool ($NO_x^-{}_{,p}$) or a residual substrate pool ($NH_4^+{}_{,res}$) when flowing through the system. The equations describing the input, output and processes considered here are presented in Fig. 2 after Fry (2006). Note that the definitions of f and $\Delta$ are inverse to the cited literature and that $\Delta_1$ and $\Delta_4$ are null because no fractionation alter the residual substrate exiting the reaction (Fry, 2006).

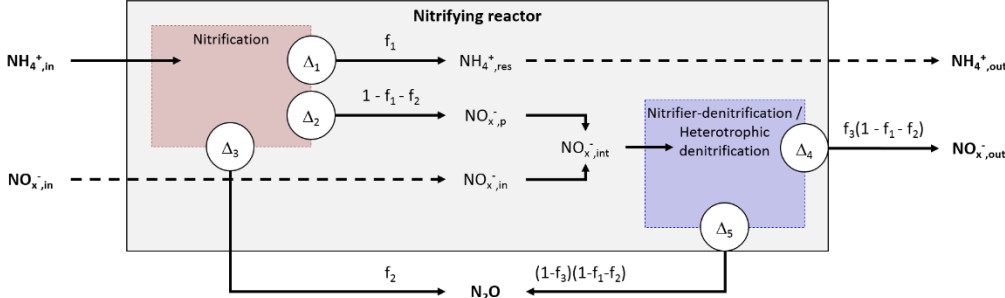

**Figure 2. Diagram and equations of the nitrifying reactor after Fry (2006). It is considered as a sequence of two reactor boxes. (i) The nitrification of inflow ammonium ($NH_4^+$,in) to a pool of nitrite and nitrate ($NO_x^-$,p), residual ammonium ($NH_4^+$,res), and nitrous oxide ($N_2O$) through the hydroxylamine oxidation pathway. (ii) The subsequent reduction of intermediate $NO_x^-$,int; mixing of inflow $NO_x^-$,in and formed $NO_x^-$,p to nitrous oxide ($N_2O$) through the nitrite reduction pathway, and residual $NO_x^-$ that exits the reactor ($NO_x^-$,out). Note that residual substrates and formed products exit the reactor without further isotope fractionation ($\Delta_1$ and $\Delta_4$ are null). See text for details.**

The balance between input and output of each reactional step allows to propose equations for calculation of the nitrogen isotope ratio of compounds in the inflow and outflow of the system (Denk et al., 2017; Fry, 2006). These equations can be simplified under the assumption that limited amount of N compounds are transformed into $N_2O$; i.e. $f_2$ close to 0 and $f_3$ close to 1. Therefore, the N isotope ratios of the residual substrate pool can be approximated from Eq. (1).

$$\delta^{15}\text{N-NH}_4^+{}_{,\text{res}} \approx \delta^{15}\text{N-NH}_4^+{}_{,\text{in}} - \Delta_2(1 - f_1) , \tag{1}$$

Where $f_1$ is the remaining substrate fraction leaving the reactor (i.e. remaining fraction of ammonium), ranging from 0 to 1 (0-100 in %), and $\Delta_2$ is the N isotope enrichment factor associated with ammonium oxidation. In their review, Denk et al. (2017) reported a mean value of -29.6 ± 4.9 ‰ for $\Delta_2$. Therefore, $\delta^{15}$N is higher for residual than the initial substrate pool ($\delta^{15}$N-$NH_4^+$,in $< \delta^{15}$N-$NH_4^+$,res). Consequently, the pool of product is depleted in heavier isotope (i.e. nitrite and nitrate hereafter defined as $NO_x^-$ pool; $\delta^{15}$N-$NO_x^-$,in $> \delta^{15}$N-$NO_x^-$,int). It is estimated from Eqs. (2-4):

$$\delta^{15}\text{N-NO}_x^-{}_{,\text{p}} \approx \delta^{15}\text{N-NH}_4^+{}_{,\text{in}} + \Delta_2 f_1 , \tag{2}$$

Where $\delta^{15}$N-$NO_x^-$,p is the nitrogen isotope ratio of the product pool produced by nitrification. The nitrogen isotope ratio of the overall intermediate $NO_x^-$ exiting this process results of mixing between initial and produced $NO_x^-$ pools ($\delta^{15}$N-$NO_x^-$,int) and can be estimated from Eqs. (3) and (4):

$$\delta^{15}\text{N-NO}_x^-{}_{,\text{in}} = \frac{(\delta^{15}\text{N-NO}_2^-{}_{,\text{in}} \times [\text{NO}_2^-]_{\text{in}} + \delta^{15}\text{N-NO}_3^-{}_{,\text{in}} \times [\text{NO}_3^-]_{\text{in}})}{([\text{NO}_3^-]_{\text{in}} + [\text{NO}_2^-]_{\text{in}})}, \tag{3}$$

$$\delta^{15}\text{N-NO}_x^-{}_{,\text{int}} \approx \frac{(\delta^{15}\text{N-NO}_x^-{}_{,\text{in}} \times ([\text{NO}_3^-]_{\text{in}} + [\text{NO}_2^-]_{\text{in}}) + \delta^{15}\text{N-NO}_x^-{}_{,p} \times (1-f_1) \times [\text{NH}_4^+]_{\text{in}})}{([\text{NO}_3^-]_{\text{in}} + [\text{NO}_2^-]_{\text{in}} + (1-f_1) \times [\text{NH}_4^+]_{\text{in}})}, \tag{4}$$

Note that $\delta^{15}\text{N-NO}_x^-{}_{,\text{out}}$ equals $\delta^{15}\text{N-NO}_x^-{}_{,\text{int}}$ when $f_3$ is close to 1 which means that nitrifier-denitrification and heterotrophic denitrification are negligible. Finally, two options must be considered to approximate the nitrogen isotope ratio of $\text{N}_2\text{O}$ that exits the reactor. On the one hand, $\delta^{15}\text{N-N}_2\text{O}$ can be estimated from Eq. (5), when the hydroxylamine oxidation is the producing process of $\text{N}_2\text{O}$:

$$\delta^{15}\text{N-N}_2\text{O} \approx \delta^{15}\text{N-NH}_4^+{}_{,\text{res}} - \Delta_2(1 - f_1) + \Delta_3, \tag{5}$$

In addition to the influence of the nitrogen isotope composition of the substrate, the $\delta^{15}\text{N-N}_2\text{O}$ depends therefore on the difference between the isotope effects related to the oxidation of $\text{NH}_4^+$ to $\text{NO}_2^-$ and the oxidation of $\text{NH}_2\text{OH}$ to $\text{N}_2\text{O}$ for complete nitrification ($f_1 = 0$), while depending only on the latter for limited nitrification ($f_1 = 1$). On the other hand, $\delta^{15}\text{N-N}_2\text{O}$ can be estimated from Eq. (6), when the nitrite reduction is the producing process of $\text{N}_2\text{O}$:

$$\delta^{15}\text{N-N}_2\text{O} \approx \delta^{15}\text{N-NO}_x^-{}_{,\text{int}}(1 - f_1)^{-1} + \Delta_5, \tag{6}$$

In addition to the influence of the nitrogen isotope composition of the substrate, when negligible amounts of $\text{N}_2\text{O}$ are produced by nitrite reduction during nitrifier-denitrification or heterotrophic denitrification its nitrogen isotope ratio depends on isotope effect related to this process ($\Delta_5$).

## 3 Results and Discussion

Changes in pH, ammonium, nitrite and nitrate concentrations confirmed nitrifying activity in the reactor system (Table S1, Fig. S3). During the ammonium concentration tests, decreases in ammonium concentrations ($[\text{NH}_4^+]$), increases in nitrite and nitrate concentrations ($[\text{NO}_2^-]$ and $[\text{NO}_3^-]$, respectively) were observed, while pH remaining below 8 prevented any relevant loss of ammonium by volatilization. For example, $[\text{NH}_4^+]$ decreased from 6.2 to 1.1, from 28.6 to 17 and from 62.1 to 49.1 mg N L$^{-1}$ by flowing through the nitrifying biomass. At the same time, $[\text{NO}_2^-]$ and $[\text{NO}_3^-]$ increased from 0 to 0.2-0.3 mg N L$^{-1}$ and from 1.4-1.8 to 5-10 mg N L$^{-1}$, respectively. Over the range of tested conditions, the ratio between ammonium oxidation rate and influent ammonium load ranged from 10 to 82 %, never exceeding 40 % for suboptimal nitrifying conditions imposed during oxygenation and temperature tests (i.e. oxygenation levels < 21 % $\text{O}_2$, and temperatures < 20 °C). The ammonium concentration, oxygenation level and temperature affected the ammonium oxidation rates, as well $\text{N}_2\text{O}$ emission rates and factors.

### 3.1 Isotope composition ranges of $\text{N}_2\text{O}$ produced by hydroxylamine oxidation and nitrite reduction

Ranges of $\delta^{15}\text{N}$ for $\text{N}_2\text{O}$ produced by different processes were hypothesized from Eqs. (1-5) for pairwise relationships with reviewed data of $\delta^{18}\text{O}$ and $^{15}\text{N-SP}$. To this aim, measurements of isotope ratios of the different nitrogen species were required. The $\delta^{15}\text{N}$ of inflow ammonium, nitrite and nitrate were $-3 \pm 0.1$ ‰ (n = 3), $-15 \pm 0.1$ ‰ (n = 2), $6.9 \pm 0.3$ ‰ (n = 3), respectively

during ammonium concentration experiments (Fig. S3 and Table S2). The $\delta^{15}N$ of residual $NH_4^+$ and intermediate $NO_x^-$ were

estimated from Eqs. (1-4) with $f_1 = 0.1$ or $0.9$ (Figs. S2d-f), $\Delta_2 = -30$ ‰, the highest $[NH_4^+]_{in}$ (62.1 mg N $L^{-1}$) and the lowest $[NO_x^-]_{in}$ (1.4 mg N $L^{-1}$). They ranged from -3 to 27 ‰ and from -32 to 7 ‰, respectively, which encompasses a few isotope compositions measured in the outflow during ammonium concentration tests (Fig. S3 and Table S2).

Prior to pairwise comparisons with $\delta^{18}O$ and $^{15}N$-SP, ranges of $\delta^{15}N$ values for $N_2O$ produced by the hydroxylamine oxidation and nitrite reduction pathways were estimated from Eq. (5). The net isotope effect of $N_2O$ production by ammonium oxidation

via hydroxylamine can be estimated by combining the isotope effects of ammonium oxidation and hydroxylamine oxidation to $N_2O$. The net isotope effect associated to the ammonium oxidation to nitrite ranges from -38.2 to -14.2 ‰ (Casciotti et al., 2003) and can approximate the nitrogen isotope ratio of hydroxylamine transitory produced. The isotope effect related to hydroxylamine oxidation to $N_2O$ ranging from -26.0 to 5.7 ‰ from data in Sutka et al. (2003, 2004, 2006), the net isotope effect of $N_2O$ production by ammonium oxidation via hydroxylamine ($\Delta_3$) can range from -64.2 ‰ (-26.0 + (-38.2)) to -8.5 ‰

(5.7 + (-14.2)). Considering the range of nitrogen isotope ratio of residual ammonium, this method provided a broad range of $\delta^{15}N$ values, from -65 ‰ ($\delta^{15}N$-$NH_4^+$,res = -3 ‰, $\Delta_2 = -30$ ‰, $f_1 = 0.9$ and $\Delta_3 = -64.2$ ‰) to 46 ‰ ($\delta^{15}N$-$NH_4^+$,res = 27 ‰, $\Delta_2 = -30$ ‰, $f_1 = 0.1$ and $\Delta_3 = -8.5$ ‰), for $N_2O$ produced from ammonium by hydroxylamine oxidation, according to Eq. (5). These values encompassed the values proposed by others (-46.5 and -32.9 ‰; Sutka et al., 2006; Yamazaki et al., 2014).

A higher range of net nitrogen isotope effect for nitrite reduction than hydroxylamine oxidation pathway was estimated for

$N_2O$ production (Figs. 3a and b). Prior to being reduced to $N_2O$ through the nitrite reduction pathway, $NO_x^-$ was mainly derived from ammonium oxidation in the nitrifying system (Eqs. (1-4)); the resulting intermediate $\delta^{15}N$-$NO_x^-$ ranging from -32 to 7 ‰. In addition to this, the net isotope effects related to the $N_2O$ production through nitrite reduction performed by nitrifiers or heterotrophic denitrifiers ($\Delta_5$) ranges from -52.8 to -6 ‰ (Lewicka-Szczebak et al., 2014; Sutka et al., 2008). Consequently, the $\delta^{15}N$ of $N_2O$ produced by nitrite reduction ranged from -89 ‰ ($\delta^{15}N$-$NO_x^-$,int = -32 ‰, $f_1 = 0.1$ and $\Delta_5 = -52.8$ ‰) to 64 ‰

($\delta^{15}N$-$NO_x^-$,int = 7 ‰, $f_1 = 0.9$ and $\Delta_5 = -6$ ‰) , according to Eq. (6). This is consistent with previous findings reporting $\delta^{15}N$-$N_2O$ between -112 and -48 ‰ for nitrifier-denitrifying systems (Mandernack et al., 2009; Pérez et al., 2006; Yamazaki et al., 2014; Yoshida, 1988). However, a similar range of nitrite-derived $\delta^{15}N$-$N_2O$ is suggested for nitrifiers and heterotrophic denitrifiers, because ammonium oxidation influences both processes in the system used in this study where there is low initial amount of $NO_2^-$ and $NO_3^-$.

Pairwise comparisons of $\delta^{15}N$, $\delta^{18}O$ and $^{15}N$-SP estimates of the different experiments are presented in Fig. 3. These comparisons provided ranges of plausible isotope compositions for $N_2O$ produced by nitrifying or heterotrophic denitrifying bacteria through the hydroxylamine oxidation and nitrite reduction pathways (red and blue boxes, respectively). The measured $N_2O$ isotope compositions were compared to these estimates to identify the $N_2O$ producing and consuming pathways likely at play in oxygenation, temperature, and ammonium concentration tests.

This approach suggests that the nitrite reduction pathway was the main contributor to the $N_2O$ emissions. Heterotrophic denitrification likely influenced the $N_2O$ emissions, as shown by oxygen isotope ratios higher than 35 ‰ (Snider et al., 2013)

(Figs. 3a and c). However, this conclusion depends highly on $\delta^{18}O$-$N_2O$ ranges. Furthermore, the application of atmospheric oxygen $\delta^{18}O$ (23.5 ‰; Kroopnick and Craig, 1972) to estimate the oxygen isotope ratio of $N_2O$ produced by hydroxylamine oxidation remains uncertain since respiratory activity and air stripping might drive isotopic fractionations and increase the

$\delta^{18}O$ of residual dissolved oxygen (Nakayama et al., 2007). To date, the oxygen isotope fractionation related to air stripping is lacking. Note that this estimate relies on the assumption that there is no accumulation of $NH_2OH$ and that its oxidation to $N_2O$ occurs before or independently of its oxidation to $NO_2^-$.

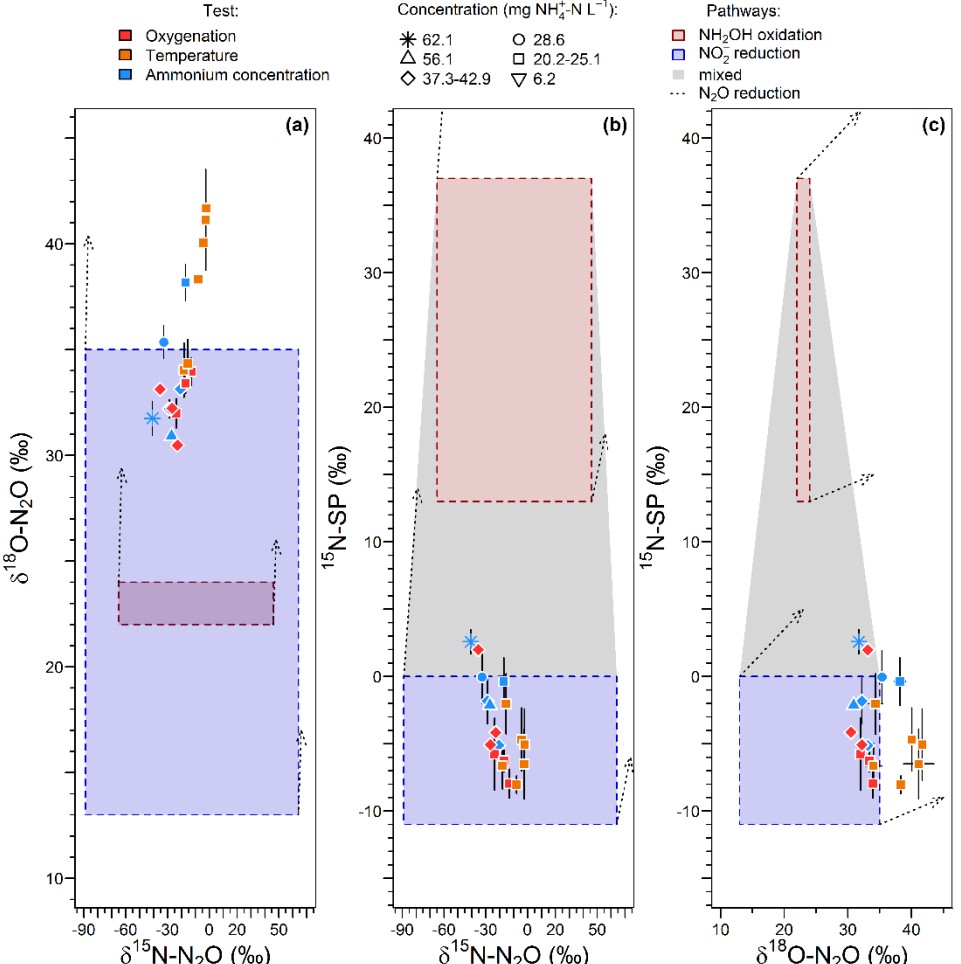

**Figure 2: Interpretation maps of the isotope signature of N₂O. Schematic maps of (a) $\delta^{15}N$-$\delta^{18}O$, (b) $\delta^{15}N$-$^{15}N$-SP, and (c) $\delta^{18}O$-$^{15}N$-SP. The red and blue squares show the range of the data for N₂O produced by 'hydroxylamine oxidation' and 'nitrite reduction', respectively. The shaded area represents mixing of N₂O produced by these pathways. The N₂O reduction increases $\delta^{15}N$, $\delta^{18}O$, and $^{15}N$-SP with slopes characterizing the pairwise relationships.**

## 3.2 The effect of oxygen limitation on the N$_2$O producing pathways

Ammonium concentrations decreased from 20.2-37.3 to 11.4-31.1 mg N L$^{-1}$; with 45 to 89 % of the inflow ammonium remaining in the outflow during the oxygenation tests (Fig. S2d). When measured, the cumulated concentrations of NO$_2^-$ and NO$_3^-$ ([NO$_x^-$]) increased from 2.4-4.1 to 4.7-11 mg N L$^{-1}$ between inflow and outflow and were composed by at least 74 and 82 % of NO$_3^-$, respectively. The mass balance between N compounds that enter and exit the reactor evidenced a default of up to 5 mg N and impacted each test. No significant amount of NO was detected during any tests (data not shown) whereas neither NH$_2$OH, N$_2$, nor N mineralization/assimilation in the biofilm were quantified. The accumulation of such amounts of NH$_2$OH is unlikely. Heterotrophic denitrification, i.e. the reduction of NO$_x^-$ and more particularly of N$_2$O to N$_2$, may explain the incomplete N mass balance. However, the measurement of small N$_2$ variations in the gas mixture that exiting the reactor and comprising at least 79 % N$_2$ was not measured.

The oxygenation level had contrasting effects on ammonium oxidation rates, and N$_2$O emission rates and factors (Figs. 4a-c). Between an oxygenation of 0 to 10.5 % O$_2$ in the gas mixture, no clear trend in ammonium oxidation rates was observed although being rather low (1.1 ± 0.5 mg NH$_4^+$-N min$^{-1}$). In the same oxygenation levels interval, the N$_2$O emission rate increased for two of three inflow [NH$_4^+$] tested. It increased from 0.35 x 10$^{-3}$ to 0.73 x 10$^{-3}$ mg N min$^{-1}$ between 0 and 10.5 % O$_2$ at 25.3 mg NH$_4^+$-N L$^{-1}$, and from 1.34 x 10$^{-3}$ to 1.4 x 10$^{-3}$ mg N min$^{-1}$ between 4.2 and 10.5 % O$_2$ at 23.8 mg NH$_4^+$-N L$^{-1}$; while it decreased from 2.86 x 10$^{-3}$ to 2.04 x 10$^{-3}$ mg N min$^{-1}$ between 4.2 and 10.5 % O$_2$ at 37.3 mg NH$_4^+$-N L$^{-1}$. Finally, the N$_2$O emission factor globally increased from 0.05 to 0.16 % in the 0-10.5 % O$_2$ interval. At oxygenation levels from 10.5 to 21 % O$_2$, the ammonium oxidation rates increased from 0.9 ± 0.2 to 2.1 ± 0.4 mg N min$^{-1}$, with N$_2$O emission rates remaining stable at 1.2 x 10$^{-3}$ ± 0.6 x 10$^{-3}$ mg N min$^{-1}$ and the emission factors decreasing from 0.15 ± 0.03 to 0.06 ± 0.03 %.

The $^{15}$N-SP varied between -9 to 2 ‰ over the range of imposed oxygenation levels, with a marked increase when oxygenation increased from 16.8 to 21 % O$_2$ (Fig. 4d). A similar marked change in nitrogen and oxygen isotope ratios of N$_2$O (decrease and increase, respectively) was observed when oxygenation increased from 16.8 to 21 % O$_2$ (Figs. 4e and f). Note that to observe the latter variations the effect of ammonium concentration was not included. One way to do so is to compare the isotope composition average at 21% O$_2$ with the isotope composition measured for 23.8 NH$_4^+$-N L$^{-1}$ at 16.8 % O$_2$. The $^{15}$N-SP values were close to the range of -11 to 0 ‰ reported for N$_2$O produced by nitrifying or denitrifying bacteria through nitrifier-denitrification and heterotrophic denitrification (Toyoda et al., 2017; Yamazaki et al., 2014). Additional suggestions can be made from the $^{15}$N-SP dynamics between and variations within the oxygenation levels. If an increase in the hydroxylamine oxidation contribution to the N$_2$O emission might explain the higher $^{15}$N-SP observed at 21 % O$_2$ as compared to lower oxygenation levels, an additional mechanism can explain the variations observed for the experiments with oxygen-depleted atmosphere. The $^{15}$N-SP dynamics suggest a higher amount of N$_2$O was reduced to N$_2$ at 4.2 than 16.8 % O$_2$. The reduction of N$_2$O to N$_2$ can increase the $^{15}$N-SP of residual N$_2$O (Mothet et al., 2013). In heterotrophic denitrifying bacteria however, the nitrous oxide reductase involved in this reaction is highly sensitive to inhibition by oxygen (Betlach and Tiedje, 1981; Otte et al., 1996). This might explain the decrease in $^{15}$N-SP from -3.8 ± 4.4 ‰ to -7.2 ± 1.7 ‰ when O$_2$ increased from 4.2 to 16.8

%. This is also consistent with a possible onset of anoxic microsites within the reactor biomass more likely at 4.2 than 16.8 % $O_2$. The dissolved oxygen (DO) concentration never decreased below 1.5 mg $O_2$ $L^{-1}$ in the bulk solution at the top of the reactor (Fig. S2). However, DO decreased from the bulk reactor solution toward the deeper layers of biofilm due to the activity of ammonium oxidizers (Sabba et al., 2018). This is further exacerbated by heterogeneous and varying distribution of air

circulation within the static bed. Therefore, oxygen depletion can be assumed within the biofilm. Finally, the $N_2O$ reduction to $N_2$ likely explains the overall decrease in $N_2O$ emission between 16.8 and 0 % $O_2$ (Fig. 4b).

In general the $N_2O$ reduction to $N_2$ is accompanied by an increase in nitrogen and oxygen isotope ratios of $N_2O$ (Ostrom et al., 2007; Vieten et al., 2007). However, our results show a decrease in $\delta^{15}N$-$N_2O$ and the $\delta^{18}O$-$N_2O$ remained stable between 30.5 and 34.7 ‰ when the $N_2O$ reduction is thought to increasingly constraint the $N_2O$ isotopocules with decreasing $O_2$ from 16.8

to 4.2 % (Figs. 4e and f). The independence of samples taken during the oxygenation test can explain this. The $N_2O$ sampled at 4.2 % $O_2$ is not a residual fraction of $N_2O$ produced at 16.8 % $O_2$ that would have undergone a partial reduction. The oxygenation level can alter the isotope fractionation factors through the control of reaction rates, as evidenced for the reduction of $N_2O$ to $N_2$ by Vieten et al. (2007). These authors reported lower reaction rates and increased isotope fractionation factors with increasing oxygenation levels. In our case, a similar phenomenon might have influenced both oxidative and reductive

processes leading to the production of $N_2O$ and occurring before its ultimate reduction to $N_2$. However, knowledge regarding the controls such as the oxygenation level have on the net isotope effect related to a sequence of non-exclusive oxidative and reductive processes is still lacking and require further investigations. Additionally, with $\delta^{18}O$ below 35 ‰ for all but one experiment the oxygenation tests did not provide evidence for heterotrophic denitrifier contribution to $N_2O$ emissions, likely due to oxygen-exchange with water (Snider et al., 2015, 2012, 2013).

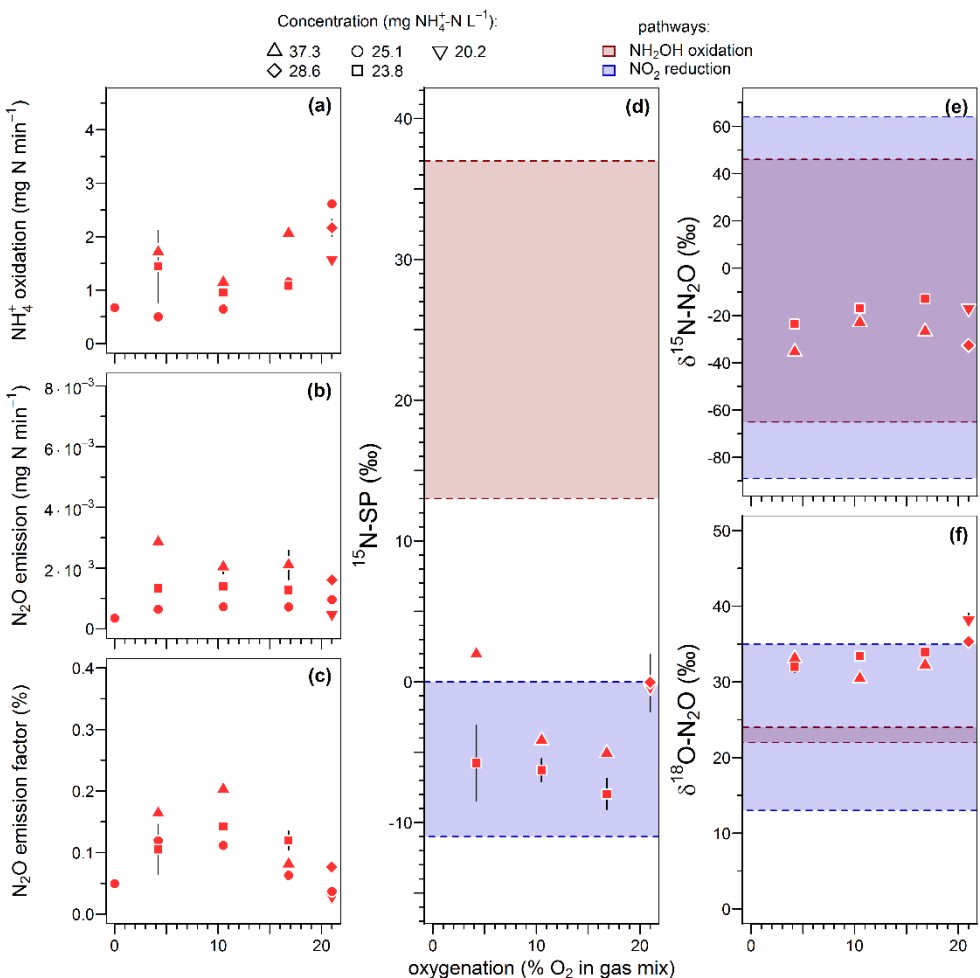

**Figure 3: Effect of oxygenation level on (a) the ammonium oxidation rate, (b) the nitrous oxide emission rate, (c) the N₂O emission factor, and (d) the nitrogen isotopomer site preference, (e) the nitrogen isotope ratio, and (f) the oxygen isotope ratio of N₂O. Average and standard deviation (error bars) are calculated for the steady-state conditions. Note that gas sampling for isotope analysis was lacking for 5 of the 13 oxygenation tests.**

**3.3 Difference in temperature dependency of hydroxylamine and ammonium oxidizers as driver of hydroxylamine oxidation contribution to N₂O emissions**

Ammonium concentrations decreased from 6.2-62.1 to 0.9-54.1 mg N L$^{-1}$; from 18 to 79 % of the inflow ammonium remaining in the outflow during the temperature and ammonium concentration tests (Figs. S2e and f). This remaining fraction was positively correlated to ammonium concentrations ($r = 0.96$), and negatively correlated to temperature within a lower range of 325 values (61-67 %; $r = -0.94$). In the ammonium tests, the cumulated concentrations of NO₂$^-$ and NO₃$^-$ ([NO$_x^-$]) increased from 1.4-6.1 to 5.1-19.6 mg N L$^{-1}$ between inflow and outflow and were composed by at least 74 and 91 % of NO₃$^-$, respectively. Noticeably, the nitrite concentrations in the outflow linearly increased with temperature ($r^2 = 0.95$; Fig. S2h).

An increase in temperature and inflow ammonium concentrations both positively influenced the rates of $NH_4^+$ oxidation and $N_2O$ emissions and the emission factor (Fig. 5). The $NH_4^+$ oxidation rate linearly increased from 1.3 to 1.5 mg $NH_4^+$-N min$^{-1}$

with temperature ($r = 0.89$; Fig. 5a) and increased from 0.97 to 3.49 mg $NH_4^+$-N min$^{-1}$ with a tenfold increase in the inflow ammonium concentration ($r = 0.82$; Fig. 5b). These positive correlations are well known in the temperature range investigated here and are likely due to enhanced enzymatic activity and Michaelis-Menten kinetics, respectively (Groeneweg et al., 1994; Kim et al., 2008; Raimonet et al., 2017). Similarly, the $N_2O$ emission rates increased from 80.4 x 10$^{-6}$ to 2.5 x 10$^{-3}$ mg $N_2O$-N min$^{-1}$, and from 83.6 x 10$^{-6}$ to 6.2 x 10$^{-3}$ mg $N_2O$-N min$^{-1}$ upon changes in temperature and ammonium concentrations,

respectively. These results are in agreement with positive correlations between $N_2O$ emissions with temperature and ammonium concentration observed from modelling and experimental studies on partial nitrification and activated sludge systems (Guo and Vanrolleghem, 2014; Law et al., 2012a; Reino et al., 2017). Altogether this confirms a correlation between the $N_2O$ emission rates and the ammonium oxidation rates. Interestingly, the increase in $N_2O$ emission factor indicates a stronger effect of temperature and ammonium concentration on the $N_2O$ emission rate than on $NH_4^+$ oxidation. The $N_2O$

emission factors increased from 0.07 to 0.16 %, and from 0.01 to 0.29 % with temperature and inflow ammonium concentration, respectively ($r > 0.94$; Figs. 5e and f). Both experiments suggest that the increase in $N_2O$ emissions results from the increasing production of $N_2O$ by hydroxylamine oxidation or nitrite reduction in combination with a slow rate of or absence of $N_2O$ reduction to $N_2$. Furthermore, no nitrite accumulation was observed with increasing ammonium oxidation rate (Fig. S2i). Therefore, if $N_2O$ emission results mainly from the nitrite reduction pathway, this suggests that the nitrite reduction

pathway is more responsive to the increasing ammonium oxidation rate than the nitrite oxidation pathway; the latter remaining the main pathway of nitrite consumption.

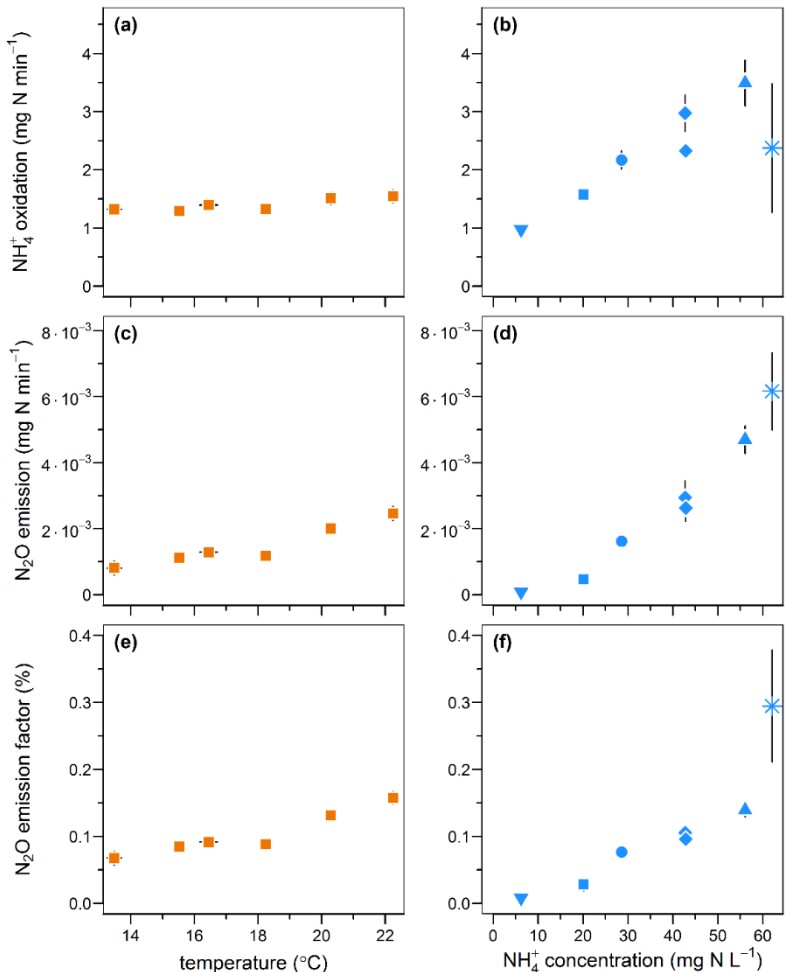

**Figure 4: Effect of temperature and inflow ammonium concentration on (a, b) the ammonium oxidation rate, (c, d) the nitrous oxide emission rate, and (e, f) the N₂O emission factor.**

The range of nitrogen isotopomer site preference observed during the temperature and concentration tests (from -8 to 2.6 ‰) was similar to those measured during the oxygenation tests; confirming the high contribution of the nitrite reduction pathway to N$_2$O emissions (Fig. 6a). This is consistent with previous findings based on the [15]N-SP of N$_2$O emitted from aerobic activated sludge (Toyoda et al., 2011; Tumendelger et al., 2016; Wunderlin et al., 2013), although authors reported [15]N-SP as high as 10 ‰. This can suggest a higher oxygen limitation being favourable to the contribution of the nitrite reduction to N$_2$O production in the nitrifying reactor studied here. The hydroxylamine oxidation can even be the main N$_2$O producing pathway, as evidenced by Tumendelger et al. (2014) in some aerated tank.

Furthermore, the [15]N-SP increased with temperature between 13.5 and 19.8 °C. Our data suggest that temperature was the main control on the change in N$_2$O producing pathways within this temperature range (Fig. 6a). This could explain higher SP obtained with 28.6 than 42.8 mg N L$^{-1}$ inflow ammonium concentration. The temperature control seems to mitigate here the

effect that ammonium concentration can have on the $N_2O$ producing pathways evidenced elsewhere. Wunderlin et al. (2012, 2013) observed an increase in $^{15}N$-SP from -1.2 to 1.1 ‰ when inflow [$NH_4^+$] increased from 9 to 15 mg N $L^{-1}$. They also observed 3-6 ‰ decreases in $^{15}N$-SP over the course of ammonium oxidation experiments and suggested that $NH_2OH$ oxidation

contribution to $N_2O$ production increased when conditions of $NH_4^+$ excess, low $NO_2^-$ concentrations and high nitrogen oxidation rate occur simultaneously. Our findings are consistent with the observation of Groeneweg et al. (1994) showing that temperature rather than ammonium concentration influenced the ammonium oxidation rate.

The $^{15}N$-SP increased from -6.5 to 2.6 ‰ with increasing temperature from 13.5 to 19.8 °C (Fig. 6a). This $^{15}N$-SP increase may either result from an increase in the $N_2O$ production by the hydroxylamine oxidation pathway or the $N_2O$ reduction to $N_2$.

Since an optimal oxygenation level was imposed and increased emissions were observed, the increasing $^{15}N$-SP is more likely due to $N_2O$ production by the hydroxylamine oxidation pathway. Reino et al. (2017) also observed an increase of $N_2O$ emissions for temperature above 15 °C in a granular sludge airlift reactor performing partial nitritation. The authors suggested two hypothesis to explain their results: (i) the difference in the kinetic dependency with temperature of enzymes involved in ammonium and hydroxylamine oxidation; (ii) the temperature dependency of the acid-base equilibrium ammonium-ammonia.

The changes in $^{15}N$-SP observed here are consistent with the former hypotheses. Hydroxylamine oxidation likely becomes the limiting step at a temperature above 15 °C, while being faster than ammonium oxidation at lower temperature (Fig. 7). At temperature above 15 °C, hydroxylamine therefore accumulates and leads to a higher contribution of the hydroxylamine oxidation pathway to $N_2O$ emissions. It would thus be interesting to determine the temperature dependency of the hydroxylamine oxidase.

The change in nitrous oxide producing and consuming pathways had contrasted effects on the nitrogen and oxygen isotope ratios of nitrous oxide (Figs. 6b and c). The $\delta^{15}N$-$N_2O$ decreased from -2.5 to -40.9 ‰ with an increasing contribution of hydroxylamine oxidation to the $N_2O$ emissions; i.e. when temperature increased from 13.5 to 19.8 °C. This is in contrast with the expected net lower isotope effect for $N_2O$ produced by hydroxylamine oxidation than nitrite reduction and points out that further investigations are needed (Snider et al., 2015; Yamazaki et al., 2014). The changes in $\delta^{18}O$-$N_2O$ were less

straightforward; likely influenced by changes in the reaction rates in addition to changes in the contribution of $N_2O$ producing pathways. The values decreased from 41.1 to 34.3 ‰ with an increasing contribution of hydroxylamine oxidation to the $N_2O$ emissions; when temperature increased from 13.5 to 18.2 °C. It decreased linearly from 38.2 to 31.8 ‰ with increasing reaction rate, when inflow ammonium concentration increased from 20.2 to 62.1 mg $NH_4^+$-N $L^{-1}$ ($r^2 = 0.83$).

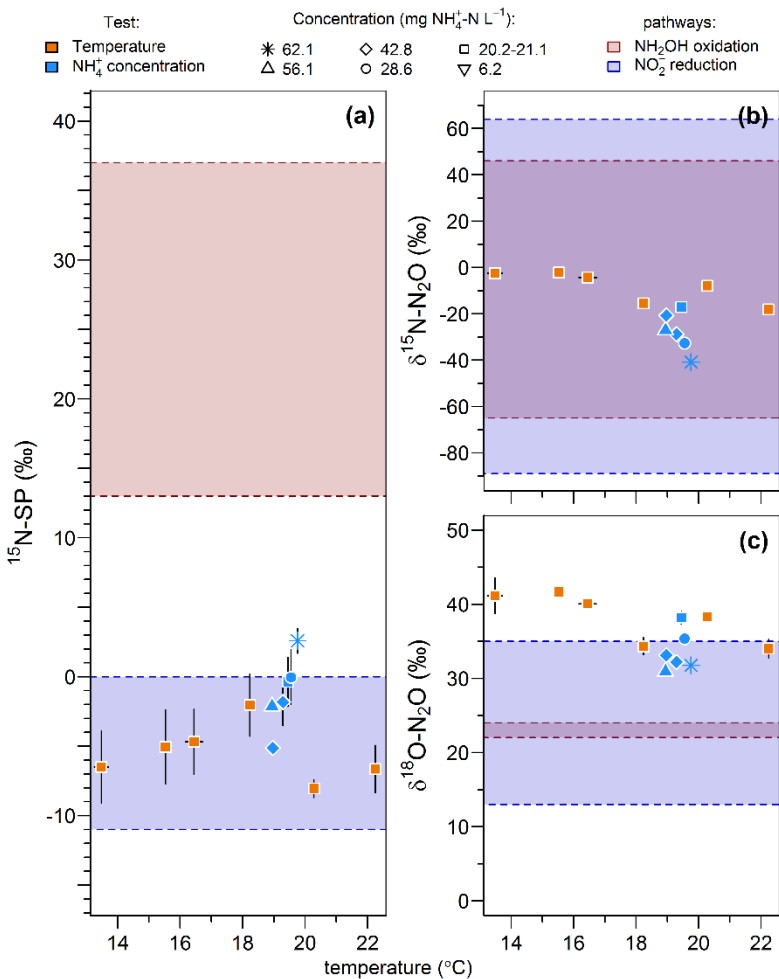

**Figure 5: Effect of temperature (orange symbols) and inflow ammonium concentration (blue symbols) on (a) the nitrogen isotopomer site preference, (b) the nitrogen isotope ratio, and (c) the oxygen isotope ratio of N₂O. Average and standard deviation (error bars) are calculated for the samples taken after one hydraulic retention time. Note that the isotopic measurements of gas samples taken at inflow ammonium concentration of 42.7 and 42.9 mg N L⁻¹ were both recorded as 42.8 mg N L⁻¹ in the legend.**

### 3.4 Difference in oxidation and reduction rates of nitrite as driver of nitrite reduction contribution to N₂O emissions

The oxygenation, temperature and ammonium concentration tests revealed a strong control of nitrite oxidizing activity and the contribution of the nitrite reduction pathway to $N_2O$ production. No relationship was observed between $NO_2^-$ concentrations and oxygenation (Fig. S2g). In addition to this, higher $^{15}N$-SP at 21 % compared to the 10.5-16.8 % $O_2$ was observed while temperature remained below 20 °C (Fig. 4d). This is most likely due to higher nitrite oxidation than nitrite reduction rates in response to increasing oxygenation levels to 21 % $O_2$, which is consistent with the nitrite oxidation step sensitivity to oxygen

limitation (Pollice et al., 2002; Tanaka and Dunn, 1982). Additionally, the $^{15}N$-SP close to 0 ‰ observed at the highest oxygenation level indicates a decreasing contribution to $N_2O$ production of nitrite reduction over hydroxylamine oxidation

pathway. The highest oxygenation level thus limits the reduction pathways (i.e. $NO_2^-$ reduction to $N_2O$ and $N_2O$ reduction to $N_2$) while favoring the ammonium and nitrite oxidation pathways.

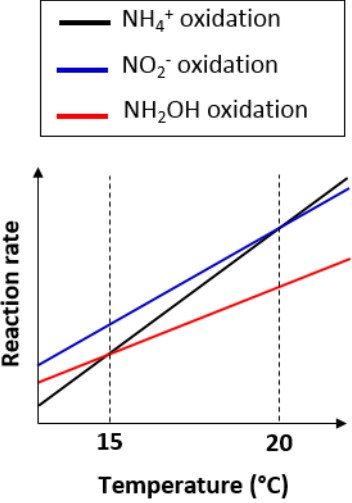

**Figure 6: Scheme of the difference in temperature dependency of the reactions involved in nitrification.**

During the temperature and ammonium concentration tests, the contribution of the hydroxylamine oxidation pathway to $N_2O$
emissions increased with temperature between 13.5 and 19.8 °C (Sect. 3.3) and decreased in favor of the nitrite reduction pathway when temperature exceeded 20 °C (Fig. 6a). The $^{15}N$-SP was low when temperature exceeded 20 °C (-7.3 ± 1 ‰), while being higher than -5 ‰ (-1.3 ± 2.4 ‰) when temperature ranged from 18.2 to 19.8 °C. At temperatures above 20 °C, ammonium oxidation rates exceed nitrite oxidation rates (Fig. 7; Kim et al., 2008; Raimonet et al., 2017). This explains most likely the increased contribution of the nitrite reduction pathway to $N_2O$ emission, as more nitrite becomes available for
nitrifier-denitrification and/or heterotrophic denitrification. As little nitrite accumulated (Fig. S2h), lower rates of nitrite consuming than producing processes can be inferred (nitrite reduction and oxidation vs. ammonium oxidation). Additionally, values of $\delta^{18}O > 35$ ‰ measured during these tests suggest a significant contribution of heterotrophic denitrifiers to $N_2O$ emissions (Snider et al., 2013). This seems to occur at the lowest hydroxylamine oxidation contribution to $N_2O$ production; below 18°C and at 20.3 °C. Furthermore, the denitrifiers were impacted to a larger extent by temperature than ammonium
concentration.

**4 Conclusion**

Our results demonstrated that whatever the imposed conditions, the nitrifying biomass produced $N_2O$ and nitrite reduction remained the main $N_2O$ producing pathway. The $N_2O$ emissions were sensitive to oxygenation, temperature, and $NH_4^+$ concentration likely due to control of enzymatic activities. The use of $N_2O$ isotopocules confirmed the processes that control
$N_2O$ emissions under oxygenation constrain and improved the knowledge of processes that control $N_2O$ under temperature

constrain. Among the environmental variables tested, temperature appears to be the main control on $N_2O$ producing pathways under nitrifying conditions, due to its dissimilar effects on ammonium and nitrite oxidizing activities. Ranges of optimal temperature for nitrification and limited $N_2O$ emissions can be recommended. The combination of low $N_2O$ emissions and high nitrification rates would occur close to 15 °C. From 15 to 20 °C, increasing nitrification rate increase the $N_2O$ emissions by the hydroxylamine oxidation pathway. Above 20 °C, an increasing nitrification rate increases the $N_2O$ emissions via the nitrite reduction pathway.

We studied the impact of environmental variables on $N_2O$ producing pathways based on the isotope analysis of a limited sample number of dissolved N compounds. The approach and conclusions based on the impact of these variables on $N_2O$ emissions most likely applies to nitrification and denitrification in soils, sediments, lake and marine systems. These systems are subject to dynamic environmental conditions, among which ammonium concentrations, oxygenation and temperature. The comparison of the $N_2O$ isotopocules measured and those hypothesized from the literature provides a useful tool to discuss the $N_2O$ producing and consuming process, as well the underlying control mechanisms at play. Ultimately, this can result in mitigation solutions of $N_2O$ emissions by constraining trough space and time the contribution of $N_2O$ producing and consuming pathways. However, additional efforts seem still needed to reduce, if possible, the ranges of $N_2O$ isotope signatures related to each producing and consuming processes.

**Supplementary material**

Additional information about the nitrifying activity of the biomass, the experimental conditions, and the time series of ammonium oxidation experiments.

**Author contribution**

JF, AF and MSp designed the experiments with contributions from GH, MSe and AML. GH, JF and LL carried the experiments out. GH performed the stable isotope measurements with contribution from VV and interpreted them with contribution from MSe. GH and JF processed the data. GH, JF and AML prepared the manuscript with contributions from all co-authors.

**Competing interests**

The authors declare that they have no conflict of interest.

**Acknowledgements**

This work is part of the 'N$_2$OTrack' project ANR-15-CE04-0014-02 funded by the French National Research Agency. Furthermore, the authors are grateful to Sam Azimi and the 'Direction Innovation Environment' of SIAAP for providing the

media colonized by the nitrifying biomass, Mansour Bounouba and Simon Dubos for their assistance in chemical analyses and in setup and development of the nitrifying reactor.

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
