# Peer review of "Isotopic evidence for alteration of nitrous oxide emissions and producing pathways contribution under nitrifying conditions"

_Biogeosciences, 2019_

## Referee Comment (RC1) · Anonymous Referee #1 · 27 Aug 2019

General comments

Humbert et al. report emissions and production/consumption processes of N2O in a nitrifying biofilm reactor which simulates a part of a biological waste water treatment system. Although several similar studies have been published, knowledge of key factors that should be controlled to mitigate N2O emission is still insufficient because there are various type of biological waste water treatment and because processes related to N2O depend on many factors.

Major findings of this paper are that N2O is mainly produced by nitrifier-denitrfication in a nitrifying biofilm reactor and that temperature control is more important than oxygen

concentration or ammonia concentration. They may be worth publishing in Biogeosciences if the authors add implications of their research not only for a specific waste water treatment system but for other systems including natural water or soils.

Although the purpose and conclusion are clearly described, I found several flaws in the manuscript. First, a couple of related studies (Tumendelger et al., 2014; 2016) are not cited and compared with the findings of this study. Second, presentation of results (tables and figures in main text and supplement) is not well organized and is confusing. For example, Table S1 seems to show all the experimental conditions but corresponding results are not shown and figures seem to show only a part of the results. Third, a part of interpretation of isotopic data is not appropriate or based on assumptions that are not clearly explained. Fourth, several sentences are not readable or clear.

In summary, I consider this paper may be acceptable after careful revisions with respect to concerns above and below.

Specific comments

L39 Add Tumendelger et al., 2014 and 2016.

L58–60 I think this statement is vague because equilibrium process is involved in biotic process (e.g., O-exchange between nitrate and water during nitrification and denitrification) and kinetic fractionation also occur in abiotic processes (e.g., diffusion in air or water).

L73 This statement is misleading because many of previous studies cited elsewhere in this manuscript did use combination of isotope data of N2O to analyze production/consumption processes.

L93–94 I think a schematic of the reactor helps readers to understand the experiment and how monitoring of environmental parameters and sample collection were conducted. What is "continuous down-flow counter current mode"?
L102 Here it can be read that the authors made 24 experiments, but in Table S1 total 26 conditions are shown. But in Table S1, the first line in the list of oxygenation tests and the second one for NH4+ concentration tests seem the exactly same condition, and the same for the second of oxygenation tests and the last of NH4 concentration tests. Are these pairs from actually a single experiment? Please explain in footnotes. Also, there is no "n.a." entry in Table S1 in spite of footnote.

L106 If the numbers in Table S1 are correct, NH4 concentration should be "20.3" –21.1 mg N/L and temperature should be "19.0 to 19.8" C. Please check the data carefully.

L108 How did the authors determine that the optimal oxygen concentration is 21%? Table 1. This table is just showing reduced information presented in Table S1 and is not helpful. I suggest to use Table S1 here.

L131 This sentence seems to explain the calibration of dN and dO for ammonium, nitrite, and nitrate. In the case of N2O, dO cannot be calibrated using nitrate standards because there is a kinetic fractionation during N-O bond rapture in nitrate reduction to N2O. SP is also not determinable using the standards listed here. Please explain more.

L140 Confirm the unit. If concentration is multiplied by flow rate, dimension should be mass per time (e.g., mg N /min).

L147 Consider more appropriate title of the section, for example, "Estimation of the range of nitrogen isotope ratio in N2O produced by each biological process".

L157–171 It is strange that these sentences describe how to estimate dN values of output NH4+ and NOx-, because in Fig. S2 concentrations and isotope ratios of N-compounds in inflow and outflow are shown as "measured" parameters. If these are really measured, I think it is worth calculating isotope enrichment factors (epsilons) in the studied system and comparing with previous studies.

L171 It seems that produced NOx- (=NOxout – NOxin) is assumed to be derived from

reacted ammonium. Then "f" in eq (4) should be "1-f".

L172–176ãĂĂThese statements are not correct in a strict sense and are misleading. In a closed system, approximation of isotope effect using difference in delta values is valid only when isotope ratio in substrate can be treated as constant as described in Denk et al. (2017). In an open system, it is true that isotope effect can be expressed as the difference in delta values between product and residual substrate that exit from the system (Fry, 2006). But d15Ns in eq (5) can be read as isotope ratio in substrate before reaction (input) and hence is not applicable to open system. Equation (5) can be derived from an equation similar to eq (2) when f=1, but the authors do not state the assumption that f=1 is appropriate in this study. In fact, the value of f decreases as low as 0.2 (Fig. S1).

L182–183 Is "the ratio between ammonium oxidation rate and influent ammonium concentrations" different from "nitrification efficiency" (=oxidation rate/NH4 feeding rate) defined in L140–141? It is odd that ratio of parameters with different dimension is additionally introduced.

L184–186 I cannot understand what this sentence means. Please rephrase.

L191–193 Here, the possible range of dN of outflowing NH4+ and NOx- is shown, but the dN for each timing is plotted as a single value in Fig. S2. How were these individual values calculated with what assumption?

L195–200 In multi-step reaction (in this case, two step reaction of NH3->NH2OH->N2O is considered), overall isotope effect does not necessarily equal to sum of the isotope effect of each step, but depends on substrate availability and ratio of backward to forward flows in the middle step of the reaction (Rees 1973). Please add basis of the authors's assumption.

L203–204 I believe this is incorrect. The authors did not "observe" the net nitrogen isotope effect for each pathway, but they used literature values.

L224–227 I would like to see the mass balance of N. Judging from Table S2 (which shows results only from NH4+ concentration tests though), increase in NOx- is always lower than decrease in NH4+. Is apparent nitrogen loss explainable by gaseous emission of N2O and NO, or was there significant nitrogen assimilation by the biofilm?

L228–231 I cannot agree that ammonium oxidation rates were "low and stable" for 0–10.5% O2 because two high values were observed at 5% (Fig. 3a). It is not clear whether the authors excluded the two data (because of large error bars?) or not. I understand that these rates are calculated from influent and effluent NH4+ concentrations measured over time as presented in Fig. S2 (again, this figure only shows results from NH4+ concentration tests though), but cannot understand why the error bar ("standard deviation") for the two data is significantly large. Please explain it as well as detailed procedure for calculating "average and standard deviation" (e.g., how many measured data were used for averaging?).

L234–235 I see 8 data points in Fig. 3d (also 3e and 3f), but Table S1 indicates total 13 data were obtained for oxygenation tests. Does this mean isotopic measurements were not conducted for all samples?

L235–236 I cannot see "similar marked change" in d15N at 16.8% O2 and 21% O2. The two data points for each O2 condition depart each other, and when average is taken, there would be no significant difference.

L238–239 Relatively higher SP value was observed not only at 4.2% O2 but also at 20% O2. But do the authors consider N2O reduction occurs at 4.2% O2 just because SP is larger than the range estimated for N2O produced NO2- reduction? It seems to me that the two high SP data is not significantly different considering the large error bars for 20% O2, and that the upper end of estimated range might be underestimate (see Fig. 6 in Denk et al.).

L244–245 Alternative explanations can be made for the decrease in N2O emission. For example, change in branching ratio between NO2- , N2O, and NO production during

NH2OH oxidation might reduce N2O emission. Are there any other evidences for N2O reduction? I agree that N2O reduction might occur when oxygen concentration is really 0%, but as shown in Fig. S1, measured DO is ca. 1.5 mg/L and this enabled NH4 oxidation. It is unlikely that NH4 oxidation and N2O reduction occur at the same time unless there are specific anoxic sites in the system.

L249 What does "independence of samples" mean?

L249–251 I cannot follow these sentences. Please rephrase and describe why the different trends of d15N and d18O can be explained with reaction rates in more detail.

L259–261 Although NH4+ oxidation rate has linear relation to NH4+ concentration (Fig. 4b), the remaining NH4+ fraction does not (Fig. S1f). It seems to increase nonlinearly. Please discuss why this happened. In addition, I cannot see that remaining fraction of NH4 or NH4+ oxidation rate is "negatively correlated to temperature" in Fig. S1e or Fig. 4a. It seems almost constant irrespective of temperature. Is stated correlation really significant? Please show p values.

L264–267 Although I think there is no significant relationship between NH4+ oxidation rate or remaining fraction and temperature, the authors argue that NH4+ remaining fraction is negatively correlated with temperature whereas NH4+ oxidation rate is positively correlated with temperature. Please explain why this apparently contradict trend was observed.

L267–268 Temperature effect (if any) might be explained with enzymatic activity, but I think NH4+ concentration effect can be explained with kinetics of enzymatic reaction like Michaelis-Menten kinetics.

L274 Based on which parameter do the authors find "stronger" effect of temperature and NH4+ concentration on the N2O emission rate than on NH4+ oxidation"? For example, slopes in Fig. 4b and 4d look similar.

L278–281 Please explain in detail why the authors consider nitrite oxidation (to nitrate)

is less important than nitrite reduction (to N2O) in this case.

L284–285 In Tumendelger et al. (2014, 2016), larger SP values were reported under aerobic condition.

L285–286 Ambiguous sentence. Do the authors intend to argue that SP value increases with temperature (13.5C<T<19.5C) and that it also increases with NH4+ concentration when T is set around 19C? Please rephrase. I cannot agree with the latter statement because SP value obtained at 42.8 mg N/L is lower than SP at 28.6 (Fig. 5a).

Figure 5 caption. As I pointed out at L228–231 above, it is not clear how the authors made data reduction based on primary data. How "average and standard deviation" were calculated? How did the authors ensure "the steady-state conditions"?

L315–317 Does "stable NO2-" mean that NO2- concentration was constant over time or that it did not depend on NH4+ concentration? If the authors intend to mean the latter, I cannot agree with them because three data points at 20% O2 in Fig. S1g show a large variation.

L323–324 Sect. 3.2 → Sect. 3.3? There, the authors wrote temperature range as 13.5–19.8ïĆřC, and did not describe "exponentially increase". It seems to me that SP increases with temperature linearly.

Technical corrections

L44 a large "extent"

L67 enriched in "15N at" central position

L81 Add question mark at the end of the sentence.

L103–105 Awkward sentence. Consider other expression than "consisted of".

L230 and elsewhere. Insert "×" between significant (i.e. 0.35) and exponent (i.e. 10-3)

Figure 3 legend. Open circle represents NH4 concentration of 25.1, not 25.3 if Table S1 is correct.

L238–239 Rephrase the subject part ("The 15N-SP . . . levels") of this sentence. A higher amount of N2O "was" reduced to N2.

L273 correlation between . . . and the ammonium oxidation rates (delete "to")

L333 a larger "extent"

References

Rees, C. E. (1973), A steady-state model for sulphur isotope fractionation in bacterial reduction processes, Geochim. Cosmochim. Acta, 37, 1141–1162.

Tumendelger, A., S. Toyoda, and N. Yoshida (2014), Isotopic analysis of N2O produced in a conventional wastewater treatment system operated under different aeration conditions, Rapid Communications in Mass Spectrometry, 28(17), 1883-1892, doi:10.1002/rcm.6973.

Tumendelger, A., S. Toyoda, N. Yoshida, H. Shiomi, and R. Kouno (2016), Isotopocule characterization of N2O dynamics during simulated wastewater treatment under oxic and anoxic conditions, Geochem. J., 50, 105-121, doi:10.2343/geochemj.2.0390.

---

## Referee Comment (RC2) · Anonymous Referee #2 · 3 Sep 2019

The authors report the use of stable isotopes of N2O (bulk and site specific d15N), complementary to N2O and dissolved inorganic nitrogen concentrations to identify the key processes producing N2O in a biofilm reactor used in a local wastewater treatment facility. They showed that nitrite reduction was the primary N2O producing pathway in the reactor irrespective of the experimental conditions (i.e. different percentage of O2, temperature and initial NH4+ concentrations). Temperature, however imposed the greatest effect on N2O emissions compared to the other factors by simultaneously promoting hydroxylamine oxidation pathway.

This study contains interesting dataset particularly on the factors controlling N2O emis-

sions; which could have broader implications to other systems. As such, I think this study has the potential to be an interesting and helpful addition to the literature but to make it so will require a concerted effort. This is because the manuscript is not very well-written. Most part of the manuscript is confusing with either no or invalid justifications were provided for the assumptions made. For example, (1) experimental conditions presented in the tables are different from the ones presented in the graphs but no explanation was provided as to why some of the data points were ignored; (2) some of the interpretations on the trends are misleading and were not supported with statistical analysis; (3) rates of processes were not well-defined and some of the terms were randomly introduced in the discussion without prior definition of the terms; (4) there was no clear distinction on which part of the results were depicted from the literature and which part was obtained from the study; (5) in the method the authors mentioned that they analysed the d15N of nitrate, nitrite and ammonium, they then indicated in the later section that they hypothesized/estimated the values from the proposed equations.

Specific comments:

Line 16: The authors argued in the text that nitrifier-denitrification was the main N2O producing pathway, remove heterotrophic denitrification if this is true

Line 17: Method/procedure to estimate nitrite oxidation rate was not discussed/mentioned. Not clear what you mean here. Consider revising the sentence.

Line 18: State the sub-optimal condition.

Line 19: You mentioned that heterotrophic denitrification could be present, if so, how do you know the N2O was produced from NH4+ not from other substrates given that the inflow also comprised of NO3-?

Line 28: Is there a more recent estimate for N2O emission? WMO?

Line 80: Biofilm reactor is only introduced here and no other explanation on its importance. Perhaps a sentence or two should be included to emphasize on the importance

of these reactors (e.g. are these reactors commonly used in waste water treatment plant and how the efficiency of the reactors affect N2O emissions, why only nitrifying reactor is considered).

Line 94: What do you mean by down-flow counter-current mode? More explanation is required especially for non-expert readers.

Line 98: Is the feeding solution described here the same as your inflow solution? If yes, why the inflow solution comprised of other DIN species not only NH4+ as described. As written, the biofilm is only fed with NH4+ not NO3- and NO2- so where did these species originate from?

Line 102: 24 or 26? There was a total of 26 experimental conditions listed in Table S1

Line 104: I suggest the authors consider removing the first two conditions for the O2 test because the different NH4+ concentrations could be compromising the effect of dissolved O2 on the N2O production. Remove from graphs as well if these data points were included in the graphs.

Line 108: What is the optimal DO level and how was this determined?

Line 110: Check the numbers and cross check with Table S1. Some of the values are different

Line 129: What were the protocols and standards for determination of d15N, d18O and d15N-SP of gaseous N2O? At present, this part of the method is missing.

Line 140: check the units between AOR and N2O-ER. The reported units are not consistent

Line 140: the authors did not seem to discuss on nitrification efficiency throughout the manuscript, please remove this from the method if this is not needed to avoid confusion. Instead consider including the calculation for nitrite oxidation rates as this was briefly mentioned in the abstract and at some stage of the manuscript.

Line 181: Why was the text being placed in supplementary section? These sentences should be moved to the main section.

Line 183: What is the suboptimal condition? Please define and explain how this condition was obtained and on what basis this condition was considered suboptimal.

Line 183 – 186: Not quite sure what you meant here. Please rephrase.

Line 191: The d15N values of the outflowing NH4+ and NOx were estimated from equation? Were they not measured using the same method as the d15N of the inflowing NH4 and NOx? If you did measure the outflowing d15N of NH4 and NOx how did that compare to the ones estimated using the equations? Where did you get the $\varepsilon$ value from and why only 0 and 1 were used for f, given f should represent the fraction of NH4+ or NOx remained in the reactor.

Line 195: I do not see the importance of discussing the net isotope effect of the overall ammonium oxidation here given the main focus of the discussion point here is the importance of hydroxylamine oxidation versus nitrite reduction. Furthermore, I doubt the validity of the assumption made by the authors in estimating the overall net isotope effect of ammonium oxidation to nitrous oxide. The net isotope effect relies heavily on the initial d15N and the availability of the substrate and do not necessarily associate with the total $\varepsilon$ from different part of the processes. Even if the d15N of the substrate is the same, different bacteria culture or organisms tend to generate different fractionation effects. Furthermore, the values cited by the authors especially for the net isotope effects related to hydroxylamine oxidation to N2O were not found in the cited references! Please recheck. It is perhaps more interesting to look at the separate effect of the two processes (i.e. ammonium oxidation to nitrite and hydroxylamine oxidation to N2O) on the overall processing of NH4+ in the reactor. You should have enough data to estimate the net isotope effect of ammonium oxidation to nitrite and discuss how that compares to the literature value?

Line 201: Which method are you referring to? And on what basis that the authors think

that d15N-N2O values here refer specifically to hydroxylamine oxidation? Were these d15-N2O values obtained from all the experiments? Or from a specific experiment (ammonium concentration or temperature or DO)?

Line 203: Did the authors observe the net isotope effects or the values were depicted from previous studies? Be more specific. If they did observe the net isotope effect in this study then why not just use these values in the rest of the discussion?

Line 206: And again, where did you get these values from? Are the values in brackets represent the averages of the d15N of respective analytes? Please specify

Line 207-209: I don't think I quite get what you mean here. Explanation is needed on how ammonium oxidation influences the denitrifiers and why is that relevant to the d15N-N2O derived from nitrite being similar for both bacteria? What if NO3 not NO2 was used as a substrate for denitrifier?

Line 229: Not entirely true because some of the rates were high.

Line 230: To me, there was no increase in N2O emission for the same NH4 concentration for different DO condition. I think the authors should carefully consider the trend by comparing the data points for the same NH4 concentration. Can you please include the slope values of the trend line so that it is easier to compare if there was an increase in the tested parameters.

Line 234: Why only 8 SP values are presented? But you have 13 data points for N2O concentration. Justification is required.

Line 237: Describe the processes rather than just the bacteria – if you mean nitrifier-denitrification and heterotrophic denitrificsation, mention this at the start

Line 239: But there was high SP value at O2 higher than 16.8%. If what you said were correct that nitrous oxide reduction was driving the high SP value at low O2 (4%) then what is driving the high SP at 21% O2?

Line 249: What were the independence samples? How were they defined/sampled? How do you know that the N2O at different O2 condition were not from the same origin? Can't they be a mixture of N2O from different processes? And not quite sure what you meant by '….then partially reduced'.

Line 261: Concentration of NOx increased for which experiment (temperature or NH4 concentration)?

Line 261: Why are these values different from the ones presented in Fig. 4c and 4d? Were you referring to the same thing?

Line 275: I agree that there was a stronger effect of temperature on N2O emission compared to NH4+ oxidation rate but for the effect of NH4 concentration, this could only be controlled by the very high NH4+ concentration – indicating a possible effect of NH4+ concentration on these points.

Line 276: Don't think the authors can draw conclusion on the N2O processes based solely on the NH4+ oxidation and N2O emission rates. Suggest discussing these processes after the discussion on the d15N values.

Line 280: You should also calculate nitrite oxidation rate the same way you did for ammonium to support the contention you made here.

Line 294: Optimal here means? Why was 21% O2 considered optimal? Justifications/explanations are also required as to why 23.8 mg/L was chosen for the temperature experiment?

Line 329: I assume you mean heterotrophic denitrification here? Why? You have just discussed that denitrifiers were sensitive to O2 and can be excluded as an important process contributing N2O at the high O2 conditions.

Figure 1: Very nice figure but references used for the ranges of the d15N d18O and SP should be included in the caption of the figure

Figure 4: Include slope, p and r2 values for each of the plot. Not clear on how the errors on the NH4+ concentration plot were derived. And why only for NH4+ conc, did you see any variations for the temperature experiment as well? You have not mentioned elsewhere that replicates samples were collected. If this was how the error bars were derived state that in the method section.

[Figure]

---

## Author Comment (AC1) · 23 Sep 2019

We would like to thank the referee for the effort and time she/he put in to review our manuscript. We are grateful for her/his valuable comments and will make every attempt to carefully address these comments that will improve the quality of the revised manuscript. Hereafter, the points raised by the referee are written in black, whereas our responses are shown in blue. Citation of our corrections that will take place in the revised version of the manuscript are highlighted in grey.

General comments

Humbert et al. report emissions and production/consumption processes of N2O in a nitrifying biofilm reactor which simulates a part of a biological waste water treatment system. Although several similar studies have been published, knowledge of key factors that should be controlled to mitigate N2O emission is still insufficient because there are various type of biological waste water treatment and because processes related to N2O depend on many factors.

Major findings of this paper are that N2O is mainly produced by nitrifier-denitrfication in a nitrifying biofilm reactor and that temperature control is more important than oxygen concentration or ammonia concentration. They may be worth publishing in Biogeosciences if the authors add implications of their research not only for a specific waste water treatment system but for other systems including natural water or soils.

Although the purpose and conclusion are clearly described, I found several flaws in the manuscript. First, a couple of related studies (Tumendelger et al., 2014; 2016) are not cited and compared with the findings of this study. Second, presentation of results (tables and figures in main text and supplement) is not well organized and is confusing. For example, Table S1 seems to show all the experimental conditions but corresponding results are not shown and figures seem to show only a part of the results. Third, a part of interpretation of isotopic data is not appropriate or based on assumptions that are not clearly explained. Fourth, several sentences are not readable or clear.

In summary, I consider this paper may be acceptable after careful revisions with respect to concerns above and below.

In the revised manuscript, we attempted to add implications of our research for a diversity of systems by adding sentences that underline these perspectives at some strategical parts of the manuscript (i.e. the ends of introduction, and discussion sections).

Our responses to the specific comments of the referee are listed below and allow to address the four concerns lastly raised by the referee to our knowledge.

Specific comments

L39 Add Tumendelger et al., 2014 and 2016.

We agree that the results presented in Tumendelger et al., 2014 and 2016 fall perfectly within the scope of our study. Therefore, we added the references suggested by the referee and removed Tallec et al. 2006 from this reference list.

Temperature, electron donor and acceptor concentrations have been identified to control the $N_2O$ emission from WRRFs (Bollon et al., 2016; Kampschreur et al., 2009; Tumendelger et al., 2014, 2016; Wunderlin et al., 2012).

L58–60 I think this statement is vague because equilibrium process is involved in biotic process (e.g., O-exchange between nitrate and water during nitrification and denitrification) and kinetic fractionation also occur in abiotic processes (e.g., diffusion in air or water).

We reworded this sentence.

The isotopic fractionation results from the difference in equilibrium constant or reaction rate observed between the heavier and lighter isotopes in both abiotic and biotic processes.

L73 This statement is misleading because many of previous studies cited elsewhere in this manuscript did use combination of isotope data of N2O to analyze production/consumption processes.

We agree and reworded this sentence.

Nitrogen and oxygen isotope ratios of $N_2O$ have lower potential for $N_2O$ source identification as compared to $^{15}$N-SP. However, we believe that the use of both isotope approaches should strengthen the conclusions from $^{15}$N-SP and reveal additional isotope effects (Fig. 1).

L93–94 I think a schematic of the reactor helps readers to understand the experiment and how monitoring of environmental parameters and sample collection were conducted. What is "continuous down-flow counter current mode"?

As suggested by the referee, we added a schematic of the reactor as new Figure S1 in the supplementary material.

A lab-scale reactor with a working volume of 9.9 L (colonized Biostyrene® beads and interstitial volume) and a headspace of 1.4 L was operated in continuous down-flow counter-current mode for seven weeks (Fig. S1).

L102 Here it can be read that the authors made 24 experiments, but in Table S1 total 26 conditions are shown. But in Table S1, the first line in the list of oxygenation tests and the second one for NH4+ concentration tests seem the exactly same condition, and the same for the second of oxygenation tests and the last of NH4 concentration tests. Are these pairs from actually a single experiment? Please explain in footnotes. Also, there is no "n.a." entry in Table S1 in spite of footnote.

Twenty-four experiments were performed. Two of them tested effects of both oxygenation and ammonium concentration. As suggested by the referee, we added explanations in the text and in the footnote of Table S1.

The influence of environmental conditions on the ammonium oxidation rates and the $N_2O$ emissions from various combinations of oxygenation levels, temperatures and ammonium concentrations were tested in twenty-four experiments (Tables 1). Note that two of them were used twice; as oxygenation and concentration tests.

*Note that two experiments tested both oxygenation and ammonium concentration.*

L106 If the numbers in Table S1 are correct, NH4 concentration should be "20.3" −21.1 mg N/L and temperature should be "19.0 to 19.8" C. Please check the data carefully.

We are grateful to the referee for raising this mistake. We rechecked our data and changed therefore the text in consistency with Table S1 which was correct.

The oxygenation tests consisted of mixing compressed air and pure nitrogen gas to reach 0 to 21 % $O_2$ in the gas mixture (Fig. S1a). They were performed at five substrate concentrations and at a temperature between 19.2 and 20.6 °C. The temperature tests consisted of cooling the feeding solution directly in the feeding tank (22.3 to 13.5 °C), with an inflow ammonium concentration of 20.3-21.1 mg $NH_4^+$-N $L^{-1}$. At temperatures ranging from 19 to 19.8 °C, the ammonium concentration tests

consisted of an increase (6.2, 28.6 and 62.1 mg $NH_4^+$-N $L^1$) and a decrease (56.1, 42.9, 42.7 and 20.2 mg $NH_4^+$-N $L^{-1}$) in the $NH_4^+$ concentrations in the feeding solution.

L108 How did the authors determine that the optimal oxygen concentration is 21%? Table 1. This table is just showing reduced information presented in Table S1 and is not helpful. I suggest to use Table S1 here.

*We reworded this sentence and added an explanation about the meaning of optimal oxygenation level for the next of the manuscript. Further, as suggested by the referee, Table S1 was used instead of Table 1 in the main text.*

Atmospheric oxygenation level (i.e. 21 % $O_2$ in the gas mixture) was imposed for both tests (Figs. S1b and c). This gas mixture using compressed air with 21 % $O_2$ was considered hereafter as optimal as compared to the oxygen-depleted atmosphere used during the oxygenation tests. Noticeably, the atmospheric oxygenation level is the condition that represents the most optimal conditions of oxygenation applied in nitrification BAF of domestic WRRF.

L131 This sentence seems to explain the calibration of dN and dO for ammonium, nitrite, and nitrate. In the case of N2O, dO cannot be calibrated using nitrate standards because there is a kinetic fractionation during N-O bond rapture in nitrate reduction to N2O. SP is also not determinable using the standards listed here. Please explain more.

*We clarified and reorganized this paragraph.*

These methods consist in the conversion of the substrate (ammonium or nitrite or nitrate) into dissolved $N_2O$. The $\delta^{15}N$ and $\delta^{18}O$ for ammonium, nitrite, and nitrate were hence determined from a calibration curve created with a combination of nitrate or ammonium standards that have undergone the same chemical conversion as the samples (USGS-32, $\delta^{15}N$-$NO_3^-$ = 180 ‰, $\delta^{18}O$-$NO_3^-$ = 25.7 ‰; USGS-34, $\delta^{15}N$-$NO_3^-$ = -1.8 ‰, $\delta^{18}O$-$NO_3^-$ = -27.9 ‰ and USGS-35 $\delta^{15}N$-$NO_3^-$ = 2.7 ‰, $\delta^{18}O$-$NO_3^-$ = 57.5 ‰; or IAEA-N1, $\delta^{15}N$-$NH_4^+$ = 0.4 ‰, IAEA-305A, $\delta^{15}N$-$NH_4^+$ = 39.8 ‰, USGS-25, $\delta^{15}N$-$NH_4^+$ = -30.4 ‰). The quality of calibration was controlled with additional international standards (IAEA-NO-3, $\delta^{15}N$-$NO_3^-$ = 4.7 ‰, $\delta^{18}O$-$NO_3^-$ = 25.6 ‰; or IAEA-N2, $\delta^{15}N$-$NH_4^+$ = 20.3 ‰). Basically, an analytical sequence was comprised of triplicate standards for calibration and quality controls and duplicate samples. The average of the analytical replicates was then used for calibration, quality control and as result.

Since no international standards were available for $N_2O$ isotopes, these were determined the same day as nitrate or ammonium standard analysis insuring correct functioning of the method and analysis. In addition to this, the internal $N_2O$ standards were previously calibrated by exchange with the laboratory of Naohiro Yoshida and Sakae Toyoda at the Tokyo Institute of Technology. All isotope measurements were determined using an isotope ratio mass spectrometer (IRMS, DeltaVplus; Thermo Scientific) in continuous-flow with a purge and trap system coupled with a Finnigan GasBench II system (Thermo Scientific). The precision was 0.8 ‰, 1.5 ‰ and 2.5 ‰ for $\delta^{15}N$, $\delta^{18}O$, and $^{15}N$-SP, respectively.

L140 Confirm the unit. If concentration is multiplied by flow rate, dimension should be mass per time (e.g., mg N /min).

*The unit was corrected.*

The ammonium oxidation rate (AOR) was estimated in each experiment for time ≥ 1 HRT from the difference between influent and effluent $NH_4^+$ concentrations multiplied by the liquid flow rate (kg $NH_4^+$-N $d^{-1}$).

L147 Consider more appropriate title of the section, for example, "Estimation of the range of nitrogen isotope ratio in N2O produced by each biological process".

We clarified the title as suggested by the referee.

2.5 Estimation of ranges of nitrogen isotope ratio in biologically produced $N_2O$

L157–171 It is strange that these sentences describe how to estimate dN values of output NH4+ and NOx-, because in Fig. S2 concentrations and isotope ratios of N compounds in inflow and outflow are shown as "measured" parameters. If these are really measured, I think it is worth calculating isotope enrichment factors (epsilons) in the studied system and comparing with previous studies.

The isotope ratios of N compounds in inflow and outflow were measured only for the three experiments shown in Fig.S2. Although these experiments allow calculating isotope enrichment factors (epsilons), these epsilons unlikely describe the whole range of operating conditions tested in our study. Therefore, we think that using isotope ratio estimates from the literature is more appropriate than values based on a limited number of experiments. We added sentences throughout the manuscript (in 2.5 and 3.1 sections) to better explain this and hence to improve its clarity. In addition, detailed data related to Figure S2 were added as new Table S1.

The isotope effect of reaction step can be determined from the isotope composition of substrates or products. Although being performed on a few tests here, the obtained value can only be applied to a limited number of environmental conditions. The use of estimates from the literature seems therefore suitable.

The $\delta^{15}N$ of inflow ammonium, nitrite and nitrate were -3 ± 0.1 ‰ (n = 3), -15 ± 0.1 ‰ (n = 2), 6.9 ± 0.3 ‰ (n = 3) respectively during ammonium concentration experiments (Fig. S2). The $\delta^{15}N$ of outflowing $NH_4^+$ and $NO_x^-$ were estimated from Eqs. (1-4) with f = 0 or 1, $\varepsilon_{ao}$ = -30 ‰, the highest $[NH_4^+]_{in}$ (62.1 mg N $L^{-1}$) and the lowest $[NO_x^-]_{in}$ (1.4 mg N $L^{-1}$). They ranged from -3 to 27 ‰ and from -32 to 7 ‰, respectively, which encompasses a few isotope compositions measured in the outflow during ammonium concentration tests (Fig. S2 and Table S1).

L171 It seems that produced NOx- (=NOxout – NOxin) is assumed to be derived from reacted ammonium. Then "f" in eq (4) should be "1-f".

We thank the referee for raising this mistake. We corrected Eq. (4) and modified subsequent information (text and figures) related to Eq. (4) throughout the manuscript.

$$\delta^{15}N\text{-}NO_{x,int}^{-} \approx \frac{(\delta^{15}N\text{-}NO_{x,in}^{-} \times ([NO_3^-]_{in} + [NO_2^-]_{in}) + \delta^{15}N\text{-}NO_{x,p}^{-} \times (1-f) \times [NH_4^+]_{in})}{([NO_3^-]_{in} + [NO_2^-]_{in} + (1-f) \times [NH_4^+]_{in})}, \quad (4)$$

L172–176 These statements are not correct in a strict sense and are misleading. In a closed system, approximation of isotope effect using difference in delta values is valid only when isotope ratio in substrate can be treated as constant as described in Denk et al. (2017). In an open system, it is true that isotope effect can be expressed as the difference in delta values between product and residual substrate that exit from the system (Fry, 2006). But d15Ns in eq (5) can be read as isotope ratio in substrate before reaction (input) and hence is not applicable to open system. Equation (5) can be derived from an equation similar to eq (2) when f=1, but the authors do not state the assumption that f=1 is appropriate in this study. In fact, the value of f decreases as low as 0.2 (Fig. S1).

We revised this paragraph considering directly the open system used in this study, and introducing in details the assumptions made and additional calculations that allow describing our system more appropriately than before. Further, we specified that the notations used here are reversed as

compared to those defined in Fry, 2006 while being similar to those presented in Denk et al. 2017. Finally, we renamed $NH_4^+{}_{,out}$ remaining in the reactor after nitrification as $NH_4^+{}_{,res}$; i.e. residual substrate, for better clarity. The $NO_x^-$ pool resulting from mixing between the $NO_x^-$ produced by ammonium oxidation and inflow $NO_x^-$ was renamed as intermediate pool ($NO_{x,int}^-$).

The nitrifying reactor used in this study can be described as an open-system continuously supplied by an infinite substrate pool with constant isotopic composition. A small amount of the infinite substrate pool is transformed into a product pool or a residual substrate pool when flowing through the system. The equations describing the input, output and processes considered here are presented in Fig. 2 after Fry (2006). Note that the definitions of f and $\Delta$ are inverse to the cited literature and that $\Delta_1$ and $\Delta_4$ are null because no fractionation alter the residual substrate exiting the reaction (Fry, 2006). The balance between input and output of each reactional step allows to propose equations for calculation of the nitrogen isotope ratio of compounds in the inflow and outflow of the system (Denk et al., 2017; Fry, 2006). These equations can be simplified under the assumption that limited amount of N compounds are transformed into $N_2O$; i.e. $f_2$ close to 0 and $f_3$ close to 1. Therefore, the N isotope ratios of the residual substrate pool can be approximated from Eq. (1).

$$\delta^{15}N\text{-}NH_4^+{}_{,res} \approx \delta^{15}N\text{-}NH_4^+{}_{,in} - \Delta_2(1 - f_1) \,, \tag{1}$$

Where $f_1$ is the remaining substrate fraction leaving the reactor (i.e. remaining fraction of ammonium), ranging from 0 to 1 (0-100 in %), and $\Delta_2$ is the N isotope enrichment factor associated with ammonium oxidation. In their review, Denk et al. (2017) reported a mean value of -29.6 ± 4.9 ‰ for $\Delta_2$. Therefore, $\delta^{15}N$ is higher for residual than the initial substrate pool ($\delta^{15}N\text{-}NH_4^+{}_{,in} < \delta^{15}N\text{-}NH_4^+{}_{,res}$; where 'in' and 'res' represent inflow and outflow, respectively). Consequently, the pool of product is depleted in heavier isotope (i.e. nitrite and nitrate hereafter defined as $NO_x^-$ pool; $\delta^{15}N\text{-}NO_{x,in}^- > \delta^{15}N\text{-}NO_{x,int}^-$). It is estimated from Eq. (2):

$$\delta^{15}N\text{-}NO_{x,p}^- \approx \delta^{15}N\text{-}NH_4^+{}_{,in} + \Delta_2 f_1 \,, \tag{2}$$

Where $\delta^{15}N\text{-}NO_{x,p}^-$ is the nitrogen isotope ratio of the product pool produced by nitrification. The nitrogen isotope ratio of the overall intermediate $NO_x^-$ exiting this process results of mixing between initial and produced $NO_x^-$ pools ($\delta^{15}N\text{-}NO_{x,int}^-$) and can be estimated from Eqs. (3) and (4):

$$\delta^{15}N\text{-}NO_{x,in}^- = \frac{(\delta^{15}N\text{-}NO_{2,in}^- \times [NO_2^-]_{in} + \delta^{15}N\text{-}NO_{3,in}^- \times [NO_3^-]_{in})}{([NO_3^-]_{in} + [NO_2^-]_{in})} \,, \tag{3}$$

$$\delta^{15}N\text{-}NO_{x,int}^- \approx \frac{(\delta^{15}N\text{-}NO_{x,in}^- \times ([NO_3^-]_{in} + [NO_2^-]_{in}) + \delta^{15}N\text{-}NO_{x,p}^- \times (1-f_1) \times [NH_4^+]_{in})}{([NO_3^-]_{in} + [NO_2^-]_{in} + (1-f_1) \times [NH_4^+]_{in})} \,, \tag{4}$$

Note that $\delta^{15}N\text{-}NO_{x,out}^-$ equals $\delta^{15}N\text{-}NO_{x,int}^-$ when $f_3$ is close to 1 which means that nitrifier-denitrification and heterotrophic denitrification are negligible. Finally, two options must be considered to approximate the nitrogen isotope ratio of $N_2O$ that exits the reactor. On the one hand, $\delta^{15}N\text{-}N_2O$ can be estimated from Eq. (5), when the hydroxylamine oxidation is the producing process of $N_2O$:

$$\delta^{15}N\text{-}N_2O \approx \delta^{15}N\text{-}NH_4^+{}_{,in} - \Delta_2(1 - f_1) + \Delta_3 \,, \tag{5}$$

In addition to the influence of the nitrogen isotope composition of the substrate, the $\delta^{15}N\text{-}N_2O$ depends therefore on the difference between the isotope effects related to the oxidation of $NH_4^+$ to $NO_2^-$ and the oxidation of $NH_2OH$ to $N_2O$ for complete nitrification ($f_1 = 0$), while depending only on the latter for limited nitrification ($f_1 = 1$). On the other hand, $\delta^{15}N\text{-}N_2O$ can be estimated from Eq. (6), when the nitrite reduction is the producing process of $N_2O$:

$$\delta^{15}N\text{-}N_2O \approx \delta^{15}N\text{-}NO_{x,int}^-(1 - f_1)^{-1} + \Delta_5 \,, \tag{6}$$

In addition to the influence of the nitrogen isotope composition of the substrate, when negligible amounts of N$_2$O are produced by nitrite reduction during nitrifier-denitrification or heterotrophic denitrification its nitrogen isotope ratio depends on isotope effect related to this process ($\Delta_5$).

L182–183 Is "the ratio between ammonium oxidation rate and influent ammonium concentrations" different from "nitrification efficiency" (=oxidation rate/NH4 feeding rate) defined in L140–141? It is odd that ratio of parameters with different dimension is additionally introduced.

We thank the referee for raising this mistake. We reworded the sentence consistently with parameters previously introduced. We are speaking about nitrification efficiency here.

Over the range of tested conditions, the ratio between ammonium oxidation rate and influent ammonium load ranged from 10 to 82 %, never exceeding 40 % for suboptimal nitrifying conditions imposed during oxygenation and temperature tests.

L184–186 I cannot understand what this sentence means. Please rephrase.

We rephrased the sentence.

The ammonium concentration, oxygenation level and temperature affected the ammonium oxidation rates, as well N$_2$O emission rates and factors.

L191–193 Here, the possible range of dN of outflowing NH4+ and NOx- is shown, but the dN for each timing is plotted as a single value in Fig. S2. How were these individual values calculated with what assumption?

The standard error plotted here depending on the number of samples after one hydraulic retention time. We added precision in the method section 2.4.

The measurements related to liquid or gas samples were average by experiment; i.e. average of data obtained from the samples taken after one hydraulic retention time.

L195–200 In multi-step reaction (in this case, two step reaction of NH3->NH2OH->N2O is considered), overall isotope effect does not necessarily equal to sum of the isotope effect of each step, but depends on substrate availability and ratio of backward to forward flows in the middle step of the reaction (Rees 1973). Please add basis of the authors's assumption.

Please refer to the previous answer regarding the estimates of $\delta^{15}$N-N$_2$O ranges.

L203–204 I believe this is incorrect. The authors did not "observe" the net nitrogen isotope effect for each pathway, but they used literature values.

We rephrased the sentence.

A higher net nitrogen isotope effect for nitrite reduction than hydroxylamine oxidation pathway was estimated for N$_2$O production.

L224–227 I would like to see the mass balance of N. Judging from Table S2 (which shows results only from NH4+ concentration tests though), increase in NOx- is always lower than decrease in NH4+. Is apparent nitrogen loss explainable by gaseous emission of N2O and NO, or was there significant nitrogen assimilation by the biofilm?

We agree that the presentation of N mass balance would have improved our discussion. However, due to technical problems, NO$_3^-$ analysis were lacking for some oxygenation tests and for the temperature tests, whereas NH$_2$OH, N$_2$ production, as well as N mineralization and assimilation in the biofilm were

not quantified. Therefore, the amount of $N_2O$ emitted alone does not explain the apparent nitrogen loss. No significant amounts of NO were detected during the tests (< 1 ppm measured in the outlet gas), and the accumulation of $NH_2OH$ is unlikely. Heterotrophic denitrification, i.e. the reduction of $NO_x^-$ to $N_2$, most likely adds up the N mass balance. However, $N_2$ measurements (in the gas mixture comprising of at least 79 % $N_2$) were not been feasible and thus not carried out. We added precisions about this in the manuscript.

Ammonium concentrations decreased from 20.2-37.3 to 11.4-31.1 mg N $L^{-1}$; with 45 to 89 % of the inflow ammonium remaining in the outflow during the oxygenation tests (Fig. S1d). When measured, the cumulated concentrations of $NO_2^-$ and $NO_3^-$ ($[NO_x^-]$) increased from 2.4-4.1 to 4.7-11 mg N $L^{-1}$ between inflow and outflow and were composed by at least 74 and 82 % of $NO_3^-$, respectively. The mass balance between N compounds that enter and exit the reactor evidenced a default of up to 5 mg N and impacted each test. No significant amount of NO was detected during any tests (data not shown) whereas neither $NH_2OH$, $N_2$, nor N mineralization/assimilation in the biofilm were quantified. The accumulation of such amounts of $NH_2OH$ is unlikely. Heterotrophic denitrification, i.e. the reduction of $NO_x^-$ and more particularly of $N_2O$ to $N_2$, may explain the incomplete N mass balance. However, the measurement of small $N_2$ variations in the gas mixture that exiting the reactor and comprising at least 79 % $N_2$ was not measured.

L228–231 I cannot agree that ammonium oxidation rates were "low and stable" for 0–10.5% O2 because two high values were observed at 5% (Fig. 3a). It is not clear whether the authors excluded the two data (because of large error bars?) or not. I understand that these rates are calculated from influent and effluent NH4+ concentrations measured over time as presented in Fig. S2 (again, this figure only shows results from NH4+ concentration tests though), but cannot understand why the error bar ("standard deviation") for the two data is significantly large. Please explain it as well as detailed procedure for calculating "average and standard deviation" (e.g., how many measured data were used for averaging?).

We improved clarity of this sentence and precise according to the referee's comment that the average presented here was calculated from all data point in the considered interval (i.e. 0-10.5 % $O_2$ in the gas mixture).

The oxygenation level had contrasting effects on ammonium oxidation rates, and $N_2O$ emission rates and factors (Figs. 3a-c). Between an oxygenation of 0 to 10.5 % $O_2$ in the gas mixture, no clear trend in ammonium oxidation rates was observed although being rather low (1.1 ± 0.5 mg $NH_4^+$-N $min^{-1}$). In the same oxygenation levels interval, the $N_2O$ emission rates and factors increased from 0.35 $10^{-3}$ to 1.6 $10^{-3}$ mg N $min^{-1}$ and from 0.05 to 0.16 %, respectively.

Further, the standard error plotted here depending on the number of gas sampled after one hydraulic retention time. We added precision in the method section 2.4.

The measurements related to liquid or gas samples were average by experiment; i.e. average of data obtained from the samples taken after one hydraulic retention time.

L234–235 I see 8 data points in Fig. 3d (also 3e and 3f), but Table S1 indicates total 13 data were obtained for oxygenation tests. Does this mean isotopic measurements were not conducted for all samples?

The referee is correct. The isotopic measurements were not conducted for all samples. However, the concentration data were all presented to capture the whole dynamic constrained by environmental

conditions tested. We added sentences in the method section 2.2 and in the caption of Fig. 6 for improved clarity.

Note that gas sampling was lacking for 5 of 13 oxygenation tests.

Note that the isotopic measurements of gas samples taken at inflow ammonium concentration of 42.7 and 42.9 mg N L-1 were averaged.

L235–236 I cannot see "similar marked change" in d15N at 16.8% O2 and 21% O2. The two data points for each O2 condition depart each other, and when average is taken, there would be no significant difference.

We understand the concern of the referee, however, it should be noticed that ammonium concentration is also modified between 16.8 and 21 % $O_2$. This additional effect of ammonium mitigates the effect of oxygenation alone that is intended to be shown here. Therefore, the average isotope ratios at 21 % $O_2$ should be compared to the data solely measured for 23.8 $NH_4^+$-N $L^{-1}$ at 16.8 % $O_2$. We specified it more clearly in the manuscript.

A similar marked change in nitrogen and oxygen isotope ratios of $N_2O$ (decrease and increase, respectively) was observed when oxygenation increased from 16.8 to 21 % $O_2$ (Figs. 3e and f). Note that to observe the latter variations the effect of ammonium concentration was not included. One way to do so is to compare the isotope composition average at 21% $O_2$ with the isotope composition measured for 23.8 $NH_4^+$-N $L^{-1}$ at 16.8 % $O_2$.

L238–239 Relatively higher SP value was observed not only at 4.2% O2 but also at 20% O2. But do the authors consider N2O reduction occurs at 4.2% O2 just because SP is larger than the range estimated for N2O produced NO2- reduction? It seems to me that the two high SP data is not significantly different considering the large error bars for 20% O2, and that the upper end of estimated range might be underestimate (see Fig. 6 in Denk et al.).

We agree with the referee. We did suggest that $N_2O$ reduction to $N_2$ likely occurs for oxygen-depleted air due to the anti-correlation observed between the oxygen-depletion and [15]N-SP for oxygenation levels between 4.2 and 16.8 % $O_2$, not due to [15]N-SP at 4.2 % $O_2$ higher than the range of [15]N-SP proposed for the $N_2O$ derived from nitrite reduction.

L244–245 Alternative explanations can be made for the decrease in N2O emission. For example, change in branching ratio between NO2- , N2O, and NO production during NH2OH oxidation might reduce N2O emission. Are there any other evidences for N2O reduction? I agree that N2O reduction might occur when oxygen concentration is really 0%, but as shown in Fig. S1, measured DO is ca. 1.5 mg/L and this enabled NH4 oxidation. It is unlikely that NH4 oxidation and N2O reduction occur at the same time unless there are specific anoxic sites in the system.

Our point here is to suggest the likely occurrence of anoxic microsites within the reactor biomass, due to heterogeneous and varying distribution of air circulation. Currently, two evidences are presented: (i) the increasing [15]N-SP with the decreasing oxygenation levels; and (ii) the decrease in $N_2O$ emission. This could explain the co-occurrence of $NH_4^+$ oxidation and $N_2O$ reduction. Further, the DO presented in Fig. S1 is unfortunately an undetailed picture of what happened within the reactor and more particularly within the colonized media. We were not able to measure the DO within the media and DO was therefore measured at the top of the reactor, where it is likely higher than within the colonized media. This is attested by a few measurements performed in the outflow of the reactor during concentration tests. We observed DO 2.8-3.9 mg $O_2$ $L^{-1}$ lower in the outflow than at the top of the

reactor. We added a reference and an explanation describing the likely co-occurrence of ammonium oxidation and $N_2O$ reduction.

This is also consistent with a possible onset of anoxic microsites within the reactor biomass more likely at 4.2 than 16.8 % $O_2$. The dissolved oxygen (DO) concentration never decreased below ca. 1.5 mg $O_2$ $L^{-1}$ in the bulk solution at the top of the reactor (Fig. S1). However, DO decreases from the bulk reactor solution toward the deeper layers of biofilm due to the activity of ammonium oxidizers (Sabba et al., 2018). This is further exacerbated by heterogeneous and varying distribution of air circulation within the static bed. Therefore, oxygen depletion can be assumed within the biofilm.

L249 What does "independence of samples" mean?

L249–251 I cannot follow these sentences. Please rephrase and describe why the different trends of d15N and d18O can be explained with reaction rates in more detail.

Independence of samples means that the samples are not temporally linked to each other. The $N_2O$ sampled at 4.2 % $O_2$ is not a residual fraction of $N_2O$ produced at 16.8 % O2. We clarified these sentences.

The independence of samples taken during the oxygenation tests can explain this. The $N_2O$ sampled at 4.2 % $O_2$ is not a residual fraction of $N_2O$ produced at 16.8 % $O_2$ that would have undergone a partial reduction. The oxygenation level can alter the isotope fractionation factors through the control of reaction rates, as evidenced for the reduction of $N_2O$ to $N_2$ by Vieten et al. (2007). These authors reported lower reaction rates and increased isotope fractionation factors with increasing oxygenation levels. In our case, a similar phenomenon might have influenced both oxidative and reductive processes leading to the production of $N_2O$ and occurring before its ultimate reduction to $N_2$. However, knowledge, that the controls such as the oxygenation level have on the net isotope effect related to a sequence of non-exclusive oxidative and reductive processes is still lacking and require further investigations.

L259–261 Although NH4+ oxidation rate has linear relation to NH4+ concentration (Fig. 4b), the remaining NH4+ fraction does not (Fig. S1f). It seems to increase nonlinearly. Please discuss why this happened. In addition, I cannot see that remaining fraction of NH4 or NH4+ oxidation rate is "negatively correlated to temperature" in Fig. S1e or Fig. 4a. It seems almost constant irrespective of temperature. Is stated correlation really significant? Please show p values.

We think that the discussion about the difference between ammonium oxidation rate and remaining ammonium fraction in relationship with the ammonium concentration (i.e. linear vs. nonlinear) is out of the scope of our study. However, the nonlinear increase that describes the relationship between the remaining ammonium fraction and the ammonium concentration, and highlighted here by the referee is mainly due to the data recorded at the lowest ammonium concentration (6.2 mg N $L^{-1}$). The low remaining ammonium fraction measured at this inflow $NH_4^+$ concentration can be explained by oxygen limitation being lower at 6.2 mg N $L^{-1}$ than at higher ammonium concentration. Indeed, the decrease in DO toward the deeper layers of biofilm due to the activity of ammonium oxidizers is likely less important at 6.2 mg N $L^{-1}$ than at higher ammonium concentration. A way to check this would have been to measure the isotope composition of $N_2O$ emitted at 6.2 mg N $L^{-1}$ inflow [$NH_4^+$]; i.e. the $^{15}$N-SP is expected much higher at 6.2 mg N $L^{-1}$ due to increase contribution of hydroxylamine oxidation pathway in $N_2O$ production. Unfortunately, the $N_2O$ emission was very low at this inflow ammonium concentration and $N_2O$ isotope composition was not analyzed.

Further, the negative correlation between the remaining ammonium fraction and temperature exists and is significant with p < 0.05 (please see statement about the p values used in our study lines 144-145 in the method section). In addition, please note that the y-axis scale of Fig. S4e is the same as Fig. S4d and f. We applied the same y-axis scale to be able to compare with each other the effects of the tested conditions had on the remaining ammonium fraction. Consequently, the variations are less visible in response to temperature than ammonium concentration. However, the negative correlation does exist.

L264–267 Although I think there is no significant relationship between NH4+ oxidation rate or remaining fraction and temperature, the authors argue that NH4+ remaining fraction is negatively correlated with temperature whereas NH4+ oxidation rate is positively correlated with temperature. Please explain why this apparently contradict trend was observed.

It is not surprising that $NH_4^+$ remaining fraction was negatively correlated to $NH_4^+$ oxidation rate in the temperature tests. At constant inflow ammonium concentration and hydraulic retention time, the increase in the ammonium oxidation rate decreases the amount of residual ammonium that exits the reactor. Consequently, the ammonium remaining fraction decreases.

L267–268 Temperature effect (if any) might be explained with enzymatic activity, but I think NH4+ concentration effect can be explained with kinetics of enzymatic reaction like Michaelis-Menten kinetics.

We agree with the referee and clarified the text.

An increase in temperature and inflow ammonium concentrations both positively influenced the rates of $NH_4^+$ oxidation and $N_2O$ emissions and the emission factor (Fig. 4). The $NH_4^+$ oxidation rate linearly increased from 1.3 to 1.5 mg $NH_4^+$-N $min^{-1}$ with temperature ($r = 0.89$; Fig. 4a) and increased from 0.97 to 3.49 mg $NH_4^+$-N $min^{-1}$ with a tenfold increase in the inflow ammonium concentration ($r = 0.82$; Fig. 4b). These positive correlations are well known in the temperature range investigated here and are likely due to enhanced enzymatic activity and Michaelis-Menten kinetics, respectively (Groeneweg et al., 1994; Kim et al., 2008; Raimonet et al., 2017).

L274 Based on which parameter do the authors find "stronger" effect of temperature and NH4+ concentration on the N2O emission rate than on NH4+ oxidation"? For example, slopes in Fig. 4b and 4d look similar.

The emission factor is calculated as the ratio between $N_2O$ emission and $NH_4^+$ oxidation rates. Therefore, the $N_2O$ emission factor enables to identify when the $N_2O$ emission is more responsive than the $NH_4^+$ oxidation to the ammonium inflow. Furthermore, note that the slopes in Figs. 4b and d cannot be compared with each other, as the y-axis scales are different.

L278–281 Please explain in detail why the authors consider nitrite oxidation (to nitrate) is less important than nitrite reduction (to N2O) in this case.

Given the mass balance of the N compounds, the oxidation of nitrite to nitrate remains the main pathway consuming nitrite. However, an increasing $N_2O$ emission factor results from increase in ammonium oxidation rate driven by temperature or $NH_4^+$ concentration. Therefore, the increase in reduction of $NO_2^-$ to $N_2O$ seems to be a bit more pronounced than the increase in oxidation of nitrite to nitrate. We added precisions in the manuscript.

Both experiments suggest that the increase in $N_2O$ emissions results from the increasing production of $N_2O$ by hydroxylamine oxidation or nitrite reduction in combination with a slow rate of or absence of

N$_2$O reduction to N$_2$. Furthermore, no nitrite accumulation was observed with increasing ammonium oxidation rate and N$_2$O emission factor (Fig. S1i). Therefore, if N$_2$O emission results mainly from the nitrite reduction pathway, this suggests that the nitrite reduction pathway is more responsive to the increasing ammonium oxidation rate than the nitrite oxidation pathway; the latter remaining the main pathway of nitrite consumption.

L284–285 In Tumendelger et al. (2014, 2016), larger SP values were reported under aerobic condition.

We added references to works performed by Tumendelger et al.

This is consistent with previous findings based on the $^{15}$N-SP of N$_2$O emitted from aerobic activated sludge (Toyoda et al., 2011; Tumendelger et al., 2016; Wunderlin et al., 2013). Authors reported $^{15}$N-SP as high as 10 ‰, which can suggest more oxygen limitation favorable to the contribution of the nitrite reduction to N$_2$O production in the nitrifying reactor studied here. The hydroxylamine oxidation can even be the main N$_2$O producing pathway, as evidenced by Tumendelger et al. (2014) in some aerated tank.

L285–286 Ambiguous sentence. Do the authors intend to argue that SP value increases with temperature (13.5C<T<19.5C) and that it also increases with NH4+ concentration when T is set around 19C? Please rephrase. I cannot agree with the latter statement because SP value obtained at 42.8 mg N/L is lower than SP at 28.6 (Fig. 5a).

We rephrased the sentence for improved clarity.

Furthermore, the $^{15}$N-SP increased with temperature between 13.5 and 19.8 °C. Our data suggest that temperature mainly controlled the change in N$_2$O producing pathways within this temperature range (Fig. 5a). The temperature control seems to mitigate here the effect that ammonium concentration can have on the N$_2$O producing pathways evidenced elsewhere. Wunderlin et al. (2012, 2013) observed an increase in $^{15}$N-SP from -1.2 to 1.1 ‰ when inflow [NH$_4^+$] increased from 9 to 15 mg N L$^{-1}$. They also observed 3-6 ‰ decreases in $^{15}$N-SP over the course of ammonium oxidation experiments and suggested that NH$_2$OH oxidation contribution to N$_2$O production increased when conditions of NH$_4^+$ excess, low NO$_2^-$ concentrations and high nitrogen oxidation rate occur simultaneously. Our findings are consistent with the observation of Groeneweg et al. (1994) showing that temperature rather than ammonium concentration influenced the ammonium oxidation rate.

Figure 5 caption. As I pointed out at L228–231 above, it is not clear how the authors made data reduction based on primary data. How "average and standard deviation" were calculated? How did the authors ensure "the steady-state conditions"?

Please see our previous answers. Furthermore, we rephrased the sentence.

Average and standard deviation (error bars) are calculated for the samples taken after one hydraulic retention time.

L315–317 Does "stable NO2-" mean that NO2- concentration was constant over time or that it did not depend on NH4+ concentration? If the authors intend to mean the latter, I cannot agree with them because three data points at 20% O2 in Fig. S1g show a large variation.

We mean that the NO$_2^-$ concentration did not depend on oxygenation levels. We modified this sentence.

The oxygenation, temperature and ammonium concentration tests revealed a strong control of nitrite oxidizing activity and the contribution of the nitrite reduction pathway to N2O production. No

relationship was observed between $NO_2^-$ concentrations and oxygenation. In addition to this, higher $^{15}N$-SP at 21 % compared to the 10.5-16.8 % $O_2$ was observed while temperature remained below 20 °C (Figs. S1g and 3d). This is most likely due to higher nitrite oxidation than nitrite reduction rates in response to increasing oxygenation levels to 21 % O2, which is consistent with the nitrite oxidation step sensitivity to oxygen limitation (Pollice et al., 2002; Tanaka and Dunn, 1982).

L323–324 Sect. 3.2 -> Sect. 3.3? There, the authors wrote temperature range as 13.5–19.8°C, and did not describe "exponentially increase". It seems to me that SP increases with temperature linearly.

We agree with the referee. We rephrased the sentence.

During the temperature and ammonium concentration tests, the contribution of the hydroxylamine oxidation pathway to $N_2O$ emissions increased with temperature between 13.5 and 19.8 °C (Sect. 3.3) and decreased in favor of the nitrite reduction pathway when temperature exceeded 20 °C (Fig. 5a).

Technical corrections

L44 a large "extent"

This change was made.

L67 enriched in "15N at" central position

This change was made.

L81 Add question mark at the end of the sentence.

This change was made.

L103–105 Awkward sentence. Consider other expression than "consisted of".

We modified our sentences.

The oxygenation tests were carried out by mixing compressed air and pure nitrogen gas to reach 0 to 21 % $O_2$ in the gas mixture (Fig. S1a). The tests were performed at five substrate concentrations and at a temperature between 19.2 and 20.6 °C. The temperature tests were carried out by cooling the feeding solution directly in the feeding tank (22.3 to 13.5 °C), with an inflow ammonium concentration of 19.9-21.1 mg $NH_4^+$-N $L^{-1}$. The ammonium concentration tests were run at an increase (6.2, 28.6 and 62.1 mg $NH_4^+$-N $L^1$) and a decrease (56.1, 42.9, 42.7 and 20.2 mg $NH_4^+$-N $L^{-1}$) in the $NH_4^+$ concentrations in the feeding solution, at temperatures ranging from 18.8 to 19.9 °C. An optimal oxygenation level was imposed for both tests (Figs. S1b and c).

L230 and elsewhere. Insert "x" between significant (i.e. 0.35) and exponent (i.e. 10-3)

This change was made.

Figure 3 legend. Open circle represents NH4 concentration of 25.1, not 25.3 if Table S1 is correct.

This change was made.

L238–239 Rephrase the subject part ("The 15N-SP … levels") of this sentence. A higher amount of N2O "was" reduced to N2.

We rephrased this sentence.

An additional suggestion can be made from the $^{15}N$-SP dynamics between and variations within the oxygenation levels. They suggest a higher amount of $N_2O$ was reduced to $N_2$ at 4.2 than 16.8 % $O_2$.

L273 correlation between … and the ammonium oxidation rates (delete "to")

This change was made.

L333 a larger "extent"

This change was made.

[Figure]

**Figure S1. Schematic of the nitrifying reactor used in this study. Note that solution was down-flowing, while air was up-flowing.**

[Figure]

**Equations:**

**Nitrification:**

$$\delta^{15}N-NH_4^+{}_{,res} \approx \delta^{15}N-NH_4^+{}_{,in} - \Delta_2(1-f_1) + (\Delta_3 - \Delta_2)f_2$$

$$\delta^{15}N-NO_x^-{}_{,p} \approx \delta^{15}N-NH_4^+{}_{,in} + \Delta_2 f_1 + (\Delta_3 - \Delta_2)f_2$$

$$\delta^{15}N-NO_x^-{}_{,int} \approx \frac{(\delta^{15}N-NO_x^-{}_{,in} \times ([NO_3^-]_{in} + [NO_2^-]_{in}) + \delta^{15}N-NO_x^-{}_{,p} \times (1-f_1-f_2) \times [NH_4^+]_{in})}{([NO_3^-]_{in} + [NO_2^-]_{in} + (1-f_1-f_2) \times [NH_4^+]_{in})}$$

$$\delta^{15}N-N_2O \approx \delta^{15}N-NH_4^+{}_{,in} - \Delta_2(1-f_1+f_2) + \Delta_3(1+f_2)$$

**Nitrifier-denitrification / Heterotrophic denitrification:**

$$\delta^{15}N-NO_x^-{}_{,out} \approx \delta^{15}N-NO_x^-{}_{,int}(1-f_1-f_2)^{-1} - \Delta_5(1-f_3)$$

$$\delta^{15}N-N_2O \approx \delta^{15}N-NO_x^-{}_{,int}(1-f_1-f_2)^{-1} + f_3\Delta_5$$

**Figure 2. Diagram and equations of the nitrifying reactor adapted from Fry (2006). It is considered as a sequence of two reactor boxes. (i) The nitrification of inflow ammonium ($NH_4^+{}_{,in}$) to a pool of nitrite and nitrate ($NO_x^-{}_{,p}$), residual ammonium ($NH_4^+{}_{,res}$), and nitrous oxide ($N_2O$) through the hydroxylamine oxidation pathway. (ii) The subsequent reduction of intermediate $NO_x^-{}_{,int}$; mixing of inflow $NO_x^-{}_{,in}$ and formed $NO_x^-{}_{,p}$ to nitrous oxide ($N_2O$) through the nitrite reduction pathway, and residual $NO_x^-$ that exits the reactor ($NO_x^-{}_{,out}$). Note that residual substrates and formed product exit the reactor without further isotope fractionation. See text for details.**

---

## Author Comment (AC2) · 23 Sep 2019

We would like to thank the referee for the effort and time she/he put in to review our manuscript. We are grateful for her/his valuable comments and will make every attempt to carefully address these comments that will improve the quality of the revised manuscript. Hereafter, the points raised by the referee are written in black, whereas our responses are shown in blue. Citation of our corrections that will take place in the revised version of the manuscript are highlighted in grey.

The authors report the use of stable isotopes of N2O (bulk and site specific d15N), complementary to N2O and dissolved inorganic nitrogen concentrations to identify the key processes producing N2O in a biofilm reactor used in a local wastewater treatment facility. They showed that nitrite reduction was the primary N2O producing pathway in the reactor irrespective of the experimental conditions (i.e. different percentage of O2, temperature and initial NH4+ concentrations). Temperature, however imposed the greatest effect on N2O emissions compared to the other factors by simultaneously promoting hydroxylamine oxidation pathway.

This study contains interesting dataset particularly on the factors controlling N2O emissions; which could have broader implications to other systems. As such, I think this study has the potential to be an interesting and helpful addition to the literature but to make it so will require a concerted effort. This is because the manuscript is not very well-written. Most part of the manuscript is confusing with either no or invalid justifications were provided for the assumptions made.

For example, (1) experimental conditions presented in the tables are different from the ones presented in the graphs but no explanation was provided as to why some of the data points were ignored;

The isotopic measurements were not conducted for all samples. However, the concentration data were all presented to capture the whole dynamic constrained by environmental conditions tested. We added sentences in the method section 2.2 and in the caption of Fig. 6 for improved clarity.

(2) some of the interpretations on the trends are misleading and were not supported with statistical analysis;

Please see our specific answers below. When lacking, we added statistics to support our statement in the results section.

(3) rates of processes were not well-defined and some of the terms were randomly introduced in the discussion without prior definition of the terms;

We carefully defined the processes rates in the Methods section in consistence with those used further in the manuscript.

(4) there was no clear distinction on which part of the results were depicted from the literature and which part was obtained from the study;

We carefully checked our manuscript and used appropriate terms throughout the manuscript to better specify where data came from.

(5) in the method the authors mentioned that they analysed the d15N of nitrate, nitrite and ammonium, they then indicated in the later section that they hypothesized/estimated the values from the proposed equations.

The isotope ratios of N compounds in inflow and outflow were measured only for the three experiments shown in Fig.S2. Although these experiments allow calculating isotope enrichment factors (epsilons), these epsilon values unlikely describe the whole range of operating conditions tested in our study. Therefore, we think that using isotope ratio estimates from the literature is more appropriate

than values based on a limited number of experiments. We added sentences throughout the manuscript (in 2.5 and 3.1 sections) to better explain this and hence to improve its clarity.

The isotope effect of reaction step can be determined from the isotope composition of substrates or products. Although being performed on a few tests here, the obtained value can only be applied to a limited number of environmental conditions. The use of estimates from the literature seems therefore suitable.

The $\delta^{15}N$ of inflow ammonium, nitrite and nitrate were -3 ± 0.1 ‰ (n = 3), -15 ± 0.1 ‰ (n = 2), 6.9 ± 0.3 ‰ (n = 3) respectively during ammonium concentration experiments (Fig. S2). The $\delta^{15}N$ of outflowing $NH_4^+$ and $NO_x^-$ were estimated from Eqs. (1-4) with f = 0 or 1, $\varepsilon_{ao}$ = -30 ‰, the highest $[NH_4^+]_{in}$ (62.1 mg N $L^{-1}$) and the lowest $[NO_x^-]_{in}$ (1.4 mg N $L^{-1}$). They ranged from -3 to 27 ‰ and from -32 to 7 ‰, respectively, which encompasses a few isotope compositions measured in the outflow during ammonium concentration tests (Fig. S2 and Table S1).

Specific comments:

Line 16: The authors argued in the text that nitrifier-denitrification was the main N2O producing pathway, remove heterotrophic denitrification if this is true

The $^{15}N$-SP does not allow differentiating between nitrifier-denitrification and heterotrophic denitrification contribution, both involving the nitrite reduction pathway that produces $N_2O$. We checked our manuscript, nitrifier-denitrification was not indicated as the main $N_2O$ producing pathway. We are not sure having correctly understood the remark of the reviewer.

Line 17: Method/procedure to estimate nitrite oxidation rate was not discussed/mentioned. Not clear what you mean here. Consider revising the sentence.

Line 18: State the sub-optimal condition.

We agree that nitrite oxidation rate was not calculated. However, we inferred lower nitrite oxidation than ammonium oxidation rate from lower $NO_x^-$ than oxidized ammonium budget. We clarified this throughout the manuscript. In addition to this, we changed sub-optimal oxygen levels to oxygen-depleted atmosphere.

Difference between oxidative and reductive rates of nitrite consumption was deduced from $NO_2^-$ concentration and $N_2O$ emission. Hence, nitrite oxidation rates seem to decrease as compared to ammonium oxidation rates at temperatures above 20 °C and under oxygen-depleted atmosphere, increasing $N_2O$ production by the nitrite reduction pathway.

Line 19: You mentioned that heterotrophic denitrification could be present, if so, how do you know the N2O was produced from NH4+ not from other substrates given that the inflow also comprised of NO3-?

We hypothesized here that the $NO_2^-$ produced from the oxidized $NH_4^+$ can be further reduced to $N_2O$ either by nitrifier-denitrification or heterotrophic denitrification. However, the $^{15}N$-SP of $N_2O$ does not allow differentiating between the contributions from both processes. In addition to this, the range of $\delta^{15}N$-$N_2O$ estimates takes into account the isotope composition of inflow $NO_2^-$ and $NO_3^-$.

Line 28: Is there a more recent estimate for N2O emission? WMO?

We did not find more recent similar data in the WMO database (i.e. fraction of $N_2O$ emitted from wastewater resource recovery facilities), and we do not think that it has changed by order of magnitude during the last 5 years. For this reason, the initial statement was not modified.

Line 80: Biofilm reactor is only introduced here and no other explanation on its importance. Perhaps a sentence or two should be included to emphasize on the importance of these reactors (e.g. are these reactors commonly used in waste water treatment plant and how the efficiency of the reactors affect N2O emissions, why only nitrifying reactor is considered).

We agree with the reviewer, we added sentences to better justify why we worked on a biofilm.

In order to achieve this, the nitrifying biomass of a submerged fixed-bed biofilm reactor was investigated. Among the wastewater treatment systems, the biofilm systems are adapted to large urban areas owing to their compactness, flexibility and reliability. An increase of their development can be thought in response to the additional 2.5 billion humans expected in urban areas by 2050 (United Nations Population Division, 2018). However, the biofilm systems have received much less attention than the suspended biomass systems and the relations between the $N_2O$ producing/consuming pathways and controls remain largely unknown (Sabba et al., 2018; Todt and Dörsch, 2016).

Line 94: What do you mean by down-flow counter-current mode? More explanation is required especially for non-expert readers.

It means that feeding solution flows through the reactor from the top to the bottom, while aeration being injected from the bottom of the reactor. We added a schematic of the reactor as new Figure S1 in the supplementary material. This should improve the clarity of the system description.

A lab-scale reactor with a working volume of 9.9 L (colonized Biostyrene® beads and interstitial volume) and a headspace of 1.4 L was operated in continuous down-flow counter-current mode for seven weeks (Fig. S1).

Line 98: Is the feeding solution described here the same as your inflow solution? If yes, why the inflow solution comprised of other DIN species not only NH4+ as described. As written, the biofilm is only fed with NH4+ not NO3- and NO2- so where did these species originate from?

As we used tap water to prepare the feeding solution, containing $NO_2^-$ and $NO_3^-$. We added in brackets the average $NO_2^-$ and $NO_3^-$ concentration in the tap water.

The feeding solution consisted of ammonium chloride ($NH_4Cl$) as substrate, monobasic potassium phosphate ($KH_2PO_4$) as phosphorus source for bacterial growth, and sodium hydrogen carbonate ($NaHCO_3$) as pH buffer and inorganic carbon source in 100 or 150 L of tap water (comprising in average $0.2 \pm 0.4$, $2.4 \pm 1.1$, and $2.5 \pm 1.3$ mg N $L^{-1}$ of $NO_2^-$, $NO_3^-$ and $NO_x^-$, respectively).

Line 102: 24 or 26? There was a total of 26 experimental conditions listed in Table S1

Twenty-four experiments were performed. Two of them tested effects of both oxygenation and ammonium concentration. We added explanations in the text and in the footnote of Table S1.

The influence of environmental conditions on the ammonium oxidation rates and the $N_2O$ emissions from various combinations of oxygenation levels, temperatures and ammonium concentrations were tested in twenty-four experiments (Tables 1). Note that two of them were used twice; as oxygenation and concentration tests.

*Note that two experiments tested both oxygenation and ammonium concentration.*

Line 104: I suggest the authors consider removing the first two conditions for the O2 test because the different NH4+ concentrations could be compromising the effect of dissolved O2 on the N2O production. Remove from graphs as well if these data points were included in the graphs.

The isotopic measurements were not conducted for all samples (please see above). However, concentration data were all presented to capture the whole dynamic constrained by oxygen, temperature and ammonia conditions. Noticeably, if we remove the first two conditions of the oxygenation tests as suggested by the referee, we will not have any isotope data at 21% $O_2$. Therefore, we kept these first two conditions. However, we agree with the referee that the additional effect of ammonium can mitigate the effect of oxygenation alone that is intended to be shown here. Therefore, the average isotope ratios at 21 % $O_2$ should be compared to the data solely measured for 23.8 $NH_4^+$-N $L^{-1}$ at 16.8 % $O_2$. We specified it more clearly in the manuscript where these results are described (Lines 235-236 of original manuscript).

A similar marked change in nitrogen and oxygen isotope ratios of $N_2O$ (decrease and increase, respectively) was observed when oxygenation increased from 16.8 to 21 % $O_2$ (Figs. 3e and f). Note that to observe the latter variations the effect of ammonium concentration was not included. One way to do so is to compare the isotope composition average at 21% $O_2$ with the isotope composition measured for 23.8 $NH_4^+$-N $L^{-1}$ at 16.8 % $O_2$.

Line 108: What is the optimal DO level and how was this determined?

We reworded this sentence and added an explanation about the meaning of optimal oxygenation level for the next of the manuscript.

Atmospheric oxygenation level (i.e. 21 % $O_2$ in the gas mixture) was imposed for both tests (Figs. S1b and c). This gas mixture using compressed air with 21 % $O_2$ was considered hereafter as optimal as compared to the oxygen-depleted atmosphere used during the oxygenation tests. Noticeably, the atmospheric oxygenation level is the condition that represents the most optimal conditions of oxygenation applied in nitrification BAF of domestic WRRF.

Line 110: Check the numbers and cross check with Table S1. Some of the values are different

We are grateful to the referee for raising this mistake. We rechecked our data and changed therefore the text and table in consistency with Table S1. Furthermore, Table S1 was used instead of Table 1 in the main text.

Line 129: What were the protocols and standards for determination of d15N, d18O and d15N-SP of gaseous N2O? At present, this part of the method is missing.

We clarified and reorganized this paragraph as follow:

These methods consist in the conversion of the substrate (ammonium or nitrite or nitrate) into dissolved $N_2O$. The $\delta^{15}N$ and $\delta^{18}O$ for ammonium, nitrite, and nitrate were hence determined from a calibration curve created with a combination of nitrate or ammonium standards that have undergone the same chemical conversion as the samples (USGS-32, $\delta^{15}N$-$NO_3^-$ = 180 ‰, $\delta^{18}O$-$NO_3^-$ = 25.7 ‰; USGS-34, $\delta^{15}N$-$NO_3^-$ = -1.8 ‰, $\delta^{18}O$-$NO_3^-$ = -27.9 ‰ and USGS-35 $\delta^{15}N$-$NO_3^-$ = 2.7 ‰, $\delta^{18}O$-$NO_3^-$ = 57.5 ‰; or IAEA-N1, $\delta^{15}N$-$NH_4^+$ = 0.4 ‰, IAEA-305A, $\delta^{15}N$-$NH_4^+$ = 39.8 ‰, USGS-25, $\delta^{15}N$-$NH_4^+$ = -30.4 ‰). The quality of calibration was controlled with additional international standards (IAEA-NO-3, $\delta^{15}N$-$NO_3^-$ = 4.7 ‰, $\delta^{18}O$-$NO_3^-$ = 25.6 ‰; or IAEA-N2, $\delta^{15}N$-$NH_4^+$ = 20.3 ‰). Basically, an analytical sequence was comprised of triplicate standards for calibration and quality controls and duplicate samples. The average of the analytical replicates was then used for calibration, quality control and as result.

Since no international standards were available for $N_2O$ isotopes, these were determined the same day as nitrate or ammonium standard analysis insuring correct functioning of the method and analysis. In addition to this, the internal $N_2O$ standards were previously calibrated by exchange with the

laboratory of Naohiro Yoshida and Sakae Toyoda at the Tokyo Institute of Technology. All isotope measurements were determined using an isotope ratio mass spectrometer (IRMS, DeltaVplus; Thermo Scientific) in continuous-flow with a purge and trap system coupled with a Finnigan GasBench II system (Thermo Scientific). The precision was 0.8 ‰, 1.5 ‰ and 2.5 ‰ for $\delta^{15}N$, $\delta^{18}O$, and $^{15}N$-SP, respectively.

Line 140: check the units between AOR and N2O-ER. The reported units are not consistent

The unit was corrected.

The ammonium oxidation rate (AOR) was estimated in each experiment for time ≥ 1 HRT from the difference between influent and effluent $NH_4^+$ concentrations multiplied by the liquid flow rate (kg $NH_4^+$-N $d^{-1}$).

Line 140: the authors did not seem to discuss on nitrification efficiency throughout the manuscript, please remove this from the method if this is not needed to avoid confusion. Instead consider including the calculation for nitrite oxidation rates as this was briefly mentioned in the abstract and at some stage of the manuscript.

We did not calculate the nitrite oxidation rate. As part of the $NO_2^-$ and $NO_3^-$ produced from the oxidized ammonium could then react to other N compounds (NO, $N_2O$, $N_2$, $NO_3^-$, $NO_2^-$), we were not able to estimate the nitrite oxidation rate. The discussion about this rate is speculative from the remaining $NO_2^-$ concentration in the reactor outflow. We clarified this throughout the manuscript.

Further, we kept the introduction of nitrification efficiency as we use it at the beginning of the results and discussion section. However, we reworded the sentence consistently with parameters introduced in the method section.

Over the range of tested conditions, the ratio between ammonium oxidation rate and influent ammonium load ranged from 10 to 82 %, never exceeding 40 % for suboptimal nitrifying conditions imposed during oxygenation and temperature tests.

Line 181: Why was the text being placed in supplementary section? These sentences should be moved to the main section.

As suggested by the referee, we moved these sentences to the main text.

The experiments that tested the influence of ammonium concentrations on ammonium oxidation and nitrous oxide emissions also supported the nitrifying activity of the reactor. During these experiments, decreases in $[NH_4^+]$, increases in $[NO_2^-]$ and $[NO_3^-]$ were observed, while pH remaining below 8 prevented any relevant loss of ammonium by volatilization. For example, $[NH_4^+]$ decreased from 6.2 to 1.1, from 28.6 to 17 and from 62.1 to 49.1 mg N L-1 by flowing through the nitrifying biomass. At the same time, $[NO_2^-]$ and $[NO_3^-]$ increased from 0 to 0.2-0.3 mg N L-1 and from 1.4-1.8 to 5-10 mg N L-1, respectively.

Line 183: What is the suboptimal condition? Please define and explain how this condition was obtained and on what basis this condition was considered suboptimal.

We specified what we mean by suboptimal conditions between brackets.

Over the range of tested conditions, the ratio between ammonium oxidation rate and influent ammonium load ranged from 10 to 82 %, never exceeding 40 % for suboptimal nitrifying conditions imposed during oxygenation and temperature tests (i.e. oxygenation levels < 21 % $O_2$, and temperatures < 20 °C).

Line 183 – 186: Not quite sure what you meant here. Please rephrase.

We rephrased the sentence.

The ammonium concentration, oxygenation level and temperature affected the ammonium oxidation rates, as well N₂O emission rates and factors.

Line 191: The d15N values of the outflowing NH4+ and NOx were estimated from equation? Were they not measured using the same method as the d15N of the inflowing NH4 and NOx? If you did measure the outflowing d15N of NH4 and NOx how did that compare to the ones estimated using the equations? Where did you get the "value from and why only 0 and 1 were used for f, given f should represent the fraction of NH4+ or NOx remained in the reactor.

The isotope ratios of N compounds in inflow and outflow were measured only for the three experiments shown in Fig.S2. Although these experiments allow calculating isotope enrichment factors (epsilons), these epsilons unlikely describe the whole range of operating conditions tested in our study. Therefore, we think that using isotope ratio estimates from the literature is more appropriate than values based on a limited number of experiments. We added text throughout the manuscript (in 2.5 and 3.1 sections) to better explain this and hence to improve its clarity. In addition, detailed data related to Figure S2 were added as new Table S1.

The isotope effect of a reaction step can be determined from the isotope composition of substrates or products. Although being performed on a few tests here, the obtained value can only be applied to a limited number of environmental conditions. The use of estimates from the literature seems therefore suitable.

The $\delta^{15}$N of inflow ammonium, nitrite and nitrate were -3 ± 0.1 ‰ (n = 3), -15 ± 0.1 ‰ (n = 2), 6.9 ± 0.3 ‰ (n = 3) respectively during ammonium concentration experiments (Fig. S2). The $\delta^{15}$N of outflowing $NH_4^+$ and $NO_x^-$ were estimated from Eqs. (1-4) with f = 0 or 1, $\varepsilon_{ao}$ = -30 ‰, the highest $[NH_4^+]_{in}$ (62.1 mg N $L^{-1}$) and the lowest $[NO_x^-]_{in}$ (1.4 mg N $L^{-1}$). They ranged from -3 to 27 ‰ and from -32 to 7 ‰, respectively, which encompasses a few isotope compositions measured in the outflow during ammonium concentration tests (Fig. S2 and Table S1).

Line 195: I do not see the importance of discussing the net isotope effect of the overall ammonium oxidation here given the main focus of the discussion point here is the importance of hydroxylamine oxidation versus nitrite reduction. Furthermore, I doubt the validity of the assumption made by the authors in estimating the overall net isotope effect of ammonium oxidation to nitrous oxide. The net isotope effect relies heavily on the initial d15N and the availability of the substrate and do not necessarily associate with the total " from different part of the processes. Even if the d15N of the substrate is the same, different bacteria culture or organisms tend to generate different fractionation effects. Furthermore, the values cited by the authors especially for the net isotope effects related to hydroxylamine oxidation to N2O were not found in the cited references! Please recheck. It is perhaps more interesting to look at the separate effect of the two processes (i.e. ammonium oxidation to nitrite and hydroxylamine oxidation to N2O) on the overall processing of NH4+ in the reactor. You should have enough data to estimate the net isotope effect of ammonium oxidation to nitrite and discuss how that compares to the literature value?

Please see our previous answer regarding the net isotope effect estimate of ammonium oxidation to nitrite. Further, we did not added hydroxylamine directly into the feeding solution. Ammonium was added into the feeding solution. However, the production of N₂O from hydroxylamine oxidation during the oxidation of ammonium cannot be excluded. Therefore, we estimated the range of $\delta^{15}$N-N₂O

produced from hydroxylamine oxidation with ammonium as substrate. Thanks to existing literature, our method considered the broad range of values taken by both the $\delta^{15}N$ of hydroxylamine produced during ammonium oxidation and the fractionation effects related to both ammonium oxidation and hydroxylamine oxidation to $N_2O$. Further, we rechecked the values used from cited references. Correction involved minor changes (<1 ‰).

The net isotope effect of $N_2O$ production by ammonium oxidation via hydroxylamine can be estimated by combining the isotope effects of ammonium oxidation and hydroxylamine oxidation to $N_2O$. The net isotope effect associated to the ammonium oxidation to nitrite ranges from -38.2 to -14.2 ‰ (Casciotti et al., 2003) and can approximate the nitrogen isotope ratio of hydroxylamine transitory produced. The isotope effect related to hydroxylamine oxidation to $N_2O$ ranging from -26 to 5.7 ‰ from data in Sutka et al. (2003, 2006), the net isotope effect of $N_2O$ production by ammonium oxidation via hydroxylamine can range from -64.2 ‰ (-26 + (-38.2)) to -8.5 ‰ (5.7 + (-14.2)). Considering the range of nitrogen isotope ratio of initial ammonium, this method provided a broad range of $\delta^{15}N$ values, from -68 to 19 ‰, for $N_2O$ produced by hydroxylamine oxidation that encompassed the values proposed by others (-46.5 and -32.9 ‰; Sutka et al., 2006; Yamazaki et al., 2014).

Line 201: Which method are you referring to? And on what basis that the authors think that d15N-N2O values here refer specifically to hydroxylamine oxidation? Were these d15-N2O values obtained from all the experiments? Or from a specific experiment (ammonium concentration or temperature or DO)?

In this paragraph, a general way to estimate ranges of nitrogen isotope ratio in $N_2O$ is proposed. This relies on the method presented in 2.5 and the use in Eqs. (5) and (6) of fractionation estimates obtained from the literature. As stated above, we think that using isotope ratio estimates from the literature is more appropriate than measured values based on a limited number of experiments. Published isotope effects were then used. We assume that the range of published isotope effects enables taking into account a high diversity of environmental conditions and bacterial community involved in these isotope effects.

Line 203: Did the authors observe the net isotope effects or the values were depicted from previous studies? Be more specific. If they did observe the net isotope effect in this study then why not just use these values in the rest of the discussion?

We rephrased the sentence.

A higher net nitrogen isotope effect for nitrite reduction than hydroxylamine oxidation pathway was estimated for $N_2O$ production.

Line 206: And again, where did you get these values from? Are the values in brackets represent the averages of the d15N of respective analytes? Please specify

These values come from the net isotope effect found in literature and mentioned in the introduction section. 'These values are imbricated between -52.8 and -6 ‰, the range of net isotope effects related to the $N_2O$ production through nitrite reduction performed by nitrifiers or denitrifiers (Lewicka-Szczebak et al., 2014; Sutka et al., 2008).' (Lines 63-65 of original manuscript). However, we agree that a reminder is required here.

Prior to being reduced to $N_2O$ through the nitrite reduction pathway, $NO_x^-$ was mainly derived from ammonium oxidation in the nitrifying system (Eqs. (1-4)); the resulting $\delta^{15}N\text{-}NO_x^-$ ranging from -32 to 7 ‰. In addition to this, the net isotope effects related to the $N_2O$ production through nitrite reduction performed by nitrifiers or denitrifiers ranges from -52.8 to -6 ‰, the (Lewicka-Szczebak et al., 2014;

Sutka et al., 2008). Consequently, the $\delta^{15}N$ of N$_2$O produced by nitrite reduction ranged from -85 ‰ (-53 + (-32)) to 1 ‰ (-6 + 7), according to Eq. (6).

Line 207-209: I don't think I quite get what you mean here. Explanation is needed on how ammonium oxidation influences the denitrifiers and why is that relevant to the d15N-N2O derived from nitrite being similar for both bacteria? What if NO3 not NO2 was used as a substrate for denitrifier?

We mean that ammonium is oxidized to NO$_2^-$ and NO$_3^-$ in the nitrifying reactor. Both NO$_2^-$ and NO$_3^-$ can be reduced through either nitrifier-denitrification or heterotrophic denitrification or a combination of both reductive processes. However, the isotope composition of the substrate (NO2- NO3-) is imposed by the prior oxidation of ammonium to NO$_2^-$/NO$_3^-$, whatever the reductive process at play (nitrifier-denitrification or heterotrophic denitrification).

Line 229: Not entirely true because some of the rates were high.

We improved clarity of this sentence.

The oxygenation level had contrasting effects on ammonium oxidation rates, and N$_2$O emission rates and factors (Figs. 3a-c). Between an oxygenation of 0 to 10.5 % O$_2$ in the gas mixture, no clear trend in ammonium oxidation rates was observed although being rather low (1.1 ± 0.5 mg NH$_4^+$-N min$^{-1}$). In the same oxygenation levels interval, the N$_2$O emission rates and factors increased from 0.35 10$^{-3}$ to 1.6 10$^{-3}$ mg N min$^{-1}$ and from 0.05 to 0.16 %, respectively.

Line 230: To me, there was no increase in N2O emission for the same NH4 concentration for different DO condition. I think the authors should carefully consider the trend by comparing the data points for the same NH4 concentration. Can you please include the slope values of the trend line so that it is easier to compare if there was an increase in the tested parameters.

We agree that this section needs precisions. Note that we applied the same y-axis scale to be able to compare with each other the effects of the tested conditions had on the N$_2$O emissions (Figs. 3b, 4c and 4d). Consequently, the variations are less visible in response to oxygenation and temperature than ammonium concentration. Further, we do not want to apply a trend line to the whole dataset that includes different inflow ammonium concentrations. In addition to this, there is a limited number of data related to each inflow ammonium concentration in the interval 0-10.5 % O$_2$ (1 to 3 data point). Therefore, the application of a trend line to this limited number of data presents a limited interest. We described the results with more precision.

The oxygenation level had contrasting effects on ammonium oxidation rates, and N$_2$O emission rates and factors (Figs. 3a-c). Between an oxygenation of 0 to 10.5 % O$_2$ in the gas mixture, no clear trend in ammonium oxidation rates was observed although being rather low (1.1 ± 0.5 mg NH$_4^+$-N min$^{-1}$). In the same oxygenation levels interval, the N$_2$O emission rate increased for two of three inflow [NH$_4^+$] tested. It increased from 0.35 10$^{-3}$ to 0.73 10$^{-3}$ mg N min$^{-1}$ between 0 and 10.5 % O$_2$ at 25.3 mg NH$_4^+$-N L$^{-1}$, and from 1.34 10$^{-3}$ to 1.4 10$^{-3}$ mg N min$^{-1}$ between 4.2 and 10.5 % O$_2$ at 23.8 mg NH$_4^+$-N L$^{-1}$; while it decreased from 2.86 10$^{-3}$ to 2.04 10$^{-3}$ mg N min$^{-1}$ between 4.2 and 10.5 % O$_2$ at 37.3 mg NH$_4^+$-N L$^{-1}$. Finally, the N$_2$O emission factor globally increased from 0.05 to 0.16 % in the 0-10.5 % O2 interval.

Line 234: Why only 8 SP values are presented? But you have 13 data points for N2O concentration. Justification is required.

The referee is correct. The isotopic measurements were not conducted for all samples. However, the concentration data were all presented to capture the whole dynamic constrained by environmental

conditions tested. We added sentences in the method section 2.2 and in the caption of Fig. 6 for improved clarity.

Note that gas sampling was lacking for 5 of 13 oxygenation tests.

Note that the isotopic measurements of gas samples taken at inflow ammonium concentration of 42.7 and 42.9 mg N L-1 were averaged.

Line 237: Describe the processes rather than just the bacteria – if you mean nitrifierdenitrification and heterotrophic denitrificsation, mention this at the start

We mean nitrifier-denitrification and heterotrophic denitrification. We modified the sentence.

The $^{15}$N-SP values were close to the range of -11 to 0 ‰ reported for $N_2O$ produced by nitrifying or denitrifying bacteria through nitrifier-denitrification and heterotrophic denitrification (Toyoda et al., 2017; Yamazaki et al., 2014).

Line 239: But there was high SP value at O2 higher than 16.8%. If what you said were correct that nitrous oxide reduction was driving the high SP value at low O2 (4%) then what is driving the high SP at 21% O2?

We agree that an explanation about the high $^{15}$N-SP observed at 21 % $O_2$ is lacking. We added a sentence.

If an increase in the hydroxylamine oxidation contribution to the $N_2O$ emission might explain the higher $^{15}$N-SP observed at 21 % $O_2$ as compared to lower oxygenation levels, an additional mechanism can explain the variations observed for the experiments with oxygen-depleted atmosphere.

Line 249: What were the independence samples? How were they defined/sampled? How do you know that the N2O at different O2 condition were not from the same origin? Can't they be a mixture of N2O from different processes? And not quite sure what you meant by '… then partially reduced'.

Independence of samples means that the samples are not temporally linked to each other. The $N_2O$ sampled at 4.2 % $O_2$ is not a residual fraction of $N_2O$ produced at 16.8 % O2. We clarified these sentences.

The independence of samples taken during the oxygenation tests can explain this. The $N_2O$ sampled at 4.2 % $O_2$ is not a residual fraction of $N_2O$ produced at 16.8 % $O_2$ that would have undergone a partial reduction. The oxygenation level can alter the isotope fractionation factors through the control of reaction rates, as evidenced for the reduction of $N_2O$ to $N_2$ by Vieten et al. (2007). These authors reported lower reaction rates and increased isotope fractionation factors with increasing oxygenation levels. In our case, a similar phenomenon might have influenced both oxidative and reductive processes leading to the production of $N_2O$ and occurring before its ultimate reduction to $N_2$. However, knowledge, that the controls such as the oxygenation level have on the net isotope effect related to a sequence of non-exclusive oxidative and reductive processes is still lacking and require further investigations.

Line 261: Concentration of NOx increased for which experiment (temperature or NH4 concentration)?

Line 261: Why are these values different from the ones presented in Fig. 4c and 4d? Were you referring to the same thing?

We are not referring to Fig.4c and d here, but unplotted data of cumulated concentration of $NO_2^-$ and $NO_3^-$. We clarified this sentence.

In the ammonium tests, the cumulated concentrations of $NO_2^-$ and $NO_3^-$ ($[NO_x^-]$) increased from 1.4-6.1 to 5.1-19.6 mg N $L^{-1}$ between inflow and outflow (data not shown) and were composed by at least 74 and 91 % of $NO_3^-$, respectively.

Line 275: I agree that there was a stronger effect of temperature on N2O emission compared to NH4+ oxidation rate but for the effect of NH4 concentration, this could only be controlled by the very high NH4+ concentration – indicating a possible effect of NH4+ concentration on these points.

We agree that very high ammonium concentrations had an important effect, however, the highest $NH_4^+$ concentration is not the only one having an effect on the $N_2O$ emission factor. We can see a progressive increase in $N_2O$ emission factor with increasing $NH_4^+$ concentration from 6 to 58 mg N $l^{-1}$.

Line 276: Don't think the authors can draw conclusion on the N2O processes based solely on the NH4+ oxidation and N2O emission rates. Suggest discussing these processes after the discussion on the d15N values.

The reviewer is right, however no conclusions were drawn here. We discuss all the processes at play (hydroxylamine oxidation, nitrite reduction and $N_2O$ reduction) and suggest from $N_2O$ emission factor that increasing temperatures or ammonium concentrations unlikely increased $N_2O$ reduction to $N_2$. As suggested by the referee, the contributions of processes are discussed more in detail further, after presenting the isotope results (Lines 282-303).

Line 280: You should also calculate nitrite oxidation rate the same way you did for ammonium to support the contention you made here.

As part of the $NO_2^-$ and $NO_3^-$ produced from the oxidized ammonium could then react to other N compounds (NO, $N_2O$, $N_2$, $NO_3^-$, $NO_2^-$), we were not able to estimate the nitrite oxidation rate. The discussion about this rate is speculative from the remaining $NO_2^-$ concentration in the reactor outflow. We clarified this throughout the manuscript when we discussed about the nitrite oxidation process.

Line 294: Optimal here means? Why was 21% O2 considered optimal? Justifications/explanations are also required as to why 23.8 mg/L was chosen for the temperature experiment?

Please refer to a previous answer regarding the meaning of optimal $O_2$. We clarified this in the method section.

Further, we chose for the temperature tests an inflow ammonium concentration close to the nominal load that receives the nitrifying unit (0.7 kg N $m^{-3}$ $d^{-1}$). We added this precision in the method section.

Line 329: I assume you mean heterotrophic denitrification here? Why? You have just discussed that denitrifiers were sensitive to O2 and can be excluded as an important process contributing N2O at the high O2 conditions.

The referee is correct, we mean heterotrophic denitrification, here. The last step of heterotrophic denitrification (i.e. $N_2O$ reduction to $N_2$) is sensitive to $O_2$, however we cannot exclude that nitrite reduction performed by heterotrophic denitrifiers can occur.

This explains most likely the increased contribution of the nitrite reduction pathway to $N_2O$ emission, as more nitrite becomes available for nitrifier-denitrification and/or heterotrophic denitrification.

Figure 1: Very nice figure but references used for the ranges of the d15N d18O and SP should be included in the caption of the figure

We thank the reviewer for the suggestion and improvement of the figure, we added references related to d15N, d18O and 15N-SP ranges in the caption of the figure.

Figure 4: Include slope, p and r2 values for each of the plot. Not clear on how the errors on the NH4+ concentration plot were derived. And why only for NH4+ conc, did you see any variations for the temperature experiment as well? You have not mentioned elsewhere that replicates samples were collected. If this was how the error bars were derived state that in the method section.

The correlations and p values were mentioned in the results and method sections, respectively (3.3 and 2.4). We specified in the method section that replicates were sampled.

The feeding solutions were characterized from 1 to 5 replicates samples collected in the feeding tank. For each tested condition, the outflow was characterized within 5 days from 1 to 14 replicates samples immediately filtered through a 0.2 µm syringe filter and stored at 4 °C.

[Figure]

**Figure S1. Schematic of the nitrifying reactor used in this study. Note that solution was down-flowing, while air was up-flowing.**

[Figure]

**Equations:**

**Nitrification:**

$$\delta^{15}N-NH_4^+{}_{,res} \approx \delta^{15}N-NH_4^+{}_{,in} - \Delta_2(1 - f_1) + (\Delta_3 - \Delta_2)f_2$$

$$\delta^{15}N-NO_x^-{}_{,p} \approx \delta^{15}N-NH_4^+{}_{,in} + \Delta_2 f_1 + (\Delta_3 - \Delta_2)f_2$$

$$\delta^{15}N-NO_x^-{}_{,int} \approx \frac{(\delta^{15}N-NO_x^-{}_{,in} \times ([NO_3^-]_{in} + [NO_2^-]_{in}) + \delta^{15}N-NO_x^-{}_{,p} \times (1 - f_1 - f_2) \times [NH_4^+]_{in})}{([NO_3^-]_{in} + [NO_2^-]_{in} + (1 - f_1 - f_2) \times [NH_4^+]_{in})}$$

$$\delta^{15}N-N_2O \approx \delta^{15}N-NH_4^+{}_{,in} - \Delta_2(1 - f_1 + f_2) + \Delta_3(1 + f_2)$$

[Figure]

**Nitrifier-denitrification /**
**Heterotrophic denitrification:**

$$\delta^{15}N-NO_x^-{}_{,out} \approx \delta^{15}N-NO_x^-{}_{,int}(1 - f_1 - f_2)^{-1} - \Delta_5(1 - f_3)$$

$$\delta^{15}N-N_2O \approx \delta^{15}N-NO_x^-{}_{,int}(1 - f_1 - f_2)^{-1} + f_3\Delta_5$$

**Figure 2. Diagram and equations of the nitrifying reactor adapted from Fry (2006). It is considered as a sequence of two reactor boxes. (i) The nitrification of inflow ammonium ($NH_4^+{}_{,in}$) to a pool of nitrite and nitrate ($NO_x^-{}_{,p}$), residual ammonium ($NH_4^+{}_{,res}$), and nitrous oxide ($N_2O$) through the hydroxylamine oxidation pathway. (ii) The subsequent reduction of intermediate $NO_x^-{}_{,int}$; mixing of inflow $NO_x^-{}_{,in}$ and formed $NO_x^-{}_{,p}$ to nitrous oxide ($N_2O$) through the nitrite reduction pathway, and residual $NO_x^-$ that exits the reactor ($NO_x^-{}_{,out}$). Note that residual substrates and formed product exit the reactor without further isotope fractionation. See text for details.**

---

## Author Response (AR1)

We would like to thank the two referees and the associate editor for the effort and time they put in to review our manuscript. We are grateful for their valuable comments. We made every attempt to carefully address these comments that allowed us having completed a major revision of our manuscript. Hereafter, referee suggestions are written in bold font, whereas our responses are shown in italics. Excerpts of text and or references to line numbers are provided as appropriate.

**Response to RC1:**

**General comments**

Humbert et al. report emissions and production/consumption processes of N2O in a nitrifying biofilm reactor which simulates a part of a biological waste water treatment system. Although several similar studies have been published, knowledge of key factors that should be controlled to mitigate N2O emission is still insufficient because there are various type of biological waste water treatment and because processes related to N2O depend on many factors.

Major findings of this paper are that N2O is mainly produced by nitrifier-denitrification in a nitrifying biofilm reactor and that temperature control is more important than oxygen concentration or ammonia concentration. They may be worth publishing in Biogeosciences if the authors add implications of their research not only for a specific waste water treatment system but for other systems including natural water or soils.

Although the purpose and conclusion are clearly described, I found several flaws in the manuscript. First, a couple of related studies (Tumendelger et al., 2014; 2016) are not cited and compared with the findings of this study. Second, presentation of results (tables and figures in main text and supplement) is not well organized and is confusing. For example, Table S1 seems to show all the experimental conditions but corresponding results are not shown and figures seem to show only a part of the results. Third, a part of interpretation of isotopic data is not appropriate or based on assumptions that are not clearly explained. Fourth, several sentences are not readable or clear.

In summary, I consider this paper may be acceptable after careful revisions with respect to concerns above and below.

In the revised manuscript, we added implications of our research for a diversity of systems by adding sentences that underline these perspectives at some parts of the manuscript (Lines 23-25, 45-46, 87-88, 417-425).

Our responses to the specific comments of the referee are listed below and allow to address the four concerns lastly raised by the referee.

**Specific comments**

**L39 Add Tumendelger et al., 2014 and 2016.**

We agree that the results presented in Tumendelger et al., 2014 and 2016 fall perfectly within the scope of our study. Therefore, we added the references suggested by the referee and removed Tallec et al. 2006 from this reference list (Lines 41-42). "Temperature, electron donor and acceptor concentrations have been identified to control the N2O emission from WRRFs (Bollon et al., 2016; Kampschreur et al., 2009; Tumendelger et al., 2014, 2016; Wunderlin et al., 2012)."

L58–60 I think this statement is vague because equilibrium process is involved in biotic process (e.g., O-exchange between nitrate and water during nitrification and denitrification) and kinetic fractionation also occur in abiotic processes (e.g., diffusion in air or water).

*We reworded this sentence (Lines 61-62). "*The isotopic fractionation results from the difference in equilibrium constant or reaction rate observed between the heavier and lighter isotopes in both abiotic and biotic processes."

L73 This statement is misleading because many of previous studies cited elsewhere in this manuscript did use combination of isotope data of N2O to analyze production/consumption processes.

We agree and reworded this sentence (Line 76). "Nitrogen and oxygen isotope ratios of  $N_2O$  have lower potential for  $N_2O$  source identification as compared to 15N-SP."

L93–94 I think a schematic of the reactor helps readers to understand the experiment and how monitoring of environmental parameters and sample collection were conducted. What is "continuous down-flow counter current mode"?

As suggested by the referee, we added details in brackets and a schematic overview of the reactor as new Figure S1 in the supplementary material. This should improve the clarity of the system description (Lines 103-104). "(i.e. solution was down-flowing, while air was up-flowing; Fig. S1)"

L102 Here it can be read that the authors made 24 experiments, but in Table S1 total 26 conditions are shown. But in Table S1, the first line in the list of oxygenation tests and the second one for NH4+ concentration tests seem the exactly same condition, and the same for the second of oxygenation tests and the last of NH4 concentration tests. Are these pairs from actually a single experiment? Please explain in footnotes. Also, there is no "n.a." entry in Table S1 in spite of footnote.

Twenty-four experiments were performed. Two of them tested effects of both oxygenation and ammonium concentration. As suggested by the referee, we added explanations in the text and in the footnote of Table S1 (Lines 112-113). "Note that two of them were used twice; as oxygenation and concentration tests."

L106 If the numbers in Table S1 are correct, NH4 concentration should be "20.3" –21.1 mg N/L and temperature should be "19.0 to 19.8" C. Please check the data carefully.

We are grateful to the referee for raising this mistake. We rechecked our data and changed therefore the text in consistency with Table S1 which was correct (Lines 113-119).

L108 How did the authors determine that the optimal oxygen concentration is 21%? Table 1. This table is just showing reduced information presented in Table S1 and is not helpful. I suggest to use Table S1 here.

We reworded this sentence and added an explanation about the meaning of optimal oxygenation level for the next of the manuscript (Lines 119-123). "Atmospheric oxygenation level (i.e. 21 %  $O_2$  in the gas mixture) was imposed for both tests (Figs. S2b and c). This gas mixture using compressed air with 21 %  $O_2$  was considered hereafter as optimal as compared to the oxygen-depleted atmosphere used during the oxygenation tests. Noticeably, the atmospheric oxygenation level is the condition that represents the most optimal conditions of oxygenation applied in nitrification BAF of domestic WRRF."

Further, as suggested by the referee, Table S1 was used instead of Table 1 in the main text.

L131 This sentence seems to explain the calibration of dN and dO for ammonium, nitrite, and nitrate. In the case of N2O, dO cannot be calibrated using nitrate standards because there is a kinetic fractionation during N-O bond rapture in nitrate reduction to N2O. SP is also not determinable using the standards listed here. Please explain more. We clarified and reorganized this paragraph (Lines 144-158).

L140 Confirm the unit. If concentration is multiplied by flow rate, dimension should be mass per time (e.g., mg N /min).

The unit was corrected (Line 162). "(kg NH4+-N d-1)"

L147 Consider more appropriate title of the section, for example, "Estimation of the range of nitrogen isotope ratio in N2O produced by each biological process".

We clarified the title as suggested by the referee (Line 170). "2.5 Estimation of ranges of nitrogen isotope ratio in biologically produced  $N_2O''$

L157–171 It is strange that these sentences describe how to estimate dN values of output NH4+ and NOx-, because in Fig. S2 concentrations and isotope ratios of N compounds in inflow and outflow are shown as "measured" parameters. If these are really measured, I think it is worth calculating isotope enrichment factors (epsilons) in the studied system and comparing with previous studies.

The isotope ratios of N compounds in inflow and outflow were measured only for the three experiments shown in Fig.S3 (old Fig. S2). Although these experiments allow calculating isotope enrichment factors (epsilons), these epsilons unlikely describe the whole range of operating conditions tested in our study. Therefore, we think that using isotope ratio estimates from the literature is more appropriate than values based on a limited number of experiments. We added sentences throughout the manuscript (in 2.5 and 3.1 sections) to better explain this and hence to improve its clarity (Lines 175-177, 228-232). In addition, detailed data related to Figure S3 were added as new Table S2.

L171 It seems that produced NOx- (=NOxout - NOxin) is assumed to be derived from reacted ammonium. Then "f" in eq (4) should be "1-f".

We thank the referee for raising this mistake. We corrected Eq. (4) and modified subsequent information (text and figures) related to Eq. (4) throughout the manuscript (Line 200).

L172–176 These statements are not correct in a strict sense and are misleading. In a closed system, approximation of isotope effect using difference in delta values is valid only when isotope ratio in substrate can be treated as constant as described in Denk et al. (2017). In an open system, it is true that isotope effect can be expressed as the difference in delta values between product and residual substrate that exit from the system (Fry, 2006). But d15Ns in eq (5) can be read as isotope ratio in substrate before reaction (input) and hence is not applicable to open system. Equation (5) can be derived from an equation similar to eq (2) when f=1, but the authors do not state the assumption that f=1 is appropriate in this study. In fact, the value of f decreases as low as 0.2 (Fig. S1).

We revised this paragraph considering directly the open system used in this study, and introducing in details the assumptions made and additional calculations that allow describing our system more appropriately than before (Lines 180-213). Further, we specified that the notations used here are reversed as compared to those defined in Fry, 2006 while being similar to those presented in Denk et al. 2017 (Lines 183-184). In addition to this, we renamed  $NH_4^+$ ,out remaining in the reactor after nitrification as  $NH_4^+$ ,res; i.e. residual substrate, for better clarity. The  $NO_x^-$  pool resulting from mixing between the  $NO_x^-$  produced by ammonium oxidation and inflow  $NO_x^-$  was renamed as intermediate pool ( $NO_{x,int}$ ). Finally, a new Figure 2 has been introduced in the main text of our manuscript. This new figure presents the system used in this study and describes the corresponding equations.

**L182–183 Is "the ratio between ammonium oxidation rate and influent ammonium concentrations" different from "nitrification efficiency" (=oxidation rate/NH4 feeding rate) defined in L140–141? It is odd that ratio of parameters with different dimension is additionally introduced.**

We thank the referee for raising this mistake. We reworded the sentence consistently with parameters previously introduced. We are speaking about nitrification efficiency here (Lines 220-222). "Over the range of tested conditions, the ratio between ammonium oxidation rate and influent ammonium load ranged from 10 to 82 %, never exceeding 40 % for suboptimal nitrifying conditions imposed during oxygenation and temperature tests (i.e. oxygenation levels < 21 % O2, and temperatures < 20 °C)."

**L184–186 I cannot understand what this sentence means. Please rephrase.**

*We rephrased the sentence (Lines 222-224). "*The ammonium concentration, oxygenation level and temperature affected the ammonium oxidation rates, as well N2O emission rates and factors."

**L191–193 Here, the possible range of dN of outflowing NH4+ and NOx- is shown, but the dN for each timing is plotted as a single value in Fig. S2. How were these individual values calculated with what assumption?**

The standard error plotted here depending on the number of samples after one hydraulic retention time. We added precision in the method section 2.4 (Lines 165-166). "The measurements related to liquid or gas samples were averaged by experiment; i.e. average of data obtained from the samples collected after one hydraulic retention time."

L195–200 In multi-step reaction (in this case, two step reaction of NH3->NH2OH->N2O is considered), overall isotope effect does not necessarily equal to sum of the isotope effect of each step, but depends on substrate availability and ratio of backward to forward flows in the middle step of the reaction (Rees 1973). Please add basis of the authors's assumption.

Please refer to the previous answer regarding the estimates of  $\delta^{15}$ N-N2O ranges (Lines 180-213).

L203–204 I believe this is incorrect. The authors did not "observe" the net nitrogen isotope effect for each pathway, but they used literature values.

We rephrased the sentence (Lines 244-245). "A higher range of net nitrogen isotope effect for nitrite reduction than hydroxylamine oxidation pathway was estimated for N2O production (Figs. 3a and b)."

L224–227 I would like to see the mass balance of N. Judging from Table S2 (which shows results only from NH4+ concentration tests though), increase in NOx- is always lower than decrease in NH4+. Is apparent nitrogen loss explainable by gaseous emission of N2O and NO, or was there significant nitrogen assimilation by the biofilm?

We agree that the presentation of N mass balance would have improved our discussion. However, due to technical problems,  $NO_3^-$  analysis were lacking for some oxygenation tests and for the temperature tests, whereas  $NH_2OH$ ,  $N_2$  production, as well as N mineralization and assimilation in the biofilm were not quantified. Therefore, the amount of  $N_2O$  emitted alone does not explain the apparent nitrogen loss. No significant amounts of NO were detected during the tests (< 1 ppm measured in the outlet gas), and the accumulation of  $NH_2OH$  is unlikely. Heterotrophic denitrification, i.e. the reduction of  $NO_x^-$  to  $N_2$ , most likely adds up the N mass balance. However,  $N_2$  measurements (in the gas mixture comprising of at least 79 %  $N_2$ ) were not been feasible and thus not carried out. We added precisions about this in the manuscript (Lines 269-277).

L228–231 I cannot agree that ammonium oxidation rates were "low and stable" for 0–10.5% O2 because two high values were observed at 5% (Fig. 3a). It is not clear whether the authors excluded the two data (because of large error bars?) or not. I understand that these rates are calculated from influent and effluent NH4+ concentrations measured over time as presented in Fig. S2 (again, this figure only shows results from NH4+ concentration tests though), but cannot understand why the error bar ("standard deviation") for the two data is significantly large. Please explain it as well as detailed procedure for calculating "average and standard deviation" (e.g., how many measured data were used for averaging?).

We improved clarity of this sentence and precise according to the referee's comment that the average presented here was calculated from all data point in the considered interval (i.e. 0-10.5 %  $O_2$  in the gas mixture) (Lines 278-283).

*Further, the standard error plotted here depending on the number of gas sampled after one hydraulic retention time. We added precision in the method section 2.4 (Lines 165-166). "*The measurements related to liquid or gas samples were averaged by experiment; i.e. average of data obtained from the samples collected after one hydraulic retention time.".

**L234–235 I see 8 data points in Fig. 3d (also 3e and 3f), but Table S1 indicates total 13 data were obtained for oxygenation tests. Does this mean isotopic measurements were not conducted for all samples?**

The referee is correct. The isotopic measurements were not conducted for all samples. However, the concentration data were all presented to capture the whole dynamic constrained by environmental conditions tested. For improved clarity, we added sentences in the method section 2.2 (Lines 132-133) and in the captions of Fig. 4 "Note that gas sampling was lacking for 5 of the 13 oxygenation tests" and Fig. 6 "Note that the isotopic measurements of gas samples taken at inflow ammonium concentration of 42.7 and 42.9 mg N L-1 were both recorded as 42.8 mg N L-1 in the legend".

**L235–236 I cannot see "similar marked change" in d15N at 16.8% O2 and 21% O2. The two data points for each O2 condition depart each other, and when average is taken, there would be no significant difference.**

We understand the concern of the referee, however, it should be noticed that ammonium concentration is also modified between 16.8 and 21 % O2. This additional effect of ammonium mitigates the effect of oxygenation alone that is intended to be shown here. Therefore, the average isotope ratios at 21 % O2 should be compared to the data solely measured for 23.8 NH4+-N L-1 at 16.8 % O2. We specified it more clearly in the manuscript (Lines 288-291). "A similar marked change in nitrogen and oxygen isotope ratios of N2O (decrease and increase, respectively) was observed when oxygenation increased from 16.8 to 21 % O2 (Figs. 4e and f). Note that to observe the latter variations the effect of ammonium concentration was not included. One way to do so is to compare the isotope composition average at 21% O2 with the isotope composition measured for 23.8 NH4+-N L-1 at 16.8 % O2."

L238–239 Relatively higher SP value was observed not only at 4.2% O2 but also at 20% O2. But do the authors consider N2O reduction occurs at 4.2% O2 just because SP is larger than the range estimated for N2O produced NO2- reduction? It seems to me that the two high SP data is not significantly different considering the large error bars for 20% O2, and that the upper end of estimated range might be underestimate (see Fig. 6 in Denk et al.).

We agree with the referee. We did suggest that  $N_2O$  reduction to  $N_2$  likely occurs for oxygen-depleted air due to the anti-correlation observed between the oxygen-depletion and 15N-SP for oxygenation

levels between 4.2 and 16.8 %  $O_2$ , not due to 15N-SP at 4.2 %  $O_2$  higher than the range of 15N-SP proposed for the N2O derived from nitrite reduction.

L244–245 Alternative explanations can be made for the decrease in N2O emission. For example, change in branching ratio between NO2-, N2O, and NO production during NH2OH oxidation might reduce N2O emission. Are there any other evidences for N2O reduction? I agree that N2O reduction might occur when oxygen concentration is really 0%, but as shown in Fig. S1, measured DO is ca. 1.5 mg/L and this enabled NH4 oxidation. It is unlikely that NH4 oxidation and N2O reduction occur at the same time unless there are specific anoxic sites in the system.

Our point here is to suggest the likely occurrence of anoxic microsites within the reactor biomass, due to heterogeneous and varying distribution of air circulation. Currently, two evidences are presented: (i) the increasing 15N-SP with the decreasing oxygenation levels; and (ii) the decrease in  $N_2O$  emission. This could explain the co-occurrence of  $NH_4^+$  oxidation and  $N_2O$  reduction. Further, the DO presented in Fig. S1 is unfortunately an undetailed picture of what happened within the reactor and more particularly within the colonized media. We were not able to measure the DO within the media and DO was therefore measured at the top of the reactor, where it is likely higher than within the colonized media. This is attested by a few measurements performed in the outflow of the reactor during concentration tests. We observed DO 2.8-3.9 mg  $O_2 L^{-1}$  lower in the outflow than at the top of the reactor. We added a reference and an explanation describing the likely co-occurrence of ammonium oxidation and  $N_2O$ reduction (Lines 301-305). "This is also consistent with a possible onset of anoxic microsites within the reactor biomass more likely at 4.2 than 16.8 % O2. The dissolved oxygen (DO) concentration never decreased below 1.5 mg  $O_2$  L-1 in the bulk solution at the top of the reactor (Fig. S2). However, DO decreased from the bulk reactor solution toward the deeper layers of biofilm due to the activity of ammonium oxidizers (Sabba et al., 2018). This is further exacerbated by heterogeneous and varying distribution of air circulation within the static bed. Therefore, oxygen depletion can be assumed within the biofilm."

**L249 What does "independence of samples" mean?**

Please see below.

**L249–251 I cannot follow these sentences. Please rephrase and describe why the different trends of d15N and d18O can be explained with reaction rates in more detail.**

Independence of samples means that the samples are not temporally linked to each other. The  $N_2O$  sampled at 4.2 %  $O_2$  is not a residual fraction of  $N_2O$  produced at 16.8 % O2. We clarified these sentences (Lines 310-317).

L259–261 Although NH4+ oxidation rate has linear relation to NH4+ concentration (Fig. 4b), the remaining NH4+ fraction does not (Fig. S1f). It seems to increase nonlinearly. Please discuss why this happened. In addition, I cannot see that remaining fraction of NH4 or NH4+ oxidation rate is "negatively correlated to temperature" in Fig. S1e or Fig. 4a. It seems almost constant irrespective of temperature. Is stated correlation really significant? Please show p values.

We think that the discussion about the difference between ammonium oxidation rate and remaining ammonium fraction in relationship with the ammonium concentration (i.e. linear vs. nonlinear) is out of the scope of our study. However, the nonlinear increase that describes the relationship between the remaining ammonium fraction and the ammonium concentration, and highlighted here by the referee is mainly due to the data recorded at the lowest ammonium concentration (6.2 mg N L-1). The low remaining ammonium fraction measured at this inflow  $NH_4^+$  concentration can be explained by oxygen

limitation being lower at 6.2 mg N L-1 than at higher ammonium concentration. Indeed, the decrease in DO toward the deeper layers of biofilm due to the activity of ammonium oxidizers is likely less important at 6.2 mg N L-1 than at higher ammonium concentration. A way to check this would have been to measure the isotope composition of N2O emitted at 6.2 mg N L-1 inflow [NH4+]; i.e. the 15N-SP is expected much higher at 6.2 mg N L-1 due to increase contribution of hydroxylamine oxidation pathway in N2O production. Unfortunately, the N2O emission was very low at this inflow ammonium concentration and N2O isotope composition was not analyzed.

Further, the negative correlation between the remaining ammonium fraction and temperature exists and is significant with p < 0.05 (please see statement about the p values used in our study lines 167-168 in the method section). In addition, please note that the y-axis scale of Fig. S2e is the same as Fig. S2d and f. We applied the same y-axis scale to be able to compare with each other the effects of the tested conditions had on the remaining ammonium fraction. Consequently, the variations are less visible in response to temperature than ammonium concentration. However, the negative correlation does exist.

L264–267 Although I think there is no significant relationship between NH4+ oxidation rate or remaining fraction and temperature, the authors argue that NH4+ remaining fraction is negatively correlated with temperature whereas NH4+ oxidation rate is positively correlated with temperature. Please explain why this apparently contradict trend was observed.

It is not surprising that  $NH_4^+$  remaining fraction was negatively correlated to  $NH_4^+$  oxidation rate in the temperature tests. At constant inflow ammonium concentration and hydraulic retention time, the increase in the ammonium oxidation rate decreases the amount of residual ammonium that exits the reactor. Consequently, the ammonium remaining fraction decreases.

**L267–268 Temperature effect (if any) might be explained with enzymatic activity, but I think NH4+ concentration effect can be explained with kinetics of enzymatic reaction like Michaelis-Menten kinetics.**

We agree with the referee and clarified the text (Lines 331-333). "These positive correlations are well known in the temperature range investigated here and are likely due to enhanced enzymatic activity and Michaelis-Menten kinetics, respectively (Groeneweg et al., 1994; Kim et al., 2008; Raimonet et al., 2017)."

**L274 Based on which parameter do the authors find "stronger" effect of temperature and NH4+ concentration on the N2O emission rate than on NH4+ oxidation"? For example, slopes in Fig. 4b and 4d look similar.**

The emission factor is calculated as the ratio between  $N_2O$  emission and  $NH_4^+$  oxidation rates. Therefore, the  $N_2O$  emission factor enables to identify when the  $N_2O$  emission is more responsive than the  $NH_4^+$  oxidation to the ammonium inflow. Furthermore, note that the slopes in Figs. 5b and d cannot be compared with each other, as the y-axis scales are different.

**L278–281 Please explain in detail why the authors consider nitrite oxidation (to nitrate) is less important than nitrite reduction (to N2O) in this case.**

Given the mass balance of the N compounds, the oxidation of nitrite to nitrate remains the main pathway consuming nitrite. However, an increasing  $N_2O$  emission factor results from increase in ammonium oxidation rate driven by temperature or  $NH_4^+$  concentration. Therefore, the increase in reduction of  $NO_2^-$  to  $N_2O$  seems to be a bit more pronounced than the increase in oxidation of nitrite to nitrate. We added precisions in the manuscript (Lines 343-346). "Furthermore, no nitrite accumulation was observed with increasing ammonium oxidation rate (Fig. S2i). Therefore, if  $N_2O$  emission results mainly from the nitrite reduction pathway, this suggests that the nitrite reduction pathway is more responsive to the increasing ammonium oxidation rate than the nitrite oxidation pathway; the latter remaining the main pathway of nitrite consumption."

**L284–285 In Tumendelger et al. (2014, 2016), larger SP values were reported under aerobic condition.**

We added references to works performed by Tumendelger et al (Lines 349-353). "This is consistent with previous findings based on the 15N-SP of N2O emitted from aerobic activated sludge (Toyoda et al., 2011; Tumendelger et al., 2016; Wunderlin et al., 2013). Authors reported 15N-SP as high as 10 ‰. This can suggest a higher oxygen limitation being favourable to the contribution of the nitrite reduction to N2O production in the nitrifying reactor studied here. The hydroxylamine oxidation can even be the main N2O producing pathway, as evidenced by Tumendelger et al. (2014) in some aerated tank."

L285–286 Ambiguous sentence. Do the authors intend to argue that SP value increases with temperature (13.5C<T<19.5C) and that it also increases with NH4+ concentration when T is set around 19C? Please rephrase. I cannot agree with the latter statement because SP value obtained at 42.8 mg N/L is lower than SP at 28.6 (Fig. 5a).

We rephrased this section for improved clarity (Lines 354-362).

Figure 5 caption. As I pointed out at L228–231 above, it is not clear how the authors made data reduction based on primary data. How "average and standard deviation" were calculated? How did the authors ensure "the steady-state conditions"?

*Please see our previous answers. Furthermore, we rephrased the Figure 6 caption. "*Average and standard deviation (error bars) are calculated for the samples taken after one hydraulic retention time."

L315–317 Does "stable NO2-" mean that NO2- concentration was constant over time or that it did not depend on NH4+ concentration? If the authors intend to mean the latter, I cannot agree with them because three data points at 20% O2 in Fig. S1g show a large variation.

We mean that the NO2- concentration did not depend on oxygenation levels. We modified this sentence (Lines 386-388). "No relationship was observed between NO2- concentrations and oxygenation (Fig. S2g). In addition to this, higher 15N-SP at 21 % compared to the 10.5-16.8 % O2 was observed while temperature remained below 20 °C (Fig. 4d)."

**L323–324 Sect. 3.2 -> Sect. 3.3? There, the authors wrote temperature range as 13.5–19.8°C, and did not describe "exponentially increase". It seems to me that SP increases with temperature linearly.**

We agree with the referee. We rephrased the sentence (Lines 394-396). "During the temperature and ammonium concentration tests, the contribution of the hydroxylamine oxidation pathway to N2O emissions increased with temperature between 13.5 and 19.8 °C (Sect. 3.3) and decreased in favor of the nitrite reduction pathway when temperature exceeded 20 °C (Fig. 6a)."

**Technical corrections**

L44 a large "extent"

This change was made.

**L67 enriched in "15N at" central position**

This change was made.

**L81 Add question mark at the end of the sentence.**

This change was made.

**L103–105 Awkward sentence. Consider other expression than "consisted of".**

We modified our sentences (Lines 113-119).

L230 and elsewhere. Insert "x" between significant (i.e. 0.35) and exponent (i.e. 10-3)

This change was made.

Figure 3 legend. Open circle represents NH4 concentration of 25.1, not 25.3 if Table S1 is correct.

This change was made.

L238–239 Rephrase the subject part ("The 15N-SP ... levels") of this sentence. A higher amount of N2O "was" reduced to N2.

We rephrased this section (Lines 293-297). "Additional suggestions can be made from the 15N-SP dynamics between and variations within the oxygenation levels. If an increase in the hydroxylamine oxidation contribution to the N2O emission might explain the higher 15N-SP observed at 21 % O2 as compared to lower oxygenation levels, an additional mechanism can explain the variations observed for the experiments with oxygen-depleted atmosphere. The 15N-SP dynamics suggest a higher amount of N2O was reduced to N2 at 4.2 than 16.8 % O2."

**L273 correlation between ... and the ammonium oxidation rates (delete "to")**

This change was made.

**L333 a larger "extent"**

This change was made.

**Response to RC2:**

The authors report the use of stable isotopes of N2O (bulk and site specific d15N), complementary to N2O and dissolved inorganic nitrogen concentrations to identify the key processes producing N2O in a biofilm reactor used in a local wastewater treatment facility. They showed that nitrite reduction was the primary N2O producing pathway in the reactor irrespective of the experimental conditions (i.e. different percentage of O2, temperature and initial NH4+ concentrations). Temperature, however imposed the greatest effect on N2O emissions compared to the other factors by simultaneously promoting hydroxylamine oxidation pathway.

This study contains interesting dataset particularly on the factors controlling N2O emissions; which could have broader implications to other systems. As such, I think this study has the potential to be an interesting and helpful addition to the literature but to make it so will require a concerted effort. This is because the manuscript is not very well-written. Most part of the manuscript is confusing with either no or invalid justifications were provided for the assumptions made.

For example, (1) experimental conditions presented in the tables are different from the ones presented in the graphs but no explanation was provided as to why some of the data points were ignored;

The isotopic measurements were not conducted for all samples. However, the concentration data were all presented to capture the whole dynamic constrained by environmental conditions tested. For improved clarity, we added sentences in the method section 2.2 (Lines 132-133) "Note that gas sampling was lacking for 5 of the 13 oxygenation tests." and in the caption of Fig. 6 "Note that the isotopic measurements of gas samples taken at inflow ammonium concentration of 42.7 and 42.9 mg N L-1 were both recorded as 42.8 mg N L-1 in the legend.".

**(2) some of the interpretations on the trends are misleading and were not supported with statistical analysis;**

Please see our specific answers below. When lacking, we added statistics to support our statement in the results section.

(3) rates of processes were not well-defined and some of the terms were randomly introduced in the discussion without prior definition of the terms;

We carefully defined the processes rates in the Methods section in consistence with those used further in the manuscript.

(4) there was no clear distinction on which part of the results were depicted from the literature and which part was obtained from the study;

We carefully checked our manuscript and used appropriate terms throughout the manuscript to better specify where data came from.

(5) in the method the authors mentioned that they analysed the d15N of nitrate, nitrite and ammonium, they then indicated in the later section that they hypothesized/estimated the values from the proposed equations.

The isotope ratios of N compounds in inflow and outflow were measured only for the three experiments shown in Fig.S3 (old Fig. S2). Although these experiments allow calculating isotope enrichment factors (epsilons), these epsilons unlikely describe the whole range of operating conditions tested in our study. Therefore, we think that using isotope ratio estimates from the literature is more appropriate than values based on a limited number of experiments. We added sentences throughout the manuscript (in 2.5 and 3.1 sections) to better explain this and hence to improve its clarity (Lines 175-177, 228-232). In addition, detailed data related to Figure S3 were added as new Table S2.

**Specific comments:**

**Line 16: The authors argued in the text that nitrifier-denitrification was the main N2O producing pathway, remove heterotrophic denitrification if this is true**

The 15N-SP does not allow differentiating between nitrifier-denitrification and heterotrophic denitrification contribution, both involving the nitrite reduction pathway that produces  $N_2O$ . We checked our manuscript, nitrifier-denitrification was not indicated as the main  $N_2O$  producing pathway. We are not sure having correctly understood the remark of the reviewer.

**Line 17: Method/procedure to estimate nitrite oxidation rate was not discussed/mentioned. Not clear what you mean here. Consider revising the sentence.**

Please see below.

Line 18: State the sub-optimal condition.

We agree that nitrite oxidation rate was not calculated. However, we inferred lower nitrite oxidation than ammonium oxidation rate from lower  $NO_x^-$  than oxidized ammonium budget. We clarified this throughout the manuscript (Lines 17-18, 343-346, 386-390).

In addition to this, we changed sub-optimal oxygen levels to oxygen-depleted atmosphere (Line 19). "Difference between oxidative and reductive rates of nitrite consumption was discussed in relation to  $NO_2^-$  concentrations and  $N_2O$  emissions. Hence, nitrite oxidation rates seem to decrease as compared to ammonium oxidation rates at temperatures above 20 °C and under oxygen-depleted atmosphere, increasing  $N_2O$  production by the nitrite reduction pathway."

**Line 19: You mentioned that heterotrophic denitrification could be present, if so, how do you know the N2O was produced from NH4+ not from other substrates given that the inflow also comprised of NO3-?**

We hypothesized here that the  $NO_2^-$  produced from the oxidized  $NH_4^+$  can be further reduced to  $N_2O$  either by nitrifier-denitrification or heterotrophic denitrification. However, the 15N-SP of  $N_2O$  does not allow differentiating between the contributions from both processes. In addition to this, the range of  $\delta^{15}N-N_2O$  estimates takes into account the isotope composition of inflow  $NO_2^-$  and  $NO_3^-$ .

**Line 28: Is there a more recent estimate for N2O emission? WMO?**

We did not find more recent similar data in the WMO database (i.e. fraction of  $N_2O$  emitted from wastewater resource recovery facilities), and we do not think that it has changed by order of magnitude during the last 5 years. For this reason, the initial statement was not modified.

Line 80: Biofilm reactor is only introduced here and no other explanation on its importance. Perhaps a sentence or two should be included to emphasize on the importance of these reactors (e.g. are these reactors commonly used in waste water treatment plant and how the efficiency of the reactors affect N2O emissions, why only nitrifying reactor is considered).

We agree with the reviewer, we added sentences to better justify why we worked on a biofilm (Lines 82-87). "Among the wastewater treatment systems, the biofilm systems are adapted to large urban areas owing to their compactness, flexibility and reliability. An increase of their development is expected in response to the additional 2.5 billion humans predicted in urban areas by 2050 (United Nations Population Division, 2018). However, the biofilm systems have received much less attention than the suspended biomass systems and the relations between the N2O producing/consuming pathways and controls remain largely unknown (Sabba et al., 2018; Todt and Dörsch, 2016)."

**Line 94: What do you mean by down-flow counter-current mode? More explanation is required especially for non-expert readers.**

It means that feeding solution flows through the reactor from the top to the bottom, while aeration being injected from the bottom of the reactor. We added details in brackets and a schematic overview of the reactor as new Figure S1 in the supplementary material. This should improve the clarity of the system description (Lines 103-104). "(i.e. solution was down-flowing, while air was up-flowing; Fig. S1)"

**Line 98: Is the feeding solution described here the same as your inflow solution? If yes, why the inflow solution comprised of other DIN species not only NH4+ as described. As written, the biofilm is only fed with NH4+ not NO3- and NO2- so where did these species originate from?**

As we used tap water to prepare the feeding solution, containing NO2- and NO3-. We added in brackets the average NO2- and NO3- concentration in the tap water (Lines 109-110). "(average 0.2 ± 0.4, 2.4 ± 1.1, and 2.5 ± 1.3 mg N L-1 of NO2-, NO3- and sum of both NOx- molecules, respectively)"

**Line 102: 24 or 26? There was a total of 26 experimental conditions listed in Table S1**

Twenty-four experiments were performed. Two of them tested effects of both oxygenation and ammonium concentration. We added explanations in the text and in the footnote of new Table 1 (old Table S1) (Lines 111-113). "The influence of environmental conditions on the ammonium oxidation rates and the N2O emissions from various combinations of oxygenation levels, temperatures and ammonium concentrations were tested in twenty-four experiments (Table 1). Note that two of them were used twice; as oxygenation and concentration tests."

**Line 104: I suggest the authors consider removing the first two conditions for the O2 test because the different NH4+ concentrations could be compromising the effect of dissolved O2 on the N2O production. Remove from graphs as well if these data points were included in the graphs.**

The isotopic measurements were not conducted for all samples (please see above). However, concentration data were all presented to capture the whole dynamic constrained by oxygen, temperature and ammonia conditions. Noticeably, if we remove the first two conditions of the oxygenation tests as suggested by the referee, we will not have any isotope data at 21%  $O_2$ . Therefore, we kept these first two conditions. However, we agree with the referee that the additional effect of ammonium can mitigate the effect of oxygenation alone that is intended to be shown here. Therefore, the average isotope ratios at 21 %  $O_2$  should be compared to the data solely measured for 23.8 NH4+-N L-1 at 16.8 %  $O_2$ . We specified it more clearly in the manuscript where these results are described (Lines 288-291). "A similar marked change in nitrogen and oxygen isotope ratios of N2O (decrease and increase, respectively) was observed when oxygenation increased from 16.8 to 21 %  $O_2$  (Figs. 4e and f). Note that to observe the latter variations the effect of ammonium concentration was not included. One way to do so is to compare the isotope composition average at 21%  $O_2$  with the isotope composition measured for 23.8 NH4+-N L-1 at 16.8 %  $O_2$ ."

In addition to this, we reorganized the Table 1 to sort the tests by increasing tested conditions.

**Line 108: What is the optimal DO level and how was this determined?**

We reworded this sentence and added an explanation about the meaning of optimal oxygenation level for the next of the manuscript (Lines 119-123). "Atmospheric oxygenation level (i.e. 21 % O2 in the gas mixture) was imposed for both tests (Figs. S2b and c). This gas mixture using compressed air with 21 % O2 was considered hereafter as optimal as compared to the oxygen-depleted atmosphere used during the oxygenation tests. Noticeably, the atmospheric oxygenation level is the condition that represents the most optimal conditions of oxygenation applied in nitrification BAF of domestic WRRF."

**Line 110: Check the numbers and cross check with Table S1. Some of the values are different**

We are grateful to the referee for raising this mistake. We rechecked our data and changed therefore the text and table in consistency with Table S1. Furthermore, Table S1 was used instead of Table 1 in the main text (Lines 113-119).

**Line 129: What were the protocols and standards for determination of d15N, d18O and d15N-SP of gaseous N2O? At present, this part of the method is missing.**

We clarified and reorganized this paragraph (Lines 144-158).

**Line 140: check the units between AOR and N2O-ER. The reported units are not consistent**

The unit was corrected (Line 162).

Line 140: the authors did not seem to discuss on nitrification efficiency throughout the manuscript, please remove this from the method if this is not needed to avoid confusion. Instead consider including the calculation for nitrite oxidation rates as this was briefly mentioned in the abstract and at some stage of the manuscript.

We did not calculate the nitrite oxidation rate. As part of the NO2- and NO3- produced from the oxidized ammonium could then react to other N compounds (NO, N2O, N2, NO3-, NO2-), we were not able to estimate the nitrite oxidation rate. The discussion about this rate is speculative from the remaining NO2- concentration in the reactor outflow. We clarified this throughout the manuscript (Lines 17-18, 343-346, 386-390).

Further, we kept the introduction of nitrification efficiency as we use it at the beginning of the results and discussion section. However, we reworded the sentence consistently with parameters introduced in the method section (Lines 220-222). "Over the range of tested conditions, the ratio between ammonium oxidation rate and influent ammonium load ranged from 10 to 82 %, never exceeding 40 % for suboptimal nitrifying conditions imposed during oxygenation and temperature tests (i.e. oxygenation levels < 21 %  $O_2$ , and temperatures < 20 °C)."

**Line 181: Why was the text being placed in supplementary section? These sentences should be moved to the main section.**

As suggested by the referee, we moved these sentences to the main text (Lines 216-220). "During the ammonium concentration tests, decreases in ammonium concentrations ( $[NH_4^+]$ ), increases in nitrite and nitrate concentrations ( $[NO_2^-]$  and  $[NO_3^-]$ , respectively) were observed, while pH remaining below 8 prevented any relevant loss of ammonium by volatilization. For example,  $[NH_4^+]$  decreased from 6.2 to 1.1, from 28.6 to 17 and from 62.1 to 49.1 mg N L-1 by flowing through the nitrifying biomass. At the same time,  $[NO_2^-]$  and  $[NO_3^-]$  increased from 0 to 0.2-0.3 mg N L-1 and from 1.4-1.8 to 5-10 mg N L-1, respectively."

**Line 183: What is the suboptimal condition? Please define and explain how this condition was obtained and on what basis this condition was considered suboptimal.**

We specified what we mean by suboptimal conditions in brackets (Line 222). "(i.e. oxygenation levels  $< 21 \% O_2$ , and temperatures  $< 20 \degree$ C)"

**Line 183 – 186: Not quite sure what you meant here. Please rephrase.**

We rephrased the sentence (Lines 222-224). "The ammonium concentration, oxygenation level and temperature affected the ammonium oxidation rates, as well N2O emission rates and factors."

Line 191: The d15N values of the outflowing NH4+ and NOx were estimated from equation? Were they not measured using the same method as the d15N of the inflowing NH4 and NOx? If you did measure the outflowing d15N of NH4 and NOx how did that compare to the ones estimated using the equations? Where did you get the "value from and why only 0 and 1 were used for f, given f should represent the fraction of NH4+ or NOx remained in the reactor.

The isotope ratios of N compounds in inflow and outflow were measured only for the three experiments shown in Fig.S3. Although these experiments allow calculating isotope enrichment factors (epsilons), these epsilons unlikely describe the whole range of operating conditions tested in our study. Therefore, we think that using isotope ratio estimates from the literature is more appropriate than values based on a limited number of experiments. We added text throughout the manuscript (in 2.5 and 3.1 sections)

to better explain this and hence to improve its clarity (Lines 175-177, 228-232). In addition, detailed data related to Figure S3 were added as new Table S2.

Line 195: I do not see the importance of discussing the net isotope effect of the overall ammonium oxidation here given the main focus of the discussion point here is the importance of hydroxylamine oxidation versus nitrite reduction. Furthermore, I doubt the validity of the assumption made by the authors in estimating the overall net isotope effect of ammonium oxidation to nitrous oxide. The net isotope effect relies heavily on the initial d15N and the availability of the substrate and do not necessarily associate with the total " from different part of the processes. Even if the d15N of the substrate is the same, different bacteria culture or organisms tend to generate different fractionation effects. Furthermore, the values cited by the authors especially for the net isotope effects related to hydroxylamine oxidation to N2O were not found in the cited references! Please recheck. It is perhaps more interesting to look at the separate effect of the two processes (i.e. ammonium oxidation to nitrite and hydroxylamine oxidation to N2O) on the overall processing of NH4+ in the reactor. You should have enough data to estimate the net isotope effect of ammonium oxidation to nitrite and discuss how that compares to the literature value?

Please see our previous answer regarding the net isotope effect estimate of ammonium oxidation to nitrite. Further, we did not added hydroxylamine directly into the feeding solution. Ammonium was added into the feeding solution. However, the production of N2O from hydroxylamine oxidation during the oxidation of ammonium cannot be excluded. Therefore, we estimated the range of  $\delta^{15}N-N_2O$  produced from hydroxylamine oxidation with ammonium as substrate. Thanks to existing literature, our method considered the broad range of values taken by both the  $\delta^{15}N$  of hydroxylamine produced during ammonium oxidation and the fractionation effects related to both ammonium oxidation to hydroxylamine oxidation to N2O. Further, we rechecked the values used from cited references. Correction involved minor changes (<1 ‰) (Lines 234-243).

Line 201: Which method are you referring to? And on what basis that the authors think that d15N-N2O values here refer specifically to hydroxylamine oxidation? Were these d15-N2O values obtained from all the experiments? Or from a specific experiment (ammonium concentration or temperature or DO)?

In this paragraph, a general way to estimate ranges of nitrogen isotope ratio in  $N_2O$  is proposed. This relies on the method presented in 2.5 and the use in Eqs. (5) and (6) of fractionation estimates obtained from the literature. As stated above, we think that using isotope ratio estimates from the literature is more appropriate than measured values based on a limited number of experiments. Published isotope effects were then used. We assume that the range of published isotope effects enables taking into account a high diversity of environmental conditions and bacterial community involved in these isotope effects.

Line 203: Did the authors observe the net isotope effects or the values were depicted from previous studies? Be more specific. If they did observe the net isotope effect in this study then why not just use these values in the rest of the discussion?

*We rephrased the sentence (Lines 244-245). "*A higher range of net nitrogen isotope effect for nitrite reduction than hydroxylamine oxidation pathway was estimated for N2O production (Figs. 3a and b)."

Line 206: And again, where did you get these values from? Are the values in brackets represent the averages of the d15N of respective analytes? Please specify

These values come from the net isotope effect found in literature and mentioned in the introduction section (Lines 66-68). However, we agree that a reminder is required here (Lines 245-250).

**Line 207-209: I don't think I quite get what you mean here. Explanation is needed on how ammonium oxidation influences the denitrifiers and why is that relevant to the d15N-N2O derived from nitrite being similar for both bacteria? What if NO3 not NO2 was used as a substrate for denitrifier?**

We mean that ammonium is oxidized to  $NO_2^-$  and  $NO_3^-$  in the nitrifying reactor. Both  $NO_2^-$  and  $NO_3^-$  can be reduced through either nitrifier-denitrification or heterotrophic denitrification or a combination of both reductive processes. However, the isotope composition of the substrate ( $NO_2^-$  and  $NO_3^-$ ) is imposed by the prior oxidation of ammonium to  $NO_2^-/NO_3^-$ , whatever the reductive process at play (nitrifierdenitrification or heterotrophic denitrification) (Lines 252-254). "However, a similar range of nitritederived  $\delta^{15}$ N-N2O is suggested for nitrifiers and heterotrophic denitrifiers, because ammonium oxidation influences both processes in the system used in this study where there is low initial amount of  $NO_2^-$  and  $NO_3^-$ ."

**Line 229: Not entirely true because some of the rates were high.**

We improved clarity of this sentence (Lines 278-283).

Line 230: To me, there was no increase in N2O emission for the same NH4 concentration for different DO condition. I think the authors should carefully consider the trend by comparing the data points for the same NH4 concentration. Can you please include the slope values of the trend line so that it is easier to compare if there was an increase in the tested parameters.

We agree that this section needs precisions. Note that we applied the same y-axis scale to be able to compare with each other the effects of the tested conditions had on the N2O emissions (Figs. 4b, 5c and 5d). Consequently, the variations are less visible in response to oxygenation and temperature than ammonium concentration. Further, we do not want to apply a trend line to the whole dataset that includes different inflow ammonium concentrations. In addition to this, there is a limited number of data related to each inflow ammonium concentration in the interval 0-10.5 %  $O_2$  (1 to 3 data point). Therefore, the application of a trend line to this limited number of data presents a limited interest. We described the results with more precision (Lines 278-286).

**Line 234: Why only 8 SP values are presented? But you have 13 data points for N2O concentration. Justification is required.**

The referee is correct. The isotopic measurements were not conducted for all samples. However, the concentration data were all presented to capture the whole dynamic constrained by environmental conditions tested. We added sentences in the method section 2.2 (Lines 132-133) and in the caption of Figs. 4 and 6 for improved clarity.

**Line 237: Describe the processes rather than just the bacteria – if you mean nitrifierdenitrification and heterotrophic denitrificsation, mention this at the start**

We mean nitrifier-denitrification and heterotrophic denitrification. We modified the sentence (Lines 291-293). "The 15N-SP values were close to the range of -11 to 0 ‰ reported for N2O produced by nitrifying or denitrifying bacteria through nitrifier-denitrification and heterotrophic denitrification (Toyoda et al., 2017; Yamazaki et al., 2014)."

Line 239: But there was high SP value at O2 higher than 16.8%. If what you said were correct that nitrous oxide reduction was driving the high SP value at low O2 (4%) then what is driving the high SP at 21% O2?

We agree that an explanation about the high 15N-SP observed at 21 % O2 is lacking. We added a sentence (Lines 294-297). "If an increase in the hydroxylamine oxidation contribution to the N2O emission might explain the higher 15N-SP observed at 21 % O2 as compared to lower oxygenation levels, an additional mechanism can explain the variations observed for the experiments with oxygen-depleted atmosphere."

Line 249: What were the independence samples? How were they defined/sampled? How do you know that the N2O at different O2 condition were not from the same origin? Can't they be a mixture of N2O from different processes? And not quite sure what you meant by '... then partially reduced'.

Independence of samples means that the samples are not temporally linked to each other. The  $N_2O$  sampled at 4.2 %  $O_2$  is not a residual fraction of  $N_2O$  produced at 16.8 % O2. We clarified these sentences (Lines 310-317).

Line 261: Concentration of NOx increased for which experiment (temperature or NH4 concentration)?

Please see below.

**Line 261: Why are these values different from the ones presented in Fig. 4c and 4d? Were you referring to the same thing?**

We are not referring to new Fig.5c and d here, but unplotted data of cumulated concentration of  $NO_2^-$  and  $NO_3^-$ . We clarified this sentence (Lines 325-326). "In the ammonium tests, the cumulated concentrations of  $NO_2^-$  and  $NO_3^-$  ([ $NO_x^-$ ]) increased from 1.4-6.1 to 5.1-19.6 mg N L-1 between inflow and outflow and were composed by at least 74 and 91 % of  $NO_3^-$ , respectively."

Line 275: I agree that there was a stronger effect of temperature on N2O emission compared to NH4+ oxidation rate but for the effect of NH4 concentration, this could only be controlled by the very high NH4+ concentration – indicating a possible effect of NH4+ concentration on these points.

We agree that very high ammonium concentrations had an important effect, however, the highest  $NH_4^+$  concentration is not the only one having an effect on the  $N_2O$  emission factor. We can see a progressive increase in  $N_2O$  emission factor with increasing  $NH_4^+$  concentration from 6 to 58 mg N  $I^{-1}$ .

Line 276: Don't think the authors can draw conclusion on the N2O processes based solely on the NH4+ oxidation and N2O emission rates. Suggest discussing these processes after the discussion on the d15N values.

The reviewer is right, however no conclusions were drawn here. We discuss all the processes at play (hydroxylamine oxidation, nitrite reduction and  $N_2O$  reduction) and suggest from  $N_2O$  emission factor that increasing temperatures or ammonium concentrations unlikely increased  $N_2O$  reduction to  $N_2$ . As suggested by the referee, the contributions of processes are discussed more in detail further, after presenting the isotope results (Lines 347-374).

**Line 280: You should also calculate nitrite oxidation rate the same way you did for ammonium to support the contention you made here.**

As part of the  $NO_2^-$  and  $NO_3^-$  produced from the oxidized ammonium could then react to other N compounds (NO,  $N_2O$ ,  $N_2$ ,  $NO_3^-$ ,  $NO_2^-$ ), we were not able to estimate the nitrite oxidation rate. The discussion about this rate is speculative from the remaining  $NO_2^-$  concentration in the reactor outflow. We clarified this throughout the manuscript when we discussed about the nitrite oxidation process (Lines 17-18, 343-346, 386-390).

**Line 294: Optimal here means? Why was 21% O2 considered optimal? Justifications/explanations are also required as to why 23.8 mg/L was chosen for the temperature experiment?**

**Please refer to a previous answer regarding the meaning of optimal O*2. We clarified this in the method section (Lines 119-123).**

Further, we chose for the temperature tests an inflow ammonium concentration close to the nominal load that receives the nitrifying unit (0.7 kg N m-3 d-1). We added this precision in the method section (Lines 115-117). "The temperature tests were carried out by cooling the feeding solution directly in the feeding tank (22.3 to 13.5 °C), with an inflow ammonium concentration close to the nominal load that received the nitrifying biomass; i.e. 20.3-21.1 mg NH4+-N L-1."

**Line 329: I assume you mean heterotrophic denitrification here? Why? You have just discussed that denitrifiers were sensitive to O2 and can be excluded as an important process contributing N2O at the high O2 conditions.**

The referee is correct, we mean heterotrophic denitrification, here and throughout the manuscript. The last step of heterotrophic denitrification (i.e.  $N_2O$  reduction to  $N_2$ ) is sensitive to  $O_2$ , however we cannot exclude that nitrite reduction performed by heterotrophic denitrifiers can occur (Lines 398-400). "This explains most likely the increased contribution of the nitrite reduction pathway to  $N_2O$  emission, as more nitrite becomes available for nitrifier-denitrification and/or heterotrophic denitrification."

**Figure 1: Very nice figure but references used for the ranges of the d15N d18O and SP should be included in the caption of the figure**

We thank the reviewer for the suggestion and improvement of the figure, we added references related to d15N, d18O and 15N-SP ranges in the caption of Figure 1.

Figure 4: Include slope, p and r2 values for each of the plot. Not clear on how the errors on the NH4+ concentration plot were derived. And why only for NH4+ conc, did you see any variations for the temperature experiment as well? You have not mentioned elsewhere that replicates samples were collected. If this was how the error bars were derived state that in the method section.

The correlations and p values were mentioned in the results and method sections, respectively (Lines 330-331 and 167-169). We specified in the method section that replicates were sampled (Lines 134-136). "The feeding solutions were characterized from 1 to 5 replicate samples collected in the feeding tank. For each tested condition, the outflow was characterized within 5 days from 1 to 14 replicate samples immediately filtered through a 0.2  $\mu$ m syringe filter and stored at 4 °C. Outflow sampling started after at least one hydraulic retention time (28 ±1 min)."

**Isotopic evidence for alteration of nitrous oxide emissions and producing pathways contribution under nitrifying conditions**

Guillaume Humbert1, 2, \*, Mathieu Sébilo1, 3, Justine Fiat4, Longqi Lang5, Ahlem Filali4, Véronique Vaury1, Mathieu Spérandio5, Anniet M. Laverman2

5 1Sorbonne Université, CNRS, INRA, IRD, UPD, UPEC, Institute of Ecology and Environmental Sciences – Paris, iEES, F-75005 Paris, France

2Centre National de la Recherche Scientifique (CNRS), ECOBIO – UMR 6553, Université de Rennes, 35042 Rennes, France 3CNRS/Univ. Pau & Pays Adour/E2S UPPA, Institut des Sciences Analytiques et de Physico-Chimie pour l'Environnement et les Matériaux, UMR 5254, 64000, Pau, France

4Irstea, UR PROSE, CS 10030, F-92761, Antony Cedex, France 5LISBP, Université de Toulouse, CNRS, INRA, INSA, Toulouse, France

Correspondence to: Guillaume Humbert (g.humbert86@gmail.com)

Abstract. Nitrous oxide (N2O) emissions by a nitrifying biofilm reactor were investigated with N2O isotopocules. The nitrogen isotopomer site preference of  $N_2O$  (15N-SP) indicated the contribution of producing and consuming pathways in response to changes in oxygenation level (from 0 to 21 % O2 in the gas mix), temperature (from 13.5 to 22.3 °C), and ammonium 15 concentrations (from 6.2 to 62.1 mg N L-1). Nitrite reduction, either nitrification or heterotrophic denitrification, was the main N2O producing pathway under the tested conditions. Difference between oxidative and reductive rates of nitrite consumption was discussed in relation to NO2- concentrations and N2O emissions. Hence, nitrite Nitrite oxidation rates seem to decreased as compared to ammonium oxidation rates at temperatures above 20 °C and under oxygen-depleted atmospheresub-optimal oxygen levels, increasing N2O production by the nitrite reduction pathway. Below 20 °C, a difference 20 in temperature sensitivity between hydroxylamine and ammonium oxidation rates is most likely responsible for an increase in the N2O production via the hydroxylamine oxidation pathway (nitrification). A negative correlation between the reaction kinetics and the apparent isotope fractionation was additionally shown from the variations of  $\delta^{15}N$  and  $\delta^{18}O$  values of N2O produced from ammonium. The approach and results obtained here, for a nitrifying biofilm reactor under variable 25 environmental conditions, should allow application and extrapolation on N2O emissions from other systems such as lakes,

soils and sediments.

**1** Introduction**

30

Nitrogen (N) cycling lies on numerous biological processes exploited and altered by anthropic activities (Bothe et al., 2007). One of the major issues related to N cycle alteration is the production of nitrous oxide ( $N_2O$ ) a potent ozone-depleting and greenhouse gas whose emissions exponentially increased during the industrial era (Crutzen et al., 1979; IPCC, 2014;

Ravishankara et al., 2009). Wastewater resource recovery facilities (WRRFs) contribute to about 3 % of annual global

anthropogenic  $N_2O$  sources (ca. 6.7  $\pm$  1.3 Tg N-N2O in 2011; IPCC, 2014); with 0 to 25 % of the influent nitrogen loads emitted as N2O (Law et al., 2012b). The challenges in mitigation of these emissions rely on the understanding of the N2O producing processes and their controls.

- 35 Two microbial processes are responsible for the production of N2O (nitrification and heterotrophic denitrification), with only one of these capable of consuming it (denitrification; Fig. 1a; Kampschreur et al., 2009). Nitrification is the oxidation of ammonium to nitrite (NO2-) via the intermediate hydroxylamine (NH2OH) conducted by ammonia oxidizers and the subsequent oxidation of NO2- to nitrate (NO3-) by nitrite oxidizers. During nitrification, N2O can be produced as reaction side-product from hydroxylamine oxidation by biotic, abiotic or hybrid processes (Caranto et al., 2016; Heil et al., 2015; Terada et al.,
- 40 2017). Heterotrophic denitrification and nitrifier-denitrification produce N2O from nitrite reduction conducted by denitrifiers and ammonium oxidizers, respectively.
- Temperature, electron donor and acceptor concentrations have been identified to control the N2O emission from WRRFs (Bollon et al., 2016; Kampschreur et al., 2009; Tumendelger et al., 2014, 2016; Wunderlin et al., 2012). These variables may induce N2O accumulation due to inhibition or disturbance of enzyme activity (Betlach and Tiedje, 1981; Kim et al., 2008; Otte
- 45 et al., 1996). In addition to this, the different N2O producing processes, nitrification, nitrifier-denitrification or heterotrophic denitrification, are rarely observed independently from each other in heterogeneous environments like wastewater, natural waters, soils or sediments. However, the understanding of the influence that environmental conditions have on the balance between these processes and the N2O producing pathways remain to a large extend-extent unexplored.
- In order to decipher N2O producing/consuming pathways, the analysis of N2O isotopocules, molecules that only differ in either the number or position of isotopic substitutions, has been applied (Koba et al., 2009; Sutka et al., 2006) (Figs. 1b-d). The isotope composition of substrates and fractionation mechanisms influence both nitrogen and oxygen isotope ratios of N2O (reported as δ15N and δ18O, respectively, Fig. 1b). Basically, the oxygen atom in the N2O molecule produced by hydroxylamine oxidation originates from atmospheric dissolved oxygen with δ18O of 23.5 ‰ (Andersson and Hooper, 1983; Hollocher et al., 1981; Kroopnick and Craig, 1972), while the oxygen atom in N2O produced by nitrite reduction originates from nitrite that
- 55 has undergone oxygen-exchange with water (Kool et al., 2007; Snider et al., 2012). Nonetheless, the δ18O-N2O resulting from the nitrite reduction conducted by the nitrifiers ranges from 13 to 35 ‰ (Snider et al., 2012). In contrast, the N2O produced by the heterotrophic denitrifiers through the nitrite reduction pathway has δ18O over 35 ‰ (Snider et al., 2013). However, the O-exchange between the N2O precursors and water can decrease it to values below 35 ‰ (Snider et al., 2015). Therefore, the δ18O alone does not enable differentiation between the N2O producing pathways.
- 60 In combination with δ18O, the δ15N-N2O allows to identify the N2O producing pathways (Fig. 1b). However, the isotope fractionations (or isotope effects) largely influence the δ15N-N2O due to wide variations between and within the reactions involved in the nitrogen cycle (Denk et al., 2017). The isotopic fractionations results from the difference in equilibrium constant (abiotic process) or reaction rate (biotic process) observed between the heavier and lighter isotopes in both abiotic and biotic processes. The net isotope effects (Δ) approximated from the difference between δ15N of product and substrate characterize

- 65 the production of compounds resulting from sequential or branched reactions and have been recently reviewed (Denk et al., 2017; Toyoda et al., 2017). So far, only two estimates of the net isotope effect of N2O production by ammonium oxidation via hydroxylamine of -46.9.5 and -32.6.9 ‰ have been proposed (Sutka et al., 2006; Yamazaki et al., 2014). These values are imbricated between -52.8 and -6 ‰, the range of net isotope effects related to the N2O production through nitrite reduction performed by nitrifiers or heterotrophic denitrifiers (Lewicka-Szczebak et al., 2014; Sutka et al., 2008).
- 70 Similarly to isotope ratios, the nitrogen isotopomer site preference (15N-SP), the difference between the relative abundances of N2O molecules enriched in 15N atthe central (Nα) position and in the terminal (Nβ) position, differ according to N2O producing pathway (Figs. 1c and d). During heterotrophic or nitrifier-denitrification the 15N-SP of N2O produced from nitrate or nitrite ranges from -10.7 to 0.1 ‰, while ranging from 13.1 to 36.6 ‰ when N2O results from hydroxylamine oxidation (Frame and Casciotti, 2010; Jung et al., 2014; Sutka et al., 2006; Yamazaki et al., 2014). Finally, N2O reduction to N2 by
- 75 heterotrophic denitrifiers increases the values of  $\delta^{15}$ N,  $\delta^{18}$ O and  $^{15}$ N-SP of residual N2O with specific pairwise ratios (Jinuntuya-Nortman et al., 2008; Webster and Hopkins, 1996; Yamagishi et al., 2007).
- Nitrogen and oxygen isotope ratios of N2O have are often disregarded, due to lower potential for N2O source identification as compared to 15N-SP. However, we believe that the use of both isotope approaches should strengthen the conclusions from 15N-SP and reveal additional isotope effects (Fig. 1).
- 80 The aim of the current study is to improve our understanding regarding the effects of key environmental variables (oxygenation, temperature, NH4+ concentrations) on N2O production and emission rates. More specifically using nitrogen and oxygen isotope ratios as well as 15N-SP of N2O should allow deciphering the different producing and consuming pathways under these different conditions. In order to achieve this, the nitrifying biomass of a submerged fixed-bed biofilm reactor was investigated. Among the wastewater treatment systems, the biofilm systems are adapted to large urban areas owing to their compactness, flexibility
- 85 and reliability. An increase of their development is expected in response to the additional 2.5 billion humans predicted in urban areas by 2050 (United Nations Population Division, 2019). However, the biofilm systems have received much less attention than the suspended biomass systems and the relations between the N2O producing/consuming pathways and controls remain largely unknown (Sabba et al., 2018; Todt and Dörsch, 2016). Although applied here to the nitrifying biomass of a WRRF, T the research questions addressed are regard a diversity of environments including natural waters, soils and sediments; i) Does
- 90 the nitrifying biomass emit N2O and what are the producing pathways at play?; ii) Do oxygenation, temperature, and NH4+ concentration alter the N2O emissions, and what are the involved processes? We hypothesize that the isotope signature of N2O allows identifying the N2O origins and the assessment of pathway contribution to N2O emissions. The results of this study should improve the mechanistic understanding as well as improved prediction of N2O emissions from WRRFs, currently suffering from high uncertainty.

Figure 1: N2O producing and consuming pathways at play during nitrification and heterotrophic denitrification. Substrate isotope composition, isotope effects and 15N-SP values from the literature were used to propose the ranges of 15N [Lewicka-Szczebak et al., 2014; Sutka et al., 2006, 2008; Yamazaki et al., 2014], 18O (Andersson and Hooper, 1983; Hollocher et al., 1981; Kool et al., 2007; Kroopnick and Craig, 1972; Snider et al., 2012), and 15N-SP (Frame and Casciotti, 2010; Jung et al., 2014; Sutka et al., 2006; Yamazaki et al., 2014), as well the slopes relating them with each other during N2O reduction to N2 (Jinuntuva-Nortman et al., 2008; Webster and Hopkins, 1996; Yamagishi et al., 2007). The assumptions made and the calculations performed are detailed in the text.

**95 2 Material and methods**

100

**2.1 Experimental setup for nitrifying experiments**

Experiments were carried out with colonized polystyrene beads (diameter 4 mm) sampled from the nitrification biologically active filters (BAF) of a domestic WRRF (Seine Centre, France). In this WRRF, wastewater (240,000 m3 d-1) passes through a pre-treatment stage, followed by a physicochemical decantation, and tertiary biological treatment. The latter is composed of three biofiltration processes; (i) carbon elimination (24 Biofor®), (ii) nitrification (29 Biostyr®), and (iii) denitrification (12

Biofor®). Nitrifying Biostyrs® are submerged fixed-bed biofilm reactors with a unitary section of 111 m2 and a filter bed of 3 m high. This unit is operated to receive a nominal load of 0.7 kg NH4+-N m-3 d-1.

A lab-scale reactor with a working volume of 9.9 L (colonized Biostyrene® beads and interstitial volume) and a headspace of 1.4 L was operated in continuous down-flow counter-current mode for seven weeks (i.e. solution was down-flowing, while air
 105 was up-flowing; Fig. S1). Mass flow meters (F-201CV, Bronkhorst, France) sustained the inflow gas rate at 0.5 L min-1. A

peristaltic pump (R3425H12B, Sirem, France) pumped feeding solution from a feeding tank into the reactor at 0.2 L min-1, in order to maintain a hydraulic retention time (HRT) of  $27.8 \pm 0.6$  min. A water jacket monitored by a cryogenic regulator (WK

|---------------------------------------------------------|

500, Lauda, Germany) controlled the reactor temperature. The feeding solution consisted of ammonium chloride (NH4Cl) as substrate, monobasic potassium phosphate (KH2PO4) as phosphorus source for bacterial growth, and sodium hydrogen
110 carbonate (NaHCO3) as pH buffer and inorganic carbon source in 100 or 150 L of tap water (average 0.2 ± 0.4, 2.4 ± 1.1, and 2.5 ± 1.3 mg N L-1 of NO2-, NO3- and sum of both NO8- molecules, respectively).

The influence of environmental conditions on the ammonium oxidation rates and the N2O emissions from various combinations of oxygenation levels, temperatures and ammonium concentrations were tested in twenty-four experiments (Tables 1-and S1). Note that two of them were used twice; as oxygenation and concentration tests. The oxygenation tests consisted of were carried

- 115 out by mixing compressed air and pure nitrogen gas to reach 0 to 21 % O2 in the gas mixture (Fig. S1aS2a). They The tests were performed at five substrate concentrations and at a temperature between 19.2 and 20.6 °C. The temperature tests <del>consisted of were carried out by cooling the feeding solution directly in the feeding tank (22.3 to 13.5 °C), with an inflow ammonium concentration close to the nominal load that received the nitrifying biomass; i.e.of 19.920.3-21.1 mg NH4+-N L-1. At temperatures ranging from 18.8 to 19.9 °C, the The ammonium concentration tests <del>consisted of were run at</del> an increase (6.2, 10.1 mg NH4+-N L-1).</del>
- 120 28.6 and 62.1 mg NH4+-N L1) and a decrease (56.1, 42.9, 42.7 and 20.2 mg NH4+-N L-1) in the NH4+ concentrations in the feeding solution, at temperatures ranging from 19.0 to 19.8 °C. An optimal Atmospheric oxygenation level (i.e. 21 % O2 in the gas mixture) was imposed for both tests (Figs. S1b-S2b and c). This gas mixture using compressed air with 21 % O2 was considered hereafter as optimal as compared to the oxygen-depleted atmosphere used during the oxygenation tests. Noticeably, the atmospheric oxygenation level is the condition that represents the most optimal conditions of oxygenation applied in
- 125
   nitrification BAF of domestic WRRF. Sampling started after at least one hydraulie retention time (28 ±1 min).

   Table 1. Detailed average conditions (± standard deviation) of oxygenation, temperature and concentration tests.

| inflow [NH4 + ] | inflow gas rate | O2 in gas mix | temperature |
|----------------------------|-----------------|---------------|--------------------|
| $mg N L^{-1}$              | $L \min^{-1}$   | %      | °C          |
|                            | oxygenat        |               |                    |
| 25.1 ±0.5           | 0.4             | 0      | 19.2 ±0.1   |
| 23.8 ±0.6           | 0.53            | 4.2           | 19.9 ±0.1   |
| 25.1 ±0.5           | 0.53            | 4.2           | 19.2 ±0.1   |
| 37.3 ±0.6           | 0.5      | 4.2           | 20.5 ±0.1   |
| 23.8 ±0.6           | 0.51            | 10.5          | 20.2 ±0.1   |
| 25.1 ±0.5           | 0.51            | 10.5          | 19.2 ±0.1   |
| 37.3 ±0.6           | 0.5             | 10.5          | 20.6 ±0.1          |
| 23.8 ±0.6           | 0.5             | 16.8          | 20.1 ±0.1   |
| 25.1 ±0.5           | 0.5      | 16.8   | 19.3 ±0.1   |
| 37.3 ±0.6           | 0.5             | 16.8          | 20.6 ±0.1   |
| 20.2 ±0.5           | 0.5      | 21     | 19.5 ±0.1   |

| 25.1 ±0.5  | 0.57                       | 21       | 19.6 ±0.5 |
|-------------------|----------------------------|-----------------|------------------|
| 28.6 ±0.5  | 0.5                 | 21       | 19.6 ±0.1 |
|                   |                            |                 |                  |
| 20.3 ±0.3  | 0.5                        | 21       | 13.5 ±0.2 |
| 21.1 ±n.a. | 0.5                        | 21       | 15.5 ±0.1 |
| 21.1 ±n.a. | 0.5                        | 21       | 16.2 ±0.1 |
| 20.3 ±0.3  | 0.5                        | 21       | 18.2 ±0.1 |
| 21.1 ±n.a. | 0.5                        | 21       | 20.3 ±0.1 |
| 20.3 ±0.3  | 0.5                        | 21       | 22.3 ±0.1 |
|                   | $\underline{NH_4^+ conce}$ | entration tests |                  |
| 6.2 ±0.1   | 0.5                        | 21       | 19.6 ±0.0 |
| 20.2 ±0.5  | 0.5                        | 21       | 19.5 ±0.1 |
| 28.6 ±0.5  | 0.5                        | 21       | 19.6 ±0.1 |
| 42.7 ±1.0         | 0.5                        | 21       | 19.3 ±0.0 |
| 42.9              | 0.5                        | 21       | 19.0 ±0.0 |
| 56.1 ±0.3  | 0.5                 | 21       | 19.0 ±0.1 |
| 62.1 ±0.4  | 0.5                 | 21       | 19.8 ±0.0 |

Note that two experiments tested both oxygenation and ammonium concentration.

Ranges of environmental conditions tested.

| tests         | inflow [NH4+]          | inflow gas rate      | O 2 -in gas mixture | temperature            |
|---------------|------------------------|----------------------|--------------------------------|------------------------|
| -             | mg N L -1   | Lmin -1   | %                       | • <del>C</del>         |
| oxygenation   | 20.2 - 37.3     | 0.4 - 0.57           | <del>0-21</del>                |  19.2 - 20.6    |
| temperature   | <del>20.2 - 21.1</del> | <del>0.5</del>       | 21                             | <del>13.5 - 22.3</del> |
| inflow [NH4+] | <del>6.2 - 62.1</del>  | <del>0.5, 0.57</del> | <del>21</del>                  | <del>19.0 - 20.3</del> |
|               |                        |                      |                                |                        |

**2.2 Reactor monitoring, sampling and concentrations analysis**

Dissolved oxygen, temperature (Visiferm DO Arc 120, Hamilton, Switzerland) and pH (H8481 HD, SI Analytics, France) 130 were continuously measured at the top of the reactor and data were recorded at 10 second intervals. The N2O concentration was continuously analyzed by an infrared photometer (Rosemount™ X-STREAM X2GP, Emerson, Germany) in outflow reactor gas after drying through a condenser and a hydrophobic gas filter (0.2 µm). Minute averages are used for monitored data hereafter. Gas samples were taken for N2O isotopic signature determination by outlet gas pipe derivation into a sealed glass vial of 20 ml. The vial was first flushed with the sampling gas for > 45 sec prior to 1-5 min sampling. Gas samples were 135 then stored in the dark at room temperature until analysis. Note that gas sampling was lacking for 5 of the 13 oxygenation tests.

The feeding solutions were characterized from 1 to 5 replicate samples collected in the feeding tank. For each tested condition, the outflow was characterized within 5 days from 1 to 14 replicate samples immediately filtered through a 0.2  $\mu$ m syringe filter and stored at 4 °C. Outflow Sampling started after at least one hydraulic retention time (28 ±1 min). Ammonium was analyzed

using the Nessler colorimetric method, according to AFNOR NF T90-015 (DR 2800, Hach, Germany). Nitrite and nitrate were

140

**2.3 Stable isotope measurements**

measured by ionic chromatography (IC25, Dionex, USA).

Atmospheric  $N_2$  and Vienna Standard Mean Ocean Water (VSMOW) are the references used for the nitrogen and oxygen isotopes ratios, respectively, expressed in the conventional  $\delta$ -notation, in per-mil (‰). Nitrogen and oxygen isotope ratios of

- 145 nitrate and nitrite were determined separately following a modified protocol of McIlvin and Altabet (McIlvin and Altabet, 2005; Semaoune et al., 2012). Nitrogen isotope ratios of ammonium were determined following the protocol of Zhang et al. (2007). These methods consist in the conversion of the substrate (ammonium or nitrite or nitrate) into dissolved N2O. The δ15N and δ18O for ammonium, nitrite, and nitrate were hence determined from a calibration curve created with a combination of nitrate or ammonium standards that have undergone the same chemical conversion as the samples (USGS-32, δ15N-NO3- = 180 ‰, δ18O-NO3- = 25.7 ‰; USGS-34, δ15N-NO3- = -1.8 ‰, δ18O-NO3- = -7.9 ‰ and USGS-35 δ15N-NO3- = 2.7 ‰, δ18O-NO3- = -1.8 ‰, δ18O-NO3- = -7.9 ‰ and USGS-35 δ15N-NO3- = -1.8 ‰, δ18O-NO3- = -7.9 ‰ and USGS-35 δ15N-NO3- = -1.8 ‰, δ18O-NO3- = -7.9 ‰ and USGS-35 δ15N-NO3- = -1.8 ‰, δ18O-NO3- = -7.9 ‰ and USGS-35 δ15N-NO3- = -1.8 ‰, δ18O-NO3- = -7.9 ‰ and USGS-35 δ15N-NO3- = -1.8 ‰, δ18O-NO3- = -7.9 ‰ and USGS-35 δ15N-NO3- = -1.8 ‰, δ18O-NO3- = -7.9 ‰ and USGS-35 δ15N-NO3- = -1.8 ‰, δ18O-NO3- = -7.9 ‰ and USGS-35 δ15N-NO3- = -1.8 ‰, δ18O-NO3- = -7.9 ‰ and USGS-35 δ15N-NO3- = -1.8 ‰, δ18O-NO3- = -7.9 ‰ and USGS-35 δ15N-NO3- = -7.9 ‰
- NO3- = 57.5 ‰; or IAEA-N1, δ15N-NH4+ = 0.4 ‰, IAEA-305A, δ15N-NH4+ = 39.8 ‰, USGS-25, δ15N-NH4+ = -30.4 ‰). The quality of calibration was controlled with additional international standards (IAEA-NO-3, δ15N-NO3- = 4.7 ‰, δ18O-NO3- = 25.6 ‰; or IAEA-N2, δ15N-NH4+ = 20.3 ‰). Basically, an analytical sequence was comprised of triplicate standards for calibration and quality controls and duplicate samples. The average of the analytical replicates was then used for calibration, quality control and as result.

Since no international standards were available for N2O isotopes, these were determined the same day as nitrate or ammonium standard analysis insuring correct functioning of the method and analysis. In addition to this, the internal N2O standards were previously calibrated by exchange with the laboratory of Naohiro Yoshida and Sakae Toyoda at the Tokyo Institute of Technology. All isotope measurements were determined using an isotope ratio mass spectrometer (IRMS, DeltaVplus; Thermo

Scientific) in continuous-flow with a purge and trap system coupled with a Finnigan GasBench II system (Thermo Scientific). The δ15N and δ18O values of N2O and 15N site preference (15N SP) values were determined using an isotope ratio mass spectrometer (IRMS, DeltaVplus; Thermo Scientific) in continuous flow with a purge and trap system coupled with a Finnigan GasBench II system (Thermo Scientific). The method was calibrated with combination of nitrate or ammonium standards (USGS 32, δ15N NO3- = 180 ‰, δ18O NO3- = 25.7 ‰; USGS 34, δ15N NO3- = -1.8 ‰, δ18O NO3- = -27.9 ‰ and USGS 35
δ15N NO3- = 2.7 ‰, δ18O NO3- = 57.5 ‰; or IAEA N1, δ15N NH4+ = -0.4 ‰, IAEA -305A, δ15N NH4+ = -39.8 ‰, USGS 25, δ15N NH4+ = -30.4 ‰). Linearity of the analysis was checked with international standards (IAEA NO 3, δ15N NO3- = -4.7 ‰).

 $\delta^{14}$ O-NO3- = 25.6 ‰; or IAEA-N2,  $\delta^{15}$ N-NH4+ = 20.3 ‰). The precision was 0.8 ‰, 1.5 ‰ and 2.5 ‰ for  $\delta^{15}$ N,  $\delta^{18}$ O, and 15N-SP, respectively.

**2.4 Data processing and statistics**

- 170 The effects of environmental conditions on nitrification were assessed from 4 indices. The ammonium oxidation rate (AOR) was estimated in each experiment for time  $\geq 1$  HRT from the difference between influent and effluent NH4+ concentrations multiplied by the liquid flow rate (kg NH4+-N  $\frac{m^2}{m^2}$ -d-1). The nitrification efficiency was defined as the ratio between AOR and influent ammonium load. The N2O emission rate (N2O-ER) was calculated by multiplying the measured N2O concentration by the gas flow rate (mg N2O-N min-1). The N2O emission factor (N2O-EF) was defined as the ratio between N2O-ER and AOR
- 175 (% of oxidized NH4+-N). The measurements related to liquid or gas samples were averaged by experiment; i.e. average of data obtained from the samples collected after one hydraulic retention time.

Statistical analysis were performed using the R software (R Core Team, 2014). The value of 0.05 was used as significance level for spearman correlations (*cor.test* function) and linear regressions (*lm* function). *Adjusted*  $r^2$  was provided as  $r^2$  for the latter.

**180 2.5 Estimation of ranges of nitrogen isotope ratio in biologically produced N2O**

As shown in Fig. 1, the pairwise relationships between  $\delta^{15}N$ ,  $\delta^{18}O$  and  $^{15}N$ -SP assist the determination of the producing and consuming pathways at play. The N atoms that compose the N2O molecule originate from NH4+ molecules when produced by hydroxylamine oxidation, while originating from the N atoms of NO3- or NO2- molecules when produced by nitrite reduction (NOx- molecules). However, the nitrogen isotope ratio of N2O does not equal those of its substrates as it depends on isotope effects associated to each reaction step of N2O producing process. The isotope effect of reaction step can be determined from the isotope composition of substrates or products. Although being performed on a few tests here, the obtained value can only be applied to a limited number of environmental conditions. The use of estimates from the literature seems therefore suitable. Several equations enable to approximate the isotope effect and its effect on the isotope ratios of substrate and product pools

- involved in a reaction. These equations vary according to the assumptions made on the system boundaries (Denk et al., 2017).
   The nitrifying reactor used in this study can be described as an open-system continuously supplied by an infinite substrate pool with constant isotopic composition (NH4±,m). A small amount of the infinite substrate pool is transformed into a product pool (NOx2,p) or a residual substrate pool (NH4±,m) when flowing through the system. The equations describing the input, output and processes considered here are presented in Fig. 2 after Fry (2006). Note that the definitions of f and Δ are inverse to the cited literature and that Δ1 and Δ4 are null because no fractionation alter the residual substrate exiting the reaction (Fry, 2006).
- 195

|     | $NH_{4^*,jn} \xrightarrow{Nitrification} NH_{4^*,pot} \xrightarrow{f_1} NH_{4^*,pot} \xrightarrow{NH_{4^*,pot}} NH_{4^*,pot} \xrightarrow{NItrification/i} \overset{I_1 \cdot f_2 \cdot f_3}{\overset{I_2 \cdot f_2 \cdot f_3}} \xrightarrow{NH_{4^*,pot}} NO_{4^*,pot} \xrightarrow{I_3 (1 - f_1 \cdot f_3)} NO_{4^*,pot}$                                                                                                                                                                                                                                                                                                                                                                                                                                                                                                                                                                                                                                                                                                                                                                                                                                                                                                                                                                                                                                                                                                                                                                                                                                                                                                                                                                                                                                                                                                                                                                                                                                                                                                                                                                                                                                                                                                                                                                                                                                                                                                                                                                                                                                                                                                                                                                                                                                             | Mis en forme : Paragraphes solidaires |
|-----|----------------------------------------------------------------------------------------------------------------------------------------------------------------------------------------------------------------------------------------------------------------------------------------------------------------------------------------------------------------------------------------------------------------------------------------------------------------------------------------------------------------------------------------------------------------------------------------------------------------------------------------------------------------------------------------------------------------------------------------------------------------------------------------------------------------------------------------------------------------------------------------------------------------------------------------------------------------------------------------------------------------------------------------------------------------------------------------------------------------------------------------------------------------------------------------------------------------------------------------------------------------------------------------------------------------------------------------------------------------------------------------------------------------------------------------------------------------------------------------------------------------------------------------------------------------------------------------------------------------------------------------------------------------------------------------------------------------------------------------------------------------------------------------------------------------------------------------------------------------------------------------------------------------------------------------------------------------------------------------------------------------------------------------------------------------------------------------------------------------------------------------------------------------------------------------------------------------------------------------------------------------------------------------------------------------------------------------------------------------------------------------------------------------------------------------------------------------------------------------------------------------------------------------------------------------------------------------------------------------------------------------------------------------------------------------------------------------------------------------------------------------------------------------------------------------------------------------------------------------------------------------------------------------------------------|---------------------------------------|
|     | $NO_{x,in} \longrightarrow NO_{x,in} \longrightarrow (1-f_3)(1-f_1-f_2) \longrightarrow (1-f_3)(1-f_3)(1-f_1-f_3) \longrightarrow (1-f_3)(1-f_3)(1-f_3)(1-f_3)(1-f_3)(1-f_3)(1-f_3)(1-f_3)(1-f_3)(1-f_3)(1-f_3)(1-f_3)(1-f_3)(1-f_3)(1-f_3)(1-f_3)(1-f_3)(1-f_3)(1-f_3)(1-f_3)(1-f_3)(1-f_3)(1-f_3)(1-f_3)(1-f_3)(1-f_3)(1-f_3)(1-f_3)(1-f_3)(1-f_3)(1-f_3)(1-f_3)(1-f_3)(1-f_3)(1-f_3)(1-f_3)(1-f_3)(1-f_3)(1-f_3)(1-f_3)(1-f_3)(1-f_3)(1-f_3)(1-f_3)(1-f_3)(1-f_3)(1-f_3)(1-f_3)(1-f_3)(1-f_3)(1-f_3)(1-f_3)(1-f_3)(1-f_3)(1-f_3)(1-f_3)(1-f_3)(1-f_3)(1-f_3)(1-f_3)(1-f_3)(1-f_3)(1-f_3)(1-f_3)(1-f_3)(1-f_3)(1-f_3)(1-f_3)(1-f_3)(1-f_3)(1-f_3)(1-f_3)(1-f_3)(1-f_3)(1-f_3)(1-f_3)(1-f_3)(1-f_3)(1-f_3)(1-f_3)(1-f_3)(1-f_3)(1-f_3)(1-f_3)(1-f_3)(1-f_3)(1-f_3)(1-f_3)(1-f_3)(1-f_3)(1-f_3)(1-f_3)(1-f_3)(1-f_3)(1-f_3)(1-f_3)(1-f_3)(1-f_3)(1-f_3)(1-f_3)(1-f_3)(1-f_3)(1-f_3)(1-f_3)(1-f_3)(1-f_3)(1-f_3)(1-f_3)(1-f_3)(1-f_3)(1-f_3)(1-f_3)(1-f_3)(1-f_3)(1-f_3)(1-f_3)(1-f_3)(1-f_3)(1-f_3)(1-f_3)(1-f_3)(1-f_3)(1-f_3)(1-f_3)(1-f_3)(1-f_3)(1-f_3)(1-f_3)(1-f$ |                                       |
|     | Equations: Nitrification                                                                                                                                                                                                                                                                                                                                                                                                                                                                                                                                                                                                                                                                                                                                                                                                                                                                                                                                                                                                                                                                                                                                                                                                                                                                                                                                                                                                                                                                                                                                                                                                                                                                                                                                                                                                                                                                                                                                                                                                                                                                                                                                                                                                                                                                                                                                                                                                                                                                                                                                                                                                                                                                                                                                                                                                                                                                                                         |                                       |
|     | $\begin{split} \delta^{15} \mathbf{N} - \mathbf{N} \mathbf{H}_{4}^{*}{}_{jee} &\approx \delta^{15} \mathbf{N} - \mathbf{N} \mathbf{H}_{4}^{*}{}_{jee} - \Delta_{2} (1 - f_{1}) + (\Delta_{3} - \Delta_{2}) f_{2} & \\ \delta^{15} \mathbf{N} - \mathbf{N} \mathbf{O}_{a}^{*}{}_{jee} &\approx \delta^{15} \mathbf{N} - \mathbf{N} \mathbf{H}_{4}^{*}{}_{jee} - \Delta_{2} f_{1} + (\Delta_{3} - \Delta_{2}) f_{2} & \\ \delta^{15} \mathbf{N} - \mathbf{N} \mathbf{O}_{a}^{*}{}_{jee} &\approx \delta^{15} \mathbf{N} - \mathbf{N} \mathbf{O}_{a}^{*}{}_{jee} (1 - f_{1} - f_{2})^{-1} - \Delta_{3} (1 - f_{1}) \\ \delta^{15} \mathbf{N} - \mathbf{N} \mathbf{O}_{a}^{*}{}_{jee} &\approx \delta^{15} \mathbf{N} - \mathbf{N} \mathbf{O}_{a}^{*}{}_{jee} (1 - f_{1} - f_{2})^{-1} - \Delta_{3} (1 - f_{1}) \\ \delta^{15} \mathbf{N} - \mathbf{N}_{2} \mathbf{O} \approx \delta^{15} \mathbf{N} - \mathbf{N} \mathbf{O}_{a}^{*}{}_{jee} (1 - f_{1} - f_{2})^{-1} + f_{3} \Delta_{3} \end{split}$                                                                                                                                                                                                                                                                                                                                                                                                                                                                                                                                                                                                                                                                                                                                                                                                                                                                                                                                                                                                                                                                                                                                                                                                                                                                                                                                                                                                                                                                                                                                                                                                                                                                                                                                                                                                                                                                                                                                |                                       |
|     | $\delta^{15} N - NO_{x',yet} \approx \frac{(1 + 1 + 1 + y_{x}) - (1 + 1 + 1 + 1 + 1 + 1 + 1 + 1 + 1 + 1 $                                                                                                                                                                                                                                                                                                                                                                                                                                                                                                                                                                                                                                                                                                                                                                                                                                                                                                                                                                                                                                                                                                                                                                                                                                                                                                                                                                                                                                                                                                                                                                                                                                                                                                                                                                                                                                                                                                                                                                                                                                                                                                                                                                                                                                                                                                                                                                                                                                                                                                                                                                                                                                                                                                                                                                                                                        |                                       |
|     | Figure 2. Diagram and equations of the nitrifying reactor after Fry (2006). It is considered as a sequence of two reactor boxes. (i) $\leftarrow$
The nitrification of inflow ammonium (NH 4 $\pm_{in}$ ) to a pool of nitrite and nitrate (NO 3 $\pm_{in}$ ), residual ammonium (NH 4 $\pm_{ires}$ ), and nitrous
oxide (N 2 O) through the hydroxylamine oxidation pathway. (ii) The subsequent reduction of intermediate NO 3 $\pm_{int}$ ; mixing of
inflow NO 3 $\pm_{in}$ and formed NO 3 $\pm_{in}$ to nitrous oxide (N 2 O) through the nitrite reduction pathway, and residual NO 3 $\pm$ that exits the                                                                                                                                                                                                                                                                                                                                                                                                                                                                                                                                                                                                                                                                                                                                                                                                                                                                                                                                                                                                                                                                                                                                                                                                                                                                                                                                                                                                                                                                                                                                                                                                                                                                                                                                                                                                                                                                                                                                                                                                                                                                                                                                                                                                    | Mis en forme : Légende                |
|     | reactor (NO x2,out ). Note that residual substrates and formed products exit the reactor without further isotope fractionation ( $\Delta_1$ and $\Delta_4$ are null). See text for details.                                                                                                                                                                                                                                                                                                                                                                                                                                                                                                                                                                                                                                                                                                                                                                                                                                                                                                                                                                                                                                                                                                                                                                                                                                                                                                                                                                                                                                                                                                                                                                                                                                                                                                                                                                                                                                                                                                                                                                                                                                                                                                                                                                                                                                                                                                                                                                                                                                                                                                                                                                                                                                                                                                                |                                       |
|     | The balance between input and output of each reactional step allows to propose equations for calculation of the nitrogen isotope                                                                                                                                                                                                                                                                                                                                                                                                                                                                                                                                                                                                                                                                                                                                                                                                                                                                                                                                                                                                                                                                                                                                                                                                                                                                                                                                                                                                                                                                                                                                                                                                                                                                                                                                                                                                                                                                                                                                                                                                                                                                                                                                                                                                                                                                                                                                                                                                                                                                                                                                                                                                                                                                                                                                                                                                 |                                       |
|     | ratio of compounds in the inflow and outflow of the system (Denk et al., 2017; Fry, 2006). These equations can be simplified                                                                                                                                                                                                                                                                                                                                                                                                                                                                                                                                                                                                                                                                                                                                                                                                                                                                                                                                                                                                                                                                                                                                                                                                                                                                                                                                                                                                                                                                                                                                                                                                                                                                                                                                                                                                                                                                                                                                                                                                                                                                                                                                                                                                                                                                                                                                                                                                                                                                                                                                                                                                                                                                                                                                                                                                     |                                       |
|     | under the assumption that limited amount of N compounds are transformed into $N_2O$ ; i.e. $f_2$ close to 0 and $f_3$ close to 1.                                                                                                                                                                                                                                                                                                                                                                                                                                                                                                                                                                                                                                                                                                                                                                                                                                                                                                                                                                                                                                                                                                                                                                                                                                                                                                                                                                                                                                                                                                                                                                                                                                                                                                                                                                                                                                                                                                                                                                                                                                                                                                                                                                                                                                                                                                                                                                                                                                                                                                                                                                                                                                                                                                                                                                                                |                                       |
|     | Therefore, the N isotope ratios of the residual substrate pool can be approximated from Eq. (1).                                                                                                                                                                                                                                                                                                                                                                                                                                                                                                                                                                                                                                                                                                                                                                                                                                                                                                                                                                                                                                                                                                                                                                                                                                                                                                                                                                                                                                                                                                                                                                                                                                                                                                                                                                                                                                                                                                                                                                                                                                                                                                                                                                                                                                                                                                                                                                                                                                                                                                                                                                                                                                                                                                                                                                                                                                 |                                       |
| 200 | $\delta^{15}$ N-NH 4 + , res $\approx \delta^{15}$ N-NH 4 + , in - $\Delta_2(1 - f_1)$ , (1)                                                                                                                                                                                                                                                                                                                                                                                                                                                                                                                                                                                                                                                                                                                                                                                                                                                                                                                                                                                                                                                                                                                                                                                                                                                                                                                                                                                                                                                                                                                                                                                                                                                                                                                                                                                                                                                                                                                                                                                                                                                                                                                                                                                                                                                                                                                                                                                                                                                                                                                                                                                                                                                                                                                                                                   |                                       |
|     | Where $f_1$ is the remaining substrate fraction leaving the reactor (i.e. remaining fraction of ammonium), ranging from 0 to 1 (0-                                                                                                                                                                                                                                                                                                                                                                                                                                                                                                                                                                                                                                                                                                                                                                                                                                                                                                                                                                                                                                                                                                                                                                                                                                                                                                                                                                                                                                                                                                                                                                                                                                                                                                                                                                                                                                                                                                                                                                                                                                                                                                                                                                                                                                                                                                                                                                                                                                                                                                                                                                                                                                                                                                                                                                                               |                                       |
|     | 100 in %), and $\Delta_2$ is the N isotope enrichment factor associated with ammonium oxidation. In their review, Denk et al. (2017)                                                                                                                                                                                                                                                                                                                                                                                                                                                                                                                                                                                                                                                                                                                                                                                                                                                                                                                                                                                                                                                                                                                                                                                                                                                                                                                                                                                                                                                                                                                                                                                                                                                                                                                                                                                                                                                                                                                                                                                                                                                                                                                                                                                                                                                                                                                                                                                                                                                                                                                                                                                                                                                                                                                                                                                             |                                       |
|     | reported a mean value of -29.6 $\pm$ 4.9 ‰ for $\Delta_2$ . Therefore, $\delta^{15}$ N is higher for residual than the initial substrate pool ( $\delta^{15}$ N-                                                                                                                                                                                                                                                                                                                                                                                                                                                                                                                                                                                                                                                                                                                                                                                                                                                                                                                                                                                                                                                                                                                                                                                                                                                                                                                                                                                                                                                                                                                                                                                                                                                                                                                                                                                                                                                                                                                                                                                                                                                                                                                                                                                                                                                                                                                                                                                                                                                                                                                                                                                                                                                                                                                                                                 |                                       |
|     | $NH_{d^{\pm}in} < \delta^{15}N-NH_{d^{\pm}res}$ . Consequently, the pool of product is depleted in heavier isotope (i.e. nitrite and nitrate hereafter